# A Hierarchical Bayesian Approach to Federated Learning

## Abstract

We propose a novel hierarchical Bayesian approach to Federated Learning (FL), where our model reasonably describes the generative process of clients' local data via hierarchical Bayesian modeling: constituting random variables of local models for clients that are governed by a higher-level global variate. Interestingly, the variational inference in our Bayesian model leads to an optimisation problem whose block-coordinate descent solution becomes a distributed algorithm that is separable over clients and allows them not to reveal their own private data at all, thus fully compatible with FL. We also highlight that our block-coordinate algorithm has particular forms that subsume the well-known FL algorithms including Fed-Avg and Fed-Prox as special cases. That is, we not only justify the previous Fed-Avg and Fed-Prox algorithms whose learning protocols look intuitive but theoretically less underpinned, but also generalise them even further via principled Bayesian approaches. Beyond introducing novel modeling and derivations, we also offer convergence analysis showing that our block-coordinate FL algorithm converges to an (local) optimum of the objective at the rate of $O(1/\sqrt{t})$, the same rate as regular (centralised) SGD, as well as the generalisation error analysis where we prove that the test error of our model on unseen data is guaranteed to vanish as we increase the training data size, thus asymptotically optimal.

## 1 Introduction

Federated Learning (FL) aims to enable a set of clients to collaboratively train a model in a privacy preserving manner, without sharing data with each other or a central server. Compared to conventional centralised optimisation problems, FL comes with a host of statistical and systems challenges – such as communication bottlenecks and sporadic participation. The key statistical challenge is non-i.i.d. data distributions across clients, each of which has a different data collection bias and potentially a different data annotation policy/labeling function – for example, in the case of any user preference learning. The classic and most popularly deployed FL algorithms are Fed-Avg (McMahan et al., 2017) and Fed-Prox (Li et al., 2018), however, even when a global model can be learned, it often underperforms on each client's local data distribution in scenarios of high heterogeneity (Li et al., 2019; Karimireddy et al., 2019; Wang et al., 2020). Studies have attempted to alleviate this by personalising learning at each client, allowing each local model to deviate from the shared global model Sun et al. (2021). However, this remains challenging given that each client may have a limited amount of local data for personalised learning.

These challenges have motivated several attempts to model the FL problem from a Bayesian perspective. Introducing distributions on model parameters $\theta$ has enabled various schemes for estimating a global model posterior $p(\theta|D_{1:N})$ from clients' local posteriors $p(\theta|D_i)$, or to regularise the learning of local models given a prior defined by the global model Zhang et al. (2022); Al-Shedivat et al. (2021); Chen & Chao (2021). However, these methods are not complete and principled solutions – having not yet have provided full Bayesian descriptions of the FL problem, and having had resort to ad-hoc treatments to achieve tractable learning. The key difference is that they fundamentally treat network weights $\theta$ as a random variable shared across all clients. We introduce a *hierarchical* Bayesian model that assigns each client it's own random variable for model weights $\theta_i$, and these are linked via a higher level random variable $\phi$ as $p(\theta_{1:N}, \phi) = p(\phi) \prod_{i=1}^{N} p(\theta_i|\phi)$. This has several crucial benefits: Firstly, given this hierarchy, variational inference in our framework decomposes into separable optimisation problems over $\theta_i$s and $\phi$, enabling a practical Bayesian learning algorithm

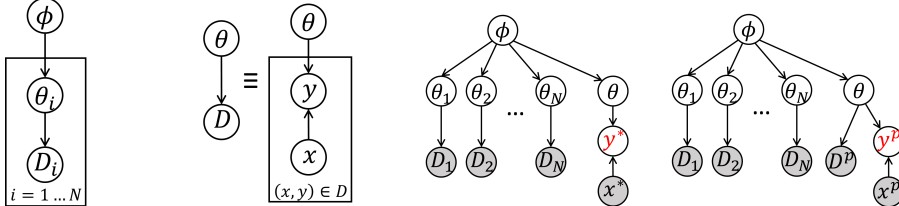

(a) Overall model    (b) Individual client    (c) Global prediction    (d) Personalisation

Figure 1: Graphical models. (a) Plate view of iid clients. (b) Individual client data with input images $x$ given and only $p(y|x)$ modeled. (c) & (d): Global prediction and personalisation as probabilistic inference problems (shaded nodes = *evidences*, red colored nodes = *targets* to infer, $x^*$ = test input in global prediction, $D^p$ = training data for personalisation and $x^p$ = test input).

to be derived that is fully compatible with FL constraints, without resorting to ad-hoc treatments or strong assumptions. Secondly, this framework can be instantiated with different assumptions on $p(\theta_i|\phi)$ to deal elegantly and robustly with different kinds of statistical heterogeneity, as well as for principled and effective model personalisation.

Our resulting algorithm, termed Federated Hierarchical Bayes (FedHB) is empirically effective, as we demonstrate in a wide range of experiments on established benchmarks. More importantly, it benefits from rigorous theoretical support. In particular, we provide convergence guarantees showing that FedHB has the same $O(1/\sqrt{T})$ convergence rate as centralised SGD algorithms, which are not provided by related prior art Zhang et al. (2022); Chen & Chao (2021). We also provide a generalisation bound showing that FedHB is asymptotically optimal, which has not been shown by prior work such as Al-Shedivat et al. (2021). Furthermore we show that FedHB subsumes classic methods FedAvg McMahan et al. (2017) and FedProx Li et al. (2018) as special cases, and ultimately provides additional justification and explanation for these seminal methods.

## 2 BAYESIAN FL: GENERAL FRAMEWORK

We introduce two types of latent random variables, $\phi$ and $\{\theta_i\}_{i=1}^N$. Each $\theta_i$ is deployed as the network weights for client $i$'s backbone. The variable $\phi$ can be viewed as a globally shared variable that is responsible for linking the individual client parameters $\theta_i$. We assume conditionally independent and identical priors, $p(\theta_{1:N}|\phi) = \prod_{i=1}^N p(\theta_i|\phi)$. Thus the prior for the latent variables $(\phi, \{\theta_i\}_{i=1}^N)$ is formed in a hierarchical manner as (1). The local data for client $i$, denoted by $D_i$, is generated[1] by $\theta_i$,

$$\text{(Prior)} \ \ p(\phi, \theta_{1:N}) = p(\phi)\prod_{i=1}^N p(\theta_i|\phi) \quad \text{(Likelihood)} \ \ p(D_i|\theta_i) = \prod_{(x,y)\in D_i} p(y|x,\theta_i), \quad (1)$$

where $p(y|x,\theta_i)$ is a conventional neural network model (e.g., softmax link for classification tasks). See the graphical model in Fig. 1(a) where the iid clients are governed by a single random variable $\phi$.

Given the data $D_1, \ldots, D_N$, we infer the posterior, $p(\phi, \theta_{1:N}|D_{1:N}) \propto p(\phi)\prod_{i=1}^N p(\theta_i|\phi)p(D_i|\theta_i)$, which is intractable in general, and we adopt the variational inference to approximate it:

$$q(\phi, \theta_{1:N}; L) := q(\phi; L_0)\prod_{i=1}^N q_i(\theta_i; L_i), \quad (2)$$

where the variational parameters $L$ consists of $L_0$ (parameters for $q(\phi)$) and $\{L_i\}_{i=1}^N$'s (parameters for $q_i(\theta_i)$'s from individual clients). Note that although $\theta_i$'s are independent across clients under (2), they are differently modeled (emphasised by the subscript $i$ in notation $q_i$), reflecting different posterior beliefs originating from heterogeneity of local data $D_i$'s.

### 2.1 FROM VARIATIONAL INFERENCE TO FEDERATED LEARNING ALGORITHM

Using the standard variational inference techniques (Blei et al., 2017; Kingma & Welling, 2014), we can derive the ELBO objective function (details in Appendix A). We denote the *negative* ELBO by $\mathcal{L}$

---

[1]Note that we do not deal with generative modeling of input images $x$. Inputs $x$ are always given, and only conditionals $p(y|x)$ are modeled. See Fig. 1(b) for the in-depth graphical model diagram.

(to be minimised over $L$) as follows:

$$\mathcal{L}(L) := \sum_{i=1}^{N} \Big( \mathbb{E}_{q_i(\theta_i)}[-\log p(D_i|\theta_i)] + \mathbb{E}_{q(\phi)}\big[\mathrm{KL}(q_i(\theta_i)||p(\theta_i|\phi))\big] \Big) + \mathrm{KL}(q(\phi)||p(\phi)), \quad (3)$$

where we drop the dependency on $L$ in notation for simplicity. Instead of optimizing (3) over the parameters $L$ jointly as usual practice, we consider block-wise optimisation, also known as *block-coordinate optimisation* (Wright, 2015), specifically alternating two steps: (i) updating/optimizing all $L_i$'s $i = 1, \ldots, N$ while fixing $L_0$, and (ii) updating $L_0$ with all $L_i$'s fixed. That is,

- Optimisation over $L_1, \ldots, L_N$ ($L_0$ fixed).

$$\min_{\{L_i\}_{i=1}^N} \sum_{i=1}^{N} \Big( \mathbb{E}_{q_i(\theta_i)}[-\log p(D_i|\theta_i)] + \mathbb{E}_{q(\phi)}\big[\mathrm{KL}(q_i(\theta_i)||p(\theta_i|\phi))\big] \Big). \quad (4)$$

As (4) is completely separable over $i$, and we can optimise each summand independently as:

$$\min_{L_i} \mathcal{L}_i(L_i) := \mathbb{E}_{q_i(\theta_i;L_i)}[-\log p(D_i|\theta_i)] + \mathbb{E}_{q(\phi;L_0)}\big[\mathrm{KL}(q_i(\theta_i;L_i)||p(\theta_i|\phi))\big]. \quad (5)$$

So (5) constitutes local update/optimisation for client $i$. Note that each client $i$ needs to access its private data $D_i$ only without data from others, thus fully compatible with FL.

- Optimisation over $L_0$ ($L_1, \ldots, L_N$ fixed).

$$\min_{L_0} \mathcal{L}_0(L_0) := \mathrm{KL}(q(\phi;L_0)||p(\phi)) - \sum_{i=1}^{N} \mathbb{E}_{q(\phi;L_0)q_i(\theta_i;L_i)}[\log p(\theta_i|\phi)]. \quad (6)$$

This constitutes server update criteria while the latest $q_i(\theta_i; L_i)$'s from local clients being fixed. Remarkably, the server needs not access any local data at all, suitable for FL. This nice property originates from the independence assumption in our approximate posterior (2).

**Interpretation.** First, server's loss function (6) tells us that the server needs to update $q(\phi; L_0)$ in such a way that (i) it puts mass on those $\phi$ that have high compatibility scores $\log p(\theta_i|\phi)$ with the current local models $\theta_i \sim q_i(\theta_i)$, thus aiming to be aligned with local models, and (ii) it does not deviate from the prior $p(\phi)$. Clients' loss function (5) indicates that each client $i$ needs to minimise the class prediction error on its own data $D_i$ (first term), and at the same time, to stay close to the current global standard $\phi \sim q(\phi)$ by reducing the KL divergence from $p(\theta_i|\phi)$ (second term).

## 2.2 FORMALISATION OF GLOBAL PREDICTION AND PERSONALISATION TASKS

Two important tasks in FL are: *global prediction* and *personalisation*. The former evaluates the trained model on novel test data sampled from a distribution possibly different from training data. Personalisation is the task of adapting the trained model on a new dataset called personalised data. In our Bayesian model, these two tasks can be formally defined as Bayesian inference problems.

**Global prediction.** The task is to predict the class label of a novel test input $x^*$ which may or may not come from the same distributions as the training data $D_1, \ldots D_N$. Under our Bayesian model, it can be turned into a probabilistic inference problem $p(y^*|x^*, D_{1:N})$. Let $\theta$ be the local model that generates the output $y^*$ given $x^*$. Exploiting conditional independence from Fig. 1(c),

$$p(y^*|x^*, D_{1:N}) = \iint p(y^*|x^*, \theta)\, p(\theta|\phi)\, p(\phi|D_{1:N})\, d\theta d\phi \quad (7)$$

$$\approx \iint p(y^*|x^*, \theta)\, p(\theta|\phi)\, q(\phi)\, d\theta d\phi \ = \ \int p(y^*|x^*, \theta) \left( \int p(\theta|\phi)\, q(\phi) d\phi \right) d\theta, \quad (8)$$

where in (8) we use $p(\phi|D_{1:N}) \approx q(\phi)$. The inner integral (in parentheses) in (8) either admits a closed form (Sec. 3.1) or can be approximated (e.g., Monte-Carlo estimation).

**Personalisation.** It formally refers to the task of learning a prediction model $\hat{p}(y|x)$ given an unseen (personal) training dataset $D^p$ that comes from some unknown distribution $p^p(x, y)$, so that the personalised model $\hat{p}$ performs well on novel (in-distribution) test points $(x^p, y^p) \sim p^p(x, y)$. Evidently we need to exploit (and benefit from) the trained model from the FL training stage. To this end many existing approaches simply resort to *finetuning*, that is, training on $D^p$ warm-starting with the FL-trained model. However, a potential issue is the lack of a solid principle on how to balance the initial FL-trained model and personal data fitting to avoid underfitting and overfitting. In our Bayesian framework, the personalisation can be seen as another posterior inference problem with *additional evidence* of the personal training data $D^p$. Prediction on a test point $x^p$ amounts to inferring:

$$p(y^p|x^p, D^p, D_{1:N}) = \int p(y^p|x^p, \theta)\, p(\theta|D^p, D_{1:N})\, d\theta. \quad (9)$$

So, it boils down to the task of posterior inference $p(\theta|D^p, D_{1:N})$ given both the personal data $D^p$ and the FL training data $D_{1:N}$. Under our hierarchical model, by exploiting conditional independence from graphical model (Fig. 1(d)), we can link the posterior to our FL-trained $q(\phi)$ as follows:

$$p(\theta|D^p, D_{1:N}) \approx \int p(\theta|D^p, \phi)\, p(\phi|D_{1:N})\, d\phi \;\approx\; \int p(\theta|D^p, \phi)\, q(\phi)\, d\phi \;\approx\; p(\theta|D^p, \phi^*), \quad (10)$$

where we disregard the impact of $D^p$ on the higher-level $\phi$ given the joint evidence, $p(\phi|D^p, D_{1:N}) \approx p(\phi|D_{1:N})$ due to the dominance of $D_{1:N}$ compared to smaller $D^p$. The last part of (10) makes approximation using the mode $\phi^*$ of $q(\phi)$, which is reasonable for our two modeling choices for $q(\phi)$ to be discussed in Sec. 3.1 and Sec. 3.2. Since dealing with $p(\theta|D^p, \phi^*)$ involves difficult marginalisation $p(D^p|\phi^*) = \int p(D^p|\theta)p(\theta|\phi^*)d\theta$, we adopt variational inference, introducing a tractable variational distribution $v(\theta) \approx p(\theta|D^p, \phi^*)$. Following the usual variational inference derivations, we have the negative ELBO objective (for personalisation):

$$\min_v \; \mathbb{E}_{v(\theta)}[-\log p(D^p|\theta)] + \mathrm{KL}(v(\theta)||p(\theta|\phi^*)). \quad (11)$$

Once we have the optimised model $v$, our predictive distribution becomes:

$$p(y^p|x^p, D^p, D_{1:N}) \approx \frac{1}{S}\sum_{s=1}^{S} p(y^p|x^p, \theta^{(s)}), \quad \text{where } \theta^{(s)} \sim v(\theta), \quad (12)$$

which simply requires feed-forwarding test input $x^p$ through the sampled networks $\theta^{(s)}$ and averaging.

Thus far, we have discussed a general framework, deriving how the variational inference for our Bayesian model fits gracefully in the FL problem. In the next section, we define specific density families for the prior ($p(\phi)$, $p(\theta_i|\phi)$) and posterior ($q(\phi)$, $q_i(\theta_i)$) as our proposed concrete models.

## 3 Bayesian FL: Two Concrete Models

We propose two different model choices that we find the most interesting: **Normal-Inverse-Wishart** (Sec. 3.1) and **Mixture** (Sec. 3.2). To avoid distraction, we make this section concise putting only the final results and discussions, and leaving all mathematical details in Appendix B and C.

### 3.1 Normal-Inverse-Wishart (NIW) Model

We define the prior as a conjugate form of Gaussian and Normal-Inverse-Wishart. With $\phi = (\mu, \Sigma)$,

$$p(\phi) = \mathcal{NIW}(\mu, \Sigma; \Lambda) = \mathcal{N}(\mu; \mu_0, \lambda_0^{-1}\Sigma) \cdot \mathcal{IW}(\Sigma; \Sigma_0, \nu_0), \quad (13)$$

$$p(\theta_i|\phi) = \mathcal{N}(\theta_i; \mu, \Sigma), \quad i = 1, \ldots, N, \quad (14)$$

where $\Lambda = \{\mu_0, \Sigma_0, \lambda_0, \nu_0\}$ is the parameters of the NIW. Although $\Lambda$ can be learned via data marginal likelihood maximisation (e.g., empirical Bayes), but for simplicity we leave it fixed as[2]: $\mu_0 = 0$, $\Sigma_0 = I$, $\lambda_0 = 1$, and $\nu_0 = d + 2$ where $d$ is the number of parameters in $\theta_i$ or $\mu$. Next, our choice of the variational density family for $q(\phi)$ is the NIW, not just because it is the most popular parametric family for a pair of mean vector and covariance matrix $\phi = (\mu, \Sigma)$, but it can also admit closed-form expressions in the ELBO function due to the conjugacy as we derive in Sec. B.1.

$$q(\phi) := \mathcal{NIW}(\phi; \{m_0, V_0, l_0, n_0\}) = \mathcal{N}(\mu; m_0, l_0^{-1}\Sigma) \cdot \mathcal{IW}(\Sigma; V_0, n_0). \quad (15)$$

Although the scalar parameters $l_0$, $n_0$ can be optimised together with $m_0$, $V_0$, their impact is less influential and we find that they make the ELBO optimisation a little bit cumbersome. So we fix $l_0$, $n_0$ with some near-optimal values by exploiting the conjugacy of the NIW under Gaussian likelihood (details in Appendix B), and regard $m_0, V_0$ as variational parameters, $L_0 = \{m_0, V_0\}$. We restrict $V_0$ to be diagonal for computational tractability. The density family for $q_i(\theta_i)$'s can be a Gaussian, but we find that it is computationally more attractive and numerically more stable to adopt the mixture of two spiky Gaussians that leads to the MC-Dropout (Gal & Ghahramani, 2016). That is,

$$q_i(\theta_i) = \prod_l \big(p \cdot \mathcal{N}(\theta_i[l]; m_i[l], \epsilon^2 I) + (1-p) \cdot \mathcal{N}(\theta_i[l]; 0, \epsilon^2 I)\big), \quad (16)$$

where (i) $m_i$ is the only variational parameters ($L_i = \{m_i\}$), (ii) $\cdot[l]$ indicates a column/layer in neural network parameters where $l$ goes over layers and columns of weight matrices, (iii) $p$ is the (user-specified) hyperparameter where $1 - p$ corresponds to the dropout probability, and (iv) $\epsilon$ is small constant (e.g., $10^{-4}$) that makes two Gaussians spiky, close to the delta functions.

**Client update.** We apply the general client update optimisation (5) to the NIW model. Following the approximation of (Gal & Ghahramani, 2016) for the KL divergence between a mixture of Gaussians

---

[2]This choice ensures that the mean of $\Sigma$ equals $I$, and $\mu$ is distributed as 0-mean Gaussian with covariance $\Sigma$.

(16) and a Gaussian (14), we have the client local optimisation (details in Appendix B):

$$\min_{m_i} \mathcal{L}_i(m_i) := -\log p(D_i|\tilde{m}_i) + \frac{p}{2}(n_0 + d + 1)(m_i - m_0)^\top V_0^{-1}(m_i - m_0), \qquad (17)$$

where $\tilde{m}_i$ is the dropout version of $m_i$, i.e., a reparametrised sample from (16). Note that $m_0$ and $V_0$ are fixed during the optimisation. Interestingly (17) generalises Fed-Avg (McMahan et al., 2017) and Fed-Prox (Li et al., 2018): With $p = 1$ (i.e., no dropout) and setting $V_0 = \alpha I$, (17) reduces to the client update formula for Fed-Prox where constant $\alpha$ controls the impact of the proximal term.

**Server update.** The general server optimisation (6) admits the closed-form solution (Appendix B):

$$m_0^* = \frac{p}{N+1}\sum_{i=1}^N m_i, \;\; V_0^* = \frac{n_0}{N+d+2}\left((1 + N\epsilon^2)I + m_0^*(m_0^*)^\top + \sum_{i=1}^N \rho(m_0^*, m_i, p)\right), \;\; (18)$$

where $\rho(m_0, m_i, p) = pm_i m_i^\top - pm_0 m_i^\top - pm_i m_0^\top + m_0 m_0^\top$. Note that $m_i$'s are fixed from clients' latest variational parameters. It is interesting to see that $m_0^*$ in (18) generalises the well-known aggregation step of averaging local models in Fed-Avg (McMahan et al., 2017) and related methods: when $p = 1$ (no dropout), it almost[3] equals client model averaging. Also, since $\rho(m_0^*, m_i, p = 1) = (m_i - m_0^*)(m_i - m_0^*)^\top$ when $p = 1$, $V_0^*$ essentially estimates the sample scatter matrix with $(N + 1)$ samples, namely clients' $m_i$'s and server's prior $\mu_0 = 0$, measuring how much they deviate from the center $m_0^*$. The dropout is known to help regularise the model and lead to better generalisation (Gal & Ghahramani, 2016), and with $p < 1$ our (18) forms a principled optimal solution.

**Global prediction.** The inner integral of (8) becomes the multivariate Student-$t$ distribution. Then the predictive distribution for a new test input $x^*$ can be estimated as[4]:

$$p(y^*|x^*, D_{1:N}) \approx \frac{1}{S}\sum_{s=1}^S p(y^*|x^*, \theta^{(s)}), \;\; \text{where} \;\; \theta^{(s)} \sim t_{n_0-d+1}\left(\theta; m_0, \frac{(l_0+1)V_0}{l_0(n_0-d+1)}\right), \;\; (19)$$

where $t_\nu(a, B)$ is the multivariate Student-$t$ with location $a$, scale matrix $B$, and d.o.f. $\nu$.

**Personalisation.** With the given personalisation training data $D^p$, we follow the general framework in (11) to find $v(\theta) \approx p(\theta|D^p, \phi^*)$ in a variational way, where $\phi^*$ obtained from (34). We adopt the same spiky mixture form (16) for $v(\theta)$, which leads to the learning objective similar to (17).

## 3.2 MIXTURE MODEL

Our motivation for mixture is to make the prior $p(\theta, \phi)$ more flexible by having multiple different prototypes, diverse enough to cover the heterogeneity in data distributions across clients. We consider:

$$p(\phi) = \prod_{j=1}^K \mathcal{N}(\mu_j; 0, I), \;\; p(\theta_i|\phi) = \sum_{j=1}^K \frac{1}{K}\mathcal{N}(\theta_i; \mu_j, \sigma^2 I), \qquad (20)$$

where $\phi = \{\mu_1, \ldots, \mu_K\}$ contains $K$ networks (prototypes) that can broadly cover the clients data distributions, and $\sigma$ is the hyperparameter that captures perturbation scale, chosen by users or learned from data. Note that we put equal mixing proportions $1/K$ due to the symmetry, a priori. That is, each client can take any of $\mu_j$'s equally likely a priori. For the variational densities, we define:

$$q_i(\theta_i) = \mathcal{N}(\theta_i; m_i, \epsilon^2 I), \;\; q(\phi) = \prod_{j=1}^K \mathcal{N}(\mu_j; r_j, \epsilon^2 I), \qquad (21)$$

where $\{r_j\}_{j=1}^K$ ($L_0$) and $m_i$ ($L_i$) are the variational parameters, and $\epsilon$ is small constant (e.g., $10^{-4}$).

**Client update.** For our model choice, the general client update (5) reduces to (details in Appendix C):

$$\min_{m_i} \mathbb{E}_{q_i(\theta_i)}[-\log p(D_i|\theta_i)] - \log \sum_{j=1}^K \exp\left(-\frac{||m_i - r_j||^2}{2\sigma^2}\right). \qquad (22)$$

It is interesting to see that (22) can be seen as generalisation of Fed-Prox (Li et al., 2018), where the proximal regularisation term in Fed-Prox is extended to *multiple* global models $r_j$'s, penalizing the local model ($m_i$) straying away from these prototypes. And if we use a single prototype ($K = 1$), the optimisation (22) exactly reduces to the local update objective of Fed-Prox. Since `log-sum-exp` is approximately equal to `max`, the regularisation term in (22) effectively focuses on the closest global prototype $r_j$ from the current local model $m_i$, which is intuitively well aligned with our motivation.

---

[3]Only the constant 1 added to the denominator, which comes from the prior and has the regularising effect.
[4]In practice we use a single sample ($S = 1$) for computational efficiency.

**Server update.** The general form (6) can be approximately turned into (Appendix C for derivations):

$$\min_{\{r_j\}_{j=1}^K} \frac{1}{2} \sum_{j=1}^K ||r_j||^2 - \sum_{i=1}^N \log \sum_{j=1}^K \exp\left( - \frac{||m_i - r_j||^2}{2\sigma^2} \right). \tag{23}$$

Interestingly, (23) generalises the well-known aggregation step of averaging local models in Fed-Avg and related methods: Especially when $K = 1$, (23) reduces to quadratic optimisation, admitting the optimal solution $r_1^* = \frac{1}{N+\sigma^2} \sum_{i=1}^N m_i$. The extra term $\sigma^2$ can be explained by incorporating an extra *zero* local model originating from the prior (interpreted as a *neutral* model) with the discounted weight $\sigma^2$ rather than 1. Although (23) for $K > 1$ can be solved by standard gradient descent, we apply the Expectation-Maximisation (EM) algorithm[5] (Dempster et al., 1977) instead:

$$\text{(E-step)} \ \ c(j|i) = \frac{e^{-||m_i - r_j||^2/(2\sigma^2)}}{\sum_{j=1}^K e^{-||m_i - r_j||^2/(2\sigma^2)}}, \quad \text{(M-step)} \ \ r_j^* = \frac{\frac{1}{N} \sum_{i=1}^N c(j|i) \cdot m_i}{\frac{\sigma^2}{N} + \frac{1}{N} \sum_{i=1}^N c(j|i)}. \tag{24}$$

The M-step (server update) has intuitive meaning that the new prototype $r_j$ becomes the *weighted* average of the local models $m_i$'s where the weights $c(j|i)$ are determined by the proximity between $m_i$ and $r_j$ (i.e., those $m_i$'s that are closer to $r_j$ have more contribution, and vice versa). This can be seen as an extension of the aggregation step in Fed-Avg to the multiple prototype case.

**Global prediction.** We slightly modify our general approach to make individual client data dominantly explained by the most relevant model $r_j$, by introducing a gating function from the mixture of experts (Jacobs et al., 1991; Jordan & Jacobs, 1994). See Appendix C for details.

**Personalisation.** With $v(\theta)$ of the same form as $q_i(\theta_i)$, the VI learning becomes similar to (22).

## 4 THEORETICAL ANALYSIS

We provide two theoretical results for our Bayesian FL algorithm: (**Convergence analysis**) As a special block-coordinate optimisation algorithm, we show that it converges to an (local) optimum of the training objective (3); (**Generalisation error bound**) We theoretically show how well this optimal model trained on empirical data performs on unseen test data points. Due to space limit, full details and proofs are described in Appendix D,E, and we only state the theorems and remarks here.

**Theorem 4.1** (Convergence analysis). *We denote the objective function in (3) by $f(x)$ where $x = [x_0, x_1, \ldots, x_N]$ corresponding to the variational parameters $x_0 := L_0$, $x_1 := L_1$, ..., $x_N := L_N$. Let $\eta_t = \overline{L} + \sqrt{t}$ for some constant $\overline{L}$, and $\overline{x}^T = \frac{1}{T} \sum_{t=1}^T x^t$, where $t$ is the batch iteration counter, $x^t$ is the iterate at $t$ by following our FL algorithm, and $N_f$ ($\leq N$) is the number of participating clients at each round. With Assumptions 1–3 in Appendix D, the following holds for any $T$:*

$$\mathbb{E}[f(\overline{x}^T)] - f(x^*) \leq \frac{N + N_f}{N_f} \cdot \frac{\frac{\sqrt{T}+\overline{L}}{2}D^2 + R_f^2\sqrt{T}}{T} = O\left(\frac{1}{\sqrt{T}}\right), \tag{25}$$

*where $x^*$ is the (local) optimum, $D$, and $R_f$ are some constants, and the expectation is taken over randomness in minibatches and selection of participating clients.*

*Remark.* It says that $\overline{x}^t$ converges to the optimal point $x^*$ in expectation at the rate of $O(1/\sqrt{t})$. This rate asymptotically equals that of the conventional (non-block-coordinate, holistic) SGD algorithm.

**Theorem 4.2** (Generalisation error bound). *Assume that the variational density family for $q_i(\theta_i)$ is rich enough to subsume Gaussian. Let $d^2(P_{\theta_i}, P^i)$ be the expected squared Hellinger distance between the true class distribution $P^i(y|x)$ and model's $P_{\theta_i}(y|x)$ for client $i$'s data. The optimal solution $(\{q_i^*(\theta_i)\}_{i=1}^N, q^*(\phi))$ of the optimisation problem (3) satisfies:*

$$\frac{1}{N} \sum_{i=1}^N \mathbb{E}_{q_i^*(\theta_i)}[d^2(P_{\theta_i}, P^i)] \ \leq \ O\left(\frac{1}{n}\right) + C \cdot \epsilon_n^2 + C'\left(r_n + \frac{1}{N} \sum_{i=1}^N \lambda_i^*\right), \tag{26}$$

*with high probability, where $C, C' > 0$ are constant, $\lambda_i^* = \min_{\theta \in \Theta} ||f_\theta - f^i||_\infty^2$ is the best error within our backbone network family $\Theta$, and $r_n, \epsilon_n \to 0$ as the training data size $n \to \infty$.*

*Remark.* It implies that the optimal solution of (3) (attainable by our block-coordinate FL algorithm) is *asymptotically optimal*, since the RHS of (26) converges to 0 as the training data size $n \to \infty$.

---

[5]Instead of performing several EM steps until convergence, in practice we find only one EM step is sufficient.

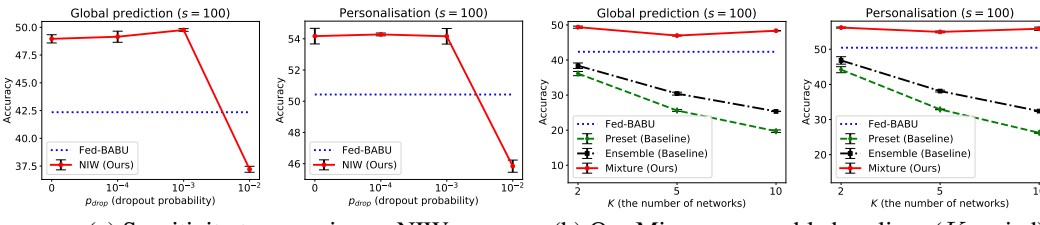

(a) Sensitivity to $p_{drop}$ in our NIW  (b) Our Mix. vs. ensemble baselines ($K$ varied)

Figure 2: Hyperparameter sensitivity analysis and comparison with simple ensemble baselines.

## 5 RELATED WORK

**General FL approaches.** FedAvg (McMahan et al., 2017) is the pioneering seminal work on FL, which proposed fairly intuitive local training and global aggregation strategies with minimal training and communication complexity. A potential issue of divergence of global and local models due to the separated steps of local training and aggregation was addressed by model regularisation in the follow-up works (Li et al., 2018; Acar et al., 2021), which is shown to help the global model converge more reliably. Recent approaches aimed to incorporate benefits from existing machine learning approaches including domain adaptation/generalisation, clustering, multi-task learning, transfer learning, and meta-learning. Due to the lack of space, we leave related references in Appendix I.

**Comparison to existing Bayesian FL approaches.** Some recent studies tried to address the FL problem using Bayesian methods. As we mentioned earlier, the key difference is that these methods do not introduce Bayesian *hierarchy*, and ultimately treat network weights $\theta$ as a random variable *shared* across all clients, while our approach assigns individual $\theta_i$ to each client $i$ governed by a common prior $p(\theta_i|\phi)$. The non-hierarchical approaches must all resort to ad hoc heuristics or strong assumptions in parts of their algorithm. Due to the lack of space, we leave related references and discussions in Appendix I.

## 6 EVALUATION

In this section we evaluate the proposed hierarchical Bayesian models on two benchmark datasets: the popular **CIFAR-100** and the challenging corrupted version (**CIFAR-C-100**) that renders the client data more heterogeneous both in input images and class distributions.

**FL settings for CIFAR-100.** We follow the settings similar to those used in (Oh et al., 2022); in particular the client data distributions are heterogeneous non-iid, formed by the sharding-based class sampling (McMahan et al., 2017). More specifically, we partition data instances in each class into non-overlapping equal-sized shards, and assign $s$ randomly sampled shards (over all classes) to each of $N$ clients. Thus the number of shards per user $s$ can control the degree of data heterogeneity: small $s$ leads to more heterogeneity, and vice versa. The number of clients $N = 100$ (each having 500 training, 100 test samples), and we denote by $f$ the fraction of participating clients. So, $N_f = \lfloor N \cdot f \rfloor$ clients are randomly sampled at each round to participate in training. Smaller $f$ makes the FL more challenging, and we test two settings: $f = 1.0$ and $0.1$. Lastly, the number of epochs for client local update at each round is denoted by $\tau$ where we test $\tau = 1$ and $10$, and the number of total rounds is determined by $\tau$ as $\lfloor 320/\tau \rfloor$ for fairness. Note that smaller $\tau$ incurs more communication cost but often leads to higher accuracy.

**FL settings for CIFAR-100-Corrupted (CIFAR-C-100).** The dataset (Hendrycks & Dietterich, 2019) makes CIFAR-100's test split ($10K$ images) corrupted by 19 different noise processes (e.g., Gaussian, motion blur, JPEG). For each corruption type, there are 5 corruption levels, and we use the severest one. Randomly chosen 10 corruption types are used for training (fixed) and the rest 9 types for personalisation. We divide $N = 100$ clients into 10 groups, each group assigned one of the 10 training corruption types exclusively (denoted by $D^c$ the corrupted data for the group $c = 1, \ldots, 10$). Each $D^c$ is partitioned into $90\%/10\%$ training/test splits, and clients in each group ($\lfloor N/10 \rfloor$ clients) gets non-iid train/test subsets from $D^c$'s train/test splits by following the sharding strategy with $s = 100$ or $50$. This way, the clients in different groups have considerable distribution shift in input images, while there also exists heterogeneity in class distributions even within the same groups. For the FL-trained models, we evaluate *global prediction* on two datasets: clients' test splits from the 10 training corruption types and the original (uncorrupted) CIFAR's training split ($50K$ images). For *personalisation*, we partition the clients into 9 groups, and assign one of the 9 corruption types to each group exclusively. Within each group we form non-iid sharding-based subsets similarly, and again we split the data into the $90\%$ training/finetuning split and $10\%$ test. Note that this personalisation

Table 1: (CIFAR-100) Global prediction and personalisation accuracy.

(a) Global prediction performance (initial accuracy)

| FL settings | | | Our Methods | | Fed-BABU | Fed-Avg | Fed-Prox | pFedBayes |
|---|---|---|---|---|---|---|---|---|
| $s$ | $f$ | $\tau$ | NIW | Mix. ($K=2$) | | | | |
| 100 | 0.1 | 1 | $\mathbf{49.76^{\pm 0.12}}$ | $49.37^{\pm 0.30}$ | $42.35^{\pm 0.42}$ | $40.87^{\pm 0.62}$ | $41.49^{\pm 0.75}$ | $37.23^{\pm 0.88}$ |
| | | 10 | $\mathbf{29.02^{\pm 0.33}}$ | $29.02^{\pm 0.29}$ | $27.93^{\pm 0.28}$ | $28.26^{\pm 0.19}$ | $27.11^{\pm 0.11}$ | $28.21^{\pm 1.42}$ |
| | 1.0 | 1 | $\mathbf{57.80^{\pm 0.10}}$ | $52.94^{\pm 0.36}$ | $48.17^{\pm 0.56}$ | $47.44^{\pm 0.20}$ | $47.66^{\pm 1.49}$ | $44.89^{\pm 0.32}$ |
| | | 10 | $29.53^{\pm 0.42}$ | $\mathbf{30.55^{\pm 0.15}}$ | $28.67^{\pm 0.51}$ | $28.79^{\pm 0.68}$ | $27.43^{\pm 0.38}$ | $28.25^{\pm 0.81}$ |
| 10 | 0.1 | 1 | $37.54^{\pm 0.25}$ | $\mathbf{38.07^{\pm 0.40}}$ | $35.04^{\pm 0.56}$ | $27.48^{\pm 0.86}$ | $34.73^{\pm 0.21}$ | $31.49^{\pm 0.18}$ |
| | | 10 | $\mathbf{18.99^{\pm 0.03}}$ | $18.95^{\pm 0.13}$ | $18.54^{\pm 0.37}$ | $14.69^{\pm 0.40}$ | $16.84^{\pm 0.48}$ | $17.93^{\pm 0.68}$ |
| | 1.0 | 1 | $\mathbf{50.40^{\pm 0.11}}$ | $49.52^{\pm 0.88}$ | $45.41^{\pm 0.11}$ | $37.10^{\pm 0.44}$ | $44.33^{\pm 0.31}$ | $39.95^{\pm 0.89}$ |
| | | 10 | $22.87^{\pm 0.41}$ | $\mathbf{23.59^{\pm 0.47}}$ | $21.92^{\pm 0.66}$ | $17.38^{\pm 0.32}$ | $19.54^{\pm 0.38}$ | $21.85^{\pm 0.50}$ |

(b) Personalisation performance

| FL settings | | | Our Methods | | Fed-BABU | Fed-Avg | Fed-Prox | pFedBayes |
|---|---|---|---|---|---|---|---|---|
| $s$ | $f$ | $\tau$ | NIW | Mix. ($K=2$) | | | | |
| 100 | 0.1 | 1 | $54.16^{\pm 0.50}$ | $\mathbf{56.17^{\pm 0.16}}$ | $50.43^{\pm 0.93}$ | $46.43^{\pm 0.82}$ | $49.91^{\pm 0.78}$ | $45.83^{\pm 1.12}$ |
| | | 10 | $\mathbf{36.68^{\pm 0.37}}$ | $36.32^{\pm 0.27}$ | $35.45^{\pm 0.34}$ | $33.57^{\pm 0.06}$ | $33.92^{\pm 0.22}$ | $35.74^{\pm 1.36}$ |
| | 1.0 | 1 | $\mathbf{60.36^{\pm 0.89}}$ | $58.82^{\pm 0.37}$ | $55.87^{\pm 0.91}$ | $53.15^{\pm 0.25}$ | $55.50^{\pm 0.90}$ | $53.00^{\pm 0.48}$ |
| | | 10 | $35.92^{\pm 0.17}$ | $\mathbf{36.22^{\pm 0.17}}$ | $35.58^{\pm 0.24}$ | $33.82^{\pm 1.04}$ | $33.70^{\pm 0.42}$ | $35.57^{\pm 1.02}$ |
| 10 | 0.1 | 1 | $79.41^{\pm 0.24}$ | $\mathbf{79.70^{\pm 0.19}}$ | $75.44^{\pm 0.36}$ | $70.36^{\pm 1.02}$ | $75.06^{\pm 0.67}$ | $73.93^{\pm 0.14}$ |
| | | 10 | $67.35^{\pm 1.02}$ | $\mathbf{67.57^{\pm 0.62}}$ | $66.24^{\pm 0.53}$ | $61.39^{\pm 0.27}$ | $64.86^{\pm 0.73}$ | $65.82^{\pm 0.33}$ |
| | 1.0 | 1 | $\mathbf{82.71^{\pm 0.37}}$ | $81.03^{\pm 0.35}$ | $78.92^{\pm 0.23}$ | $76.98^{\pm 0.66}$ | $78.56^{\pm 0.55}$ | $78.08^{\pm 0.28}$ |
| | | 10 | $\mathbf{67.78^{\pm 1.02}}$ | $66.74^{\pm 0.27}$ | $66.25^{\pm 0.46}$ | $63.81^{\pm 0.40}$ | $63.81^{\pm 0.51}$ | $66.15^{\pm 1.29}$ |

setting is more challenging compared to CIFAR-100 since the data for personalisation are utterly unseen during the FL training stage. We test $\tau = 1$ and 4 scenarios.

**Experimental settings.** Our implementation is based on Oh et al. (2022) where we use the MobileNet (Howard et al., 2017) as a backbone, and follow the body-update strategy: the classification head (the last layer) is randomly initialised and fixed during training, with only the network body updated (and both the body and head updated at the personalisation stage). We report experimental results all based on this body-update strategy since we observe that it considerably outperforms the full update for our models and all other competing methods. The hyperparameters in our models are: (**NIW**) $\epsilon = 10^{-4}$ and $p = 1 - 0.001$ (See below for ablation study of other values); (**Mixture**) $\sigma^2 = 0.1$, $\epsilon = 10^{-4}$, mixture order $K = 2$ (See Appendix F.1 for results with other values), and the gating network has the same architecture as the main backbone, but with output cardinality changed to $K$. The other hyperparameters including batch size (50), learning rate (0.1 initially and decayed by 0.1) and the number of epochs in personalisation (5), are the same as those in (Oh et al., 2022).

**Main results.** In Table 1 (CIFAR-100) and Table 2 (CIFAR-C-100), we compare our methods (NIW and Mixture with $K = 2$) against the popular FL methods, Fed-BABU (Oh et al., 2022), Fed-Avg (McMahan et al., 2017), Fed-Prox (Li et al., 2018), and the recent pFedBayes (Zhang et al., 2022). The latter is especially interesting to contrast with as it is based on variational inference, most closely related to ours. We run the competing methods (implementation based on their public codes) with default hyperparameters (e.g., $\mu = 0.01$ for FedProx) and report the results. First of all, our two models (NIW and Mix.) consistently perform the best (by large margins most of the time) in terms of both global prediction and personalisation for nearly all FL settings on the two datasets. This is attributed to the principled Bayesian modeling of the underlying FL data generative process in our approaches that can be seen as rigorous generalisation and extension of the existing intuitive algorithms such as Fed-Avg and Fed-Prox. In particular, the superiority of our methods to the other Bayesian approach pFedBayes verifies the effectiveness of modeling client-wise latent variables $\theta_i$ against the commonly used shared $\theta$ modeling, especially for the scenarios of significant client data heterogeneity (e.g., CIFAR-C-100 personalisation on data with unseen corruption types).

**(Ablation) Hyperparameter sensitivity.** We test sensitivity to some key hyperparameters in our models. For NIW, we have $p = 1 - p_{drop}$, the MC-dropout probability, where we used $p_{drop} = 0.001$

Table 2: (CIFAR-C-100) Global prediction and personalisation accuracy.

(a) Global prediction (initial accuracy) on test splits for the 10 training corruption types

| FL settings | | | Our Methods | | Fed-BABU | Fed-Avg | Fed-Prox | pFedBayes |
|---|---|---|---|---|---|---|---|---|
| $s$ | $f$ | $\tau$ | NIW | Mix. ($K=2$) | | | | |
| 100 | 0.1 | 1 | $\mathbf{81.22^{\pm0.14}}$ | $80.34^{\pm1.44}$ | $79.45^{\pm0.71}$ | $70.01^{\pm0.77}$ | $78.94^{\pm0.82}$ | $71.14^{\pm0.33}$ |
| | | 4 | $\mathbf{67.69^{\pm0.74}}$ | $65.81^{\pm0.84}$ | $63.58^{\pm1.28}$ | $48.47^{\pm1.26}$ | $60.96^{\pm1.11}$ | $57.88^{\pm1.51}$ |
| | 1.0 | 1 | $\mathbf{91.26^{\pm0.83}}$ | $86.84^{\pm0.22}$ | $86.84^{\pm0.83}$ | $85.10^{\pm0.18}$ | $87.03^{\pm0.55}$ | $82.80^{\pm1.28}$ |
| | | 4 | $73.58^{\pm1.02}$ | $\mathbf{74.55^{\pm0.22}}$ | $71.03^{\pm0.75}$ | $58.32^{\pm0.07}$ | $65.76^{\pm0.10}$ | $64.07^{\pm1.89}$ |
| 50 | 0.1 | 1 | $78.63^{\pm0.39}$ | $\mathbf{79.36^{\pm0.24}}$ | $77.44^{\pm1.17}$ | $68.27^{\pm0.53}$ | $78.31^{\pm0.93}$ | $67.95^{\pm0.22}$ |
| | | 4 | $\mathbf{65.08^{\pm1.75}}$ | $63.52^{\pm0.48}$ | $62.65^{\pm0.12}$ | $43.57^{\pm1.99}$ | $58.70^{\pm1.89}$ | $54.20^{\pm1.30}$ |
| | 1.0 | 1 | $\mathbf{89.31^{\pm0.17}}$ | $88.24^{\pm0.71}$ | $86.44^{\pm0.97}$ | $83.31^{\pm0.67}$ | $86.00^{\pm0.93}$ | $82.32^{\pm0.37}$ |
| | | 4 | $\mathbf{70.33^{\pm0.18}}$ | $70.19^{\pm1.41}$ | $67.66^{\pm0.78}$ | $52.48^{\pm1.06}$ | $60.72^{\pm1.26}$ | $60.17^{\pm0.23}$ |

(b) Global prediction (initial accuracy) on the original (uncorrupted) CIFAR-100 training sets

| FL settings | | | Our Methods | | Fed-BABU | Fed-Avg | Fed-Prox | pFedBayes |
|---|---|---|---|---|---|---|---|---|
| $s$ | $f$ | $\tau$ | NIW | Mix. ($K=2$) | | | | |
| 100 | 0.1 | 1 | $\mathbf{41.55^{\pm0.11}}$ | $36.99^{\pm0.09}$ | $34.76^{\pm0.50}$ | $34.40^{\pm0.31}$ | $35.44^{\pm0.66}$ | $35.78^{\pm0.73}$ |
| | | 4 | $\mathbf{30.84^{\pm0.07}}$ | $30.60^{\pm0.27}$ | $28.31^{\pm0.28}$ | $29.24^{\pm0.79}$ | $28.09^{\pm0.71}$ | $29.12^{\pm0.30}$ |
| | 1.0 | 1 | $\mathbf{41.32^{\pm0.32}}$ | $38.35^{\pm0.73}$ | $35.58^{\pm0.41}$ | $36.34^{\pm0.14}$ | $36.32^{\pm0.58}$ | $37.37^{\pm0.62}$ |
| | | 4 | $\mathbf{30.67^{\pm0.12}}$ | $30.40^{\pm0.44}$ | $28.60^{\pm0.28}$ | $30.31^{\pm0.87}$ | $27.95^{\pm0.28}$ | $29.14^{\pm0.57}$ |
| 50 | 0.1 | 1 | $\mathbf{41.04^{\pm0.14}}$ | $36.41^{\pm0.47}$ | $35.44^{\pm0.58}$ | $34.13^{\pm0.50}$ | $36.37^{\pm0.50}$ | $35.68^{\pm0.27}$ |
| | | 4 | $\mathbf{32.29^{\pm0.36}}$ | $31.50^{\pm0.34}$ | $29.68^{\pm0.08}$ | $29.19^{\pm0.14}$ | $29.20^{\pm0.50}$ | $30.10^{\pm0.44}$ |
| | 1.0 | 1 | $\mathbf{41.64^{\pm0.21}}$ | $38.54^{\pm0.42}$ | $36.09^{\pm0.28}$ | $35.78^{\pm0.83}$ | $37.13^{\pm0.55}$ | $38.39^{\pm0.21}$ |
| | | 4 | $\mathbf{32.17^{\pm0.48}}$ | $30.68^{\pm0.46}$ | $29.28^{\pm0.17}$ | $30.45^{\pm0.45}$ | $28.73^{\pm0.26}$ | $29.74^{\pm0.25}$ |

(c) Personalisation performance on the 9 held-out corruption types

| FL settings | | | Our Methods | | Fed-BABU | Fed-Avg | Fed-Prox | pFedBayes |
|---|---|---|---|---|---|---|---|---|
| $s$ | $f$ | $\tau$ | NIW | Mix. ($K=2$) | | | | |
| 100 | 0.1 | 1 | $72.63^{\pm2.13}$ | $\mathbf{74.16^{\pm3.04}}$ | $69.93^{\pm1.24}$ | $62.32^{\pm1.46}$ | $72.94^{\pm0.37}$ | $62.52^{\pm4.01}$ |
| | | 4 | $\mathbf{62.74^{\pm0.94}}$ | $61.56^{\pm1.69}$ | $60.33^{\pm2.12}$ | $53.35^{\pm1.39}$ | $56.91^{\pm1.40}$ | $53.71^{\pm2.13}$ |
| | 1.0 | 1 | $83.62^{\pm1.84}$ | $\mathbf{84.88^{\pm0.85}}$ | $77.55^{\pm2.05}$ | $83.44^{\pm1.68}$ | $80.96^{\pm3.18}$ | $72.44^{\pm0.13}$ |
| | | 4 | $64.84^{\pm1.05}$ | $\mathbf{67.35^{\pm1.46}}$ | $53.25^{\pm2.19}$ | $53.34^{\pm1.22}$ | $41.39^{\pm1.18}$ | $43.41^{\pm1.92}$ |
| 50 | 0.1 | 1 | $\mathbf{75.50^{\pm0.74}}$ | $67.33^{\pm2.83}$ | $59.47^{\pm2.48}$ | $57.77^{\pm0.80}$ | $56.34^{\pm2.80}$ | $53.47^{\pm0.51}$ |
| | | 4 | $44.90^{\pm1.23}$ | $\mathbf{46.39^{\pm0.83}}$ | $44.74^{\pm1.30}$ | $44.60^{\pm2.36}$ | $39.94^{\pm1.81}$ | $37.24^{\pm1.80}$ |
| | 1.0 | 1 | $81.46^{\pm0.67}$ | $\mathbf{81.77^{\pm3.11}}$ | $67.43^{\pm2.58}$ | $72.52^{\pm1.02}$ | $71.52^{\pm2.02}$ | $68.47^{\pm1.43}$ |
| | | 4 | $\mathbf{50.84^{\pm0.74}}$ | $48.90^{\pm0.51}$ | $40.87^{\pm3.01}$ | $45.10^{\pm0.50}$ | $41.95^{\pm1.28}$ | $41.27^{\pm0.14}$ |

in the main experiments. In Fig. 2(a) we report the performance of NIW for different values ($p_{drop} = 0, 10^{-4}, 10^{-2}$) on CIFAR-100 with ($s = 100, f = 0.1, \tau = 1$) setting. We see that the performance is not very sensitive to $p_{drop}$ unless it is too large (e.g., 0.01). For the Mixture model, different mixture orders $K = 2, 5, 10$ are contrasted in Fig. 2(b). As seen, having more mixture components does no harm (no overfitting), but we do not see further improvement over $K = 2$ in our experiments (See also results on CIFAR-C-100 in Table 4 in Appendix).

**Further analysis.** In Appendix, we provide further empirical results: (i) comparison between our mixture model and simple ensemble baselines (Fig. 2(b) and F.2) and (ii) actual running times (F.3).

## 7 CONCLUSION

We have proposed a novel hierarchical Bayesian approach to FL where the block-coordinate descent solution to the variational inference leads to a viable algorithm for FL. Our method not only justifies the previous FL algorithms that look intuitive but theoretically less underpinned, but also generalises them even further via principled Bayesian approaches. With strong theoretical support in convergence rate and generalisation error, our approach is also empirically shown to be superior to recent FL approaches by large margin on several benchmarks with various FL settings.

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

## A    ELBO DERIVATION FOR GENERAL FRAMEWORK

We derive the ELBO objective (3) for the general Bayesian FL framework.

$$
\text{KL}\big(q(\phi, \theta_{1:N}) \,\|\, p(\phi, \theta_{1:N}|D_{1:N})\big) \;=\; \mathbb{E}_q \left[ \log \frac{q(\phi) \cdot \prod_i q_i(\theta_i) \cdot p(D_{1:N})}{p(\phi) \cdot \prod_i p(\theta_i|\phi) \cdot \prod_i p(D_i|\theta_i)} \right] \tag{27}
$$

$$
= \underbrace{\text{KL}(q(\phi)\|p(\phi)) + \sum_{i=1}^{N} \Big( \mathbb{E}_{q_i(\theta_i)}[-\log p(D_i|\theta_i)] + \mathbb{E}_{q(\phi)}\big[\text{KL}(q_i(\theta_i)\|p(\theta_i|\phi))\big] \Big)}_{=:\mathcal{L}(L)}
$$

$$
+ \; \log p(D_{1:N}). \tag{28}
$$

Since KL divergence is non-negative, $-\mathcal{L}(L)$ must be lower bound of the data log-likelihood $\log p(D_{1:N})$, rendering $\mathcal{L}(L)$ as our objective function (to be minimised).

## B    NORMAL-INVERSE-WISHART (NIW) MODEL (DETAILED VERSION)

We define the prior as a conjugate form of Gaussian and Normal-Inverse-Wishart. More specifically, each local client has Gaussian prior $p(\theta_i|\phi) = \mathcal{N}(\theta_i; \mu, \Sigma)$ where $\phi = (\mu, \Sigma)$, and the global latent variable $\phi$ is distributed as a conjugate prior which is Normal-Inverse-Wishart (NIW),

$$
p(\phi) = \mathcal{NIW}(\mu, \Sigma; \Lambda) = \mathcal{N}(\mu; \mu_0, \lambda_0^{-1}\Sigma) \cdot \mathcal{IW}(\Sigma; \Sigma_0, \nu_0), \tag{29}
$$

$$
p(\theta_i|\phi) = \mathcal{N}(\theta_i; \mu, \Sigma), \;\; i = 1, \dots, N, \tag{30}
$$

where $\Lambda = \{\mu_0, \Sigma_0, \lambda_0, \nu_0\}$ is the parameters of the NIW. Although $\Lambda$ can be learned via data marginal likelihood maximisation (e.g., empirical Bayes), but for simplicity we leave it fixed as: $\mu_0 = 0$, $\Sigma_0 = I$, $\lambda_0 = 1$, and $\nu_0 = d + 2$ where $d$ is the number of parameters in $\theta_i$ or $\mu$. This choice ensures that the mean of $\Sigma$ equals $I$, and $\mu$ is distributed as zero-mean Gaussian with covariance $\Sigma$.

Next, our choice of the variational density family for $q(\phi)$ is the NIW, not just because it is the most popular parametric family for a pair of mean vector and covariance matrix $\phi = (\mu, \Sigma)$, but it can also admit closed-form expressions in the ELBO function due to the conjugacy as we derive in Sec. B.1.

$$
q(\phi) := \mathcal{NIW}(\phi; \{m_0, V_0, l_0, n_0\}) = \mathcal{N}(\mu; m_0, l_0^{-1}\Sigma) \cdot \mathcal{IW}(\Sigma; V_0, n_0). \tag{31}
$$

Although the scalar parameters $l_0, n_0$ can be optimised together with $m_0$, $V_0$, their impact is less influential and we find that they make the ELBO optimisation a little bit cumbersome. So we aim to estimate their optimal values in advance with reasonably good quality. To this end, we exploit the conjugacy of the NIW prior-posterior under the Gaussian likelihood. For each $\theta_i$, we pretend that we have instance-wise representative estimates $\theta_i(x, y)$, one for each $(x, y) \in D_i$. For instance, one can view $\theta_i(x, y)$ as the network parameters optimised with the single training instance $(x, y)$. Then this amounts to observing $|D| (= \sum_{i=1}^{N} |D_i|)$ Gaussian samples $\theta_i(x, y) \sim \mathcal{N}(\theta_i; \mu, \Sigma)$ for $(x, y) \sim D_i$ and $i = 1, \dots, N$. Then applying the NIW conjugacy, the posterior is the NIW with $l_0 = \lambda_0 + |D| = |D| + 1$ and $n_0 = \nu_0 + |D| = |D| + d + 2$. This gives us good approximate estimates for the optimal $l_0, n_0$, and we fix them throughout the variational optimisation. Note that this is only heuristics for estimating the scalar parameters $l_0, n_0$ quickly, and the parameters $m_0, V_0$ are determined by the principled ELBO optimisation (Sec. B.1). That is, $L_0 = \{m_0, V_0\}$. Since the dimension $d$ is large (the number of neural network parameters), we restrict $V_0$ to be diagonal for computational tractability.

The density family for $q_i(\theta_i)$'s can be a Gaussian, but we find that it is computationally more attractive and numerically more stable to adopt the mixture of two spiky Gaussians that leads to the MC-Dropout (Gal & Ghahramani, 2016). That is,

$$
q_i(\theta_i) = \prod_l \Big( p \cdot \mathcal{N}(\theta_i[l]; m_i[l], \epsilon^2 I) + (1 - p) \cdot \mathcal{N}(\theta_i[l]; 0, \epsilon^2 I) \Big), \tag{32}
$$

where (i) $m_i$ is the only variational parameters ($L_i = \{m_i\}$), (ii) $\cdot[l]$ indicates the specific column/layer in neural network parameters where $l$ goes over layers and columns of weight matrices, (iii) $p$ is the (user-specified) hyperparameter where $1 - p$ corresponds to the dropout probability, and (iv) $\epsilon$ is a tiny constant (e.g., $10^{-6}$) that makes two Gaussians spiky, close to the delta function. Now we provide more detailed derivations for the client optimisation and server optimisation.

### B.1 DETAILED DERIVATIONS FOR NIW MODEL

**Client update.** We work on the objective function in the general client update optimisation (5). We note that $q(\phi)$ is spiky since our pre-estimated NIW parameters $l_0$ and $n_0$ are large (as the entire training data size $|D|$ is added to the initial prior parameters). Due to the spiky $q(\phi)$, we can accurately approximate the second term in (5) as:

$$\mathbb{E}_{q(\phi)}\big[\mathrm{KL}(q_i(\theta_i)||p(\theta_i|\phi))\big] \approx \mathrm{KL}(q_i(\theta_i)||p(\theta_i|\phi^*)), \tag{33}$$

where $\phi^* = (\mu^*, \Sigma^*)$ is the mode of $q(\phi)$, which has closed forms for the NIW distribution:

$$\mu^* = m_0, \quad \Sigma^* = \frac{V_0}{n_0 + d + 1}. \tag{34}$$

In (33) we have the KL divergence between a mixture of Gaussians (32) and a Gaussian (30). Similar to (Gal & Ghahramani, 2016), we apply the approximation $\mathrm{KL}(\sum_i \alpha_i \mathcal{N}_i || \mathcal{N}) \approx \sum_i \alpha_i \mathrm{KL}(\mathcal{N}_i || \mathcal{N})$ as well as the reparametrised sampling for (32), which allows us to rewrite (5) as:

$$\min_{m_i} \mathcal{L}_i(m_i) := -\log p(D_i|\tilde{m}_i) + \frac{p}{2}(n_0 + d + 1)(m_i - m_0)^\top V_0^{-1}(m_i - m_0), \tag{35}$$

where $\tilde{m}_i$ is the dropout version of $m_i$, i.e., a reparametrised sample from (32). Also, we use a minibatch version of the first term for a tractable SGD update, which amounts to replacing the first term by the batch average $\mathbb{E}_{(x,y)\sim\text{Batch}}[-\log p(y|x, \tilde{m}_i)]$ while downweighing the second term by the factor of $1/|D_i|$. Note that $m_0$ and $V_0$ are fixed during the optimisation. Interestingly (35) generalises the famous Fed-Avg (McMahan et al., 2017) and Fed-Prox (Li et al., 2018): With $p = 1$ (i.e., no dropout) and setting $V_0 = \alpha I$ for some constant $\alpha$, we see that (35) reduces to the client update formula for Fed-Prox where $\alpha$ controls the impact of the proximal term.

**Server update.** The server optimisation (6) involves two terms, both of which we will show admit closed-form expressions thanks to the conjugacy. Furthermore, we show that the optimal solution $(m_0, V_0)$ of (6) has an analytic form. First, the KL term in (6) is decomposed as:

$$\mathrm{KL}(\mathcal{IW}(\Sigma; V_0, n_0)||\mathcal{IW}(\Sigma; \Sigma_0, \nu_0)) + \mathbb{E}_{\mathcal{IW}(\Sigma;V_0,n_0)}[\mathrm{KL}(\mathcal{N}(\mu; m_0, l_0^{-1}\Sigma)||\mathcal{N}(\mu; \mu_0, \lambda_0^{-1}\Sigma))] \tag{36}$$

By some algebra, (36) becomes identical to the following, up to constant, removing those terms that are not dependent on $m_0$,$V_0$ (See Appendix B.2 for derivations):

$$\frac{1}{2}\Big(n_0\mathrm{Tr}(\Sigma_0 V_0^{-1}) + \nu_0 \log |V_0| + \lambda_0 n_0(\mu_0 - m_0)^\top V_0^{-1}(\mu_0 - m_0)\Big). \tag{37}$$

Next, the second term of (6) also admits a closed form as follows (Appendix B.2 for details):

$$-\mathbb{E}_{q(\phi)q_i(\theta_i)}[\log p(\theta_i|\phi)] = \frac{n_0}{2}\Big(pm_i^\top V_0^{-1}m_i - pm_0^\top V_0^{-1}m_i - pm_i^\top V_0^{-1}m_0 + m_0^\top V_0^{-1}m_0$$
$$+ \frac{1}{n_0}\log|V_0| + \epsilon^2\mathrm{Tr}(V_0^{-1})\Big) + \text{const.} \tag{38}$$

That is, server's loss function $\mathcal{L}_0$ is the sum of (37) and (38). We can take the gradients of the loss with respect to $m_0, V_0$ as follows (also plugging $\mu_0 = 0, \Sigma_0 = I, \lambda_0 = 1, \nu_0 = d + 2$):

$$\frac{\partial \mathcal{L}_0}{\partial m_0} = n_0 V_0^{-1}\left((N+1)m_0 - p\sum_{i=1}^N m_i\right), \tag{39}$$

$$\frac{\partial \mathcal{L}_0}{\partial V_0^{-1}} = \frac{1}{2}\left(n_0(1 + N\epsilon^2)I - (N+d+2)V_0 + n_0 m_0 m_0^\top + n_0\sum_{i=1}^N \rho(m_0, m_i, p)\right), \tag{40}$$

$$\text{where } \rho(m_0, m_i, p) = pm_i m_i^\top - pm_0 m_i^\top - pm_i m_0^\top + m_0 m_0^\top.$$

We set the gradients to zero and solve for them, which yields the optimal solution:

$$m_0^* = \frac{p}{N+1}\sum_{i=1}^N m_i, \quad V_0^* = \frac{n_0}{N+d+2}\left((1+N\epsilon^2)I + m_0^*(m_0^*)^\top + \sum_{i=1}^N \rho(m_0^*, m_i, p)\right). \tag{41}$$

Note that $m_i$'s are fixed from clients' latest variational parameters.

It is interesting to see that $m_0^*$ in (41) generalises the well-known aggregation step of averaging local models in Fed-Avg (McMahan et al., 2017) and related methods: when $p = 1$ (i.e., no dropout), it almost[6] equals client model averaging. Also, since $\rho(m_0^*, m_i, p = 1) = (m_i - m_0^*)(m_i - m_0^*)^\top$ when $p = 1$, we can see that $V_0^*$ in (41) essentially estimates the sample scatter matrix with $(N+1)$ samples,

---

[6]Only the constant 1 added to the denominator, which comes from the prior and has the regularising effect.

namely clients' $m_i$'s and server's prior $\mu_0 = 0$, measuring how much they deviate from the center $m_0^*$. It is known that the dropout can help regularise the model and lead to better generalisation (Gal & Ghahramani, 2016), and with $p < 1$ our (41) forms a principled optimal solution.

**Global prediction.** In the inner integral of (8) of the general predictive distribution, we plug $p(\theta|\phi) = \mathcal{N}(\theta; \mu, \Sigma)$ and NIW $q(\phi)$ of (31). This leads to the multivariate Student-$t$ distribution:

$$\int p(\theta|\phi)\, q(\phi)\, d\phi \;=\; \int \mathcal{N}(\theta; \mu, \Sigma) \cdot \mathcal{NIW}(\phi)\, d\phi \;=\; t_{n_0-d+1}\left(\theta; m_0, \frac{(l_0+1)V_0}{l_0(n_0-d+1)}\right), \quad (42)$$

where $t_\nu(a, B)$ is the multivariate Student-$t$ with location $a$, scale matrix $b$, and d.o.f. $\nu$. Then the predictive distribution for a new test input $x^*$ can be estimated as[7]:

$$p(y^*|x^*, D_1, \ldots, D_N) \;=\; \int p(y^*|x^*, \theta) \cdot t_{n_0-d+1}\left(\theta; m_0, \frac{(l_0+1)V_0}{l_0(n_0-d+1)}\right) d\theta \quad (43)$$

$$\approx \frac{1}{S}\sum_{s=1}^{S} p(y^*|x^*, \theta^{(s)}), \quad \text{where } \theta^{(s)} \sim t_{n_0-d+1}\left(\theta; m_0, \frac{(l_0+1)V_0}{l_0(n_0-d+1)}\right). \quad (44)$$

**Personalisation.** With the given personalisation training data $D^p$, we follow the general framework in (11) to find $v(\theta) \approx p(\theta|D^p, \phi^*)$ in a variational way, where $\phi^*$ obtained from (34). For the density family for $v(\theta)$ we adopt the same spiky mixture form as (32),

$$v(\theta) = \prod_l \left( p \cdot \mathcal{N}(\theta[l]; m[l], \epsilon^2 I) + (1-p) \cdot \mathcal{N}(\theta[l]; 0, \epsilon^2 I) \right), \quad (45)$$

where $m$ is the variational parameters. This leads to the MC-dropout-like learning objective,

$$\min_m \; -\log p(D^p|\tilde{m}) + \frac{p}{2}(n_0 + d + 1)(m - m_0)^\top V_0^{-1}(m - m_0), \quad (46)$$

Once $v$ is trained, our predictive distribution follows the MC sampling (12).

## B.2 MATHEMATICAL DETAILS

The server optimisation (6) in our NIW model involves two terms, both of which we will show admit closed-form expressions thanks to the conjugacy. Furthermore, we show that the optimal solution $(m_0, V_0)$ of (6) has an analytic form. First, the KL term in (6) is decomposed as:

$$\text{KL}(q(\phi)||p(\phi)) = \text{KL}(q(\mu|\Sigma)q(\Sigma) \,||\, p(\mu|\Sigma)p(\Sigma)) \quad (47)$$

$$= \mathbb{E}_{q(\Sigma)}\left[ \log \frac{q(\Sigma)}{p(\Sigma)} \right] + \mathbb{E}_{q(\Sigma)}\mathbb{E}_{q(\mu|\Sigma)}\left[ \log \frac{q(\mu|\Sigma)}{p(\mu|\Sigma)} \right] \quad (48)$$

$$= \underbrace{\text{KL}(\mathcal{IW}(\Sigma; V_0, n_0)||\mathcal{IW}(\Sigma; \Sigma_0, \nu_0))}_{=:kl_a} + \underbrace{\mathbb{E}_{\mathcal{IW}(\Sigma; V_0, n_0)}[\text{KL}(\mathcal{N}(\mu; m_0, l_0^{-1}\Sigma)||\mathcal{N}(\mu; \mu_0, \lambda_0^{-1}\Sigma))]}_{=:kl_b}. \quad (49)$$

First we work on $kl_a = \mathbb{E}_{\mathcal{IW}(\Sigma; V_0, n_0)}[\log \mathcal{IW}(\Sigma; V_0, n_0)] - \mathbb{E}_{\mathcal{IW}(\Sigma; V_0, n_0)}[\log \mathcal{IW}(\Sigma; \Sigma_0, \nu_0)]$. From the definition of Inverse-Wishart (assuming $\Sigma = (d \times d)$),

$$\log \mathcal{IW}(\Sigma; \Psi, \nu) = \frac{\nu}{2} \log |\Psi| - \frac{\nu+d+1}{2} \log |\Sigma| - \frac{1}{2}\text{Tr}(\Psi\Sigma^{-1}) - \log \Gamma_d(\nu/2) - \frac{\nu d}{2} \log 2, \quad (50)$$

where $\Gamma_d(\cdot)$ is the multivariate Gamma function. We use the following facts from (Bishop, 2006; Braun & McAuliffe, 2008):

$$\mathbb{E}_{\mathcal{IW}(\Sigma; \Psi, \nu)} \log |\Sigma| \;=\; -d \log 2 + \log |\Psi| - \sum_{i=1}^{d} \psi((\nu - i + 1)/2) \quad (51)$$

$$\mathbb{E}_{\mathcal{IW}(\Sigma; \Psi, \nu)} \Sigma^{-1} \;=\; \nu \Psi^{-1}, \quad (52)$$

where $\psi(\cdot)$ is the digamma function. Applying these to the terms in $kl_a$ yields:

$$kl_a = \frac{1}{2}\left( n_0 \text{Tr}(\Sigma_0 V_0^{-1}) + \nu_0 \log |V_0| \right) + \text{const (w.r.t. } m_0, V_0). \quad (53)$$

---

[7]In practice we use a single sample ($S = 1$) for computational efficiency.

Next, using the closed-form expression for the KL between Gaussians, $kl_b$ becomes:

$$kl_b = \frac{1}{2}\mathbb{E}_{\mathcal{IW}(\Sigma;V_0,n_0)}\left[\lambda_0(\mu_0 - m_0)^\top\Sigma^{-1}(\mu_0 - m_0)\right] + \text{const (w.r.t. } m_0, V_0) \tag{54}$$

$$= \frac{\lambda_0 n_0}{2}(\mu_0 - m_0)^\top V_0^{-1}(\mu_0 - m_0) + \text{const (w.r.t. } m_0, V_0), \tag{55}$$

where in (55) we use the fact (52). Combining (56) and (55), we have:

$$\text{KL}(q(\phi)||p(\phi)) = \frac{1}{2}\Big(n_0\text{Tr}(\Sigma_0 V_0^{-1}) + \nu_0 \log|V_0| + \lambda_0 n_0(\mu_0 - m_0)^\top V_0^{-1}(\mu_0 - m_0)\Big) + \text{const.} \tag{56}$$

Next, we derive the second term of (6) ($=_{utc}$ stands for equality up to constant (w.r.t. $m_0, V_0$)).

$$\mathbb{E}_{q(\phi)q_i(\theta_i)}[\log p(\theta_i|\phi)] = -\frac{1}{2}\mathbb{E}\big[\log|\Sigma| + (\theta_i - \mu)^\top\Sigma^{-1}(\theta_i - \mu)\big] + \text{const (w.r.t. } m_0, V_0) \tag{57}$$

$$=_{utc} -\frac{1}{2}\mathbb{E}_{\mathcal{IW}(\Sigma;V_0,n_0)}\big[\log|\Sigma|\big] - \frac{1}{2}\mathbb{E}\big[(\theta_i - \mu)^\top\Sigma^{-1}(\theta_i - \mu)\big] \tag{58}$$

$$=_{utc} -\frac{1}{2}\log|V_0| - \frac{1}{2}\text{Tr}\,\mathbb{E}_{\mathcal{IW}(\Sigma;V_0,n_0)}\Big[\Sigma^{-1}\mathbb{E}_{\mathcal{N}(\mu;m_0,l_0^{-1}\Sigma)q_i(\theta_i)}\big[(\theta_i - \mu)(\theta_i - \mu)^\top\big]\Big] \tag{59}$$

$$=_{utc} -\frac{1}{2}\log|V_0| - \frac{1}{2}\text{Tr}\,\mathbb{E}_{\mathcal{IW}(\Sigma;V_0,n_0)}\Big[\Sigma^{-1}\big(\rho(m_0, m_i, p) + \epsilon^2 I + l_0^{-1}\Sigma\big)\Big] \tag{60}$$

$$=_{utc} -\frac{1}{2}\log|V_0| - \frac{1}{2}\text{Tr}\Big(\big(\rho(m_0, m_i, p) + \epsilon^2 I\big)n_0 V_0^{-1}\Big) \tag{61}$$

$$=_{utc} -\frac{n_0}{2}\Big(pm_i^\top V_0^{-1}m_i - pm_0^\top V_0^{-1}m_i - pm_i^\top V_0^{-1}m_0 + m_0^\top V_0^{-1}m_0 + \frac{\log|V_0|}{n_0} + \epsilon^2\text{Tr}V_0^{-1}\Big), \tag{62}$$

where $\rho(m_0, m_i, p) = pm_i m_i^\top - pm_0 m_i^\top - pm_i m_0^\top + m_0 m_0^\top$, and we use the definition of $q_i(\theta_i)$ in (32) and the fact (52).

## C   MIXTURE MODEL (DETAILED VERSION)

Previously, the NIW model expresses our prior belief where each client $i$ acquires its own network parameters $\theta_i$ a priori as a Gaussian-perturbed version of the shared parameters $\mu$, namely $\theta_i|\phi \sim \mathcal{N}(\mu, \Sigma)$, as in (14). This is intuitively appealing, but may not be adequate for capturing more drastic diversity in local data across clients. In the situations where clients' local data distributions, as well as their domains and class label semantics, are highly heterogeneous (possibly even set up for adversarial purpose), it would be ideal to consider *multiple different prototypes* for the network parameters, diverse enough to cover the heterogeneity in data distributions across clients. Motivated from this idea, we introduce a mixture prior model as follows.

First we consider that there are $K$ network parameters (prototypes) that can broadly cover the clients data distributions. They are denoted as high-level latent variables, $\phi = \{\mu_1, \ldots, \mu_K\}$, and we let them distributed independently as standard normal a priori,

$$p(\phi) = \prod_{j=1}^{K}\mathcal{N}(\mu_j; 0, I). \tag{63}$$

We here note some clear distinction from the NIW prior. Whereas the NIW prior (13) only controls the mean $\mu$ and covariance $\Sigma$ in the Gaussian, from which local models $\theta_i$ are sampled, the mixture prior (63) is far more flexible in covering highly heterogeneous distributions. Each local model is then assumed to be chosen from one of these $K$ prototypes. Thus the prior distribution for $\theta_i$ can be modeled as a mixture,

$$p(\theta_i|\phi) = \sum_{j=1}^{K}\frac{1}{K}\mathcal{N}(\theta_i; \mu_j; \sigma^2 I), \tag{64}$$

where $\sigma$ is the hyperparameter that captures perturbation scale, and can be chosen by users or learned from data. Note that we put equal mixing proportions $1/K$ due to the symmetry, a priori. That is, each client can take any of $\mu_j$'s equally likely a priori.

We then describe our choice of the variational density $q(\phi) \prod_i q_i(\theta_i)$ to approximate the posterior $p(\phi, \theta_{1:N} | D_{1:N})$. First, $q_i(\theta_i)$ is chosen as a Gaussian,

$$q_i(\theta_i) = \mathcal{N}(\theta_i; m_i, \epsilon^2 I), \tag{65}$$

with small $\epsilon$. For $q(\phi)$ we consider a Gaussian factorised over $\mu_j$'s, but with small variances, that is,

$$q(\phi) = \prod_{j=1}^{K} \mathcal{N}(\mu_j; r_j, \epsilon^2 I), \tag{66}$$

where $\{r_j\}_{j=1}^{K}$ are variational parameters ($L_0$) and $\epsilon$ is small (e.g., $10^{-4}$). The main reason why we make $q(\phi)$ spiky is that the resulting near-deterministic $q(\phi)$ allows for computationally efficient and accurate MC sampling during ELBO optimisation as well as test time (global) prediction, avoiding difficult marginalisation (Sec. C.1 for details). Although Bayesian inference in general encourages to retain as many plausible latent states as possible under the given evidence (observed data), we aim to model this uncertainty by having many (possibly redundant) prototypes $\mu_j$'s rather than imposing larger variance for a single one (e.g., finite-sample approximation of a smooth distribution).

## C.1 Detailed Derivations for Mixture Model

With the full specification of the prior distribution and the variational density family, we are ready to dig into the client objective function (5) and the server (6).

**Client update.** Since $q(\phi)$ is spiky, we can accurately approximate the second term of (5) as $\text{KL}(q_i(\theta_i)||p(\theta_i|\phi^*))$ where $\phi^* = \{\mu_j^* = r_j\}_{j=1}^{K}$ is the mode of $q(\phi)$ since

$$\mathbb{E}_{q(\phi)q_i(\theta_i)}[\log p(\theta_i|\phi)] \approx \mathbb{E}_{q_i(\theta_i)}[\log p(\theta_i|\phi^*)]. \tag{67}$$

Since $q_i(\theta_i)$ is also spiky, $\text{KL}(q_i(\theta_i)||p(\theta_i|\phi^*))$, the KL divergence between a Gaussian and a Gaussian mixture, can be approximated accurately using the single mode sample $m_i \sim q_i(\theta_i)$, that is,

$$\text{KL}(q_i(\theta_i)||p(\theta_i|\phi^*)) \approx \log q_i(m_i) - \log p(m_i|\phi^*) \tag{68}$$

$$= -\log \sum_{j=1}^{K} \mathcal{N}(m_i; r_j, \sigma^2 I) + \text{const.} = -\log \sum_{j=1}^{K} \exp\left( -\frac{||m_i - r_j||^2}{2\sigma^2} \right) + \text{const.} \tag{69}$$

Note here that we use the fact that $m_i$ disappears in $\log q_i(m_i)$. Plugging it into (5) yields the following optimisation for client $i$:

$$\min_{m_i} \mathbb{E}_{q_i(\theta_i)}[-\log p(D_i|\theta_i)] - \log \sum_{j=1}^{K} \exp\left( -\frac{||m_i - r_j||^2}{2\sigma^2} \right). \tag{70}$$

It is interesting to see that (70) can be seen as generalisation of Fed-Prox (Li et al., 2018), where the proximal regularisation term in Fed-Prox is extended to *multiple* global models $r_j$'s, penalizing the local model ($m_i$) straying away from these prototypes. And if we use a single prototype ($K = 1$), the optimisation (70) exactly reduces to the local update objective of Fed-Prox. Since `log-sum-exp` is approximately equal to `max`, the regularisation term in (70) effectively focuses on the closest global prototype $r_j$ from the current local model $m_i$, which is intuitively well aligned with our initial modeling motivation, namely each local data distribution is explained by one of the global prototypes. Lastly, we also note that in the SGD optimisation setting where we can only access a minibatch $B \sim D_i$ during the optimisation of (70), we follow the conventional practice: replacing the first term of the negative log-likelihood by a stochastic estimate $\mathbb{E}_{q_i(\theta_i)}\mathbb{E}_{(x,y)\sim B}[-\log p(y|x, \theta_i)]$ and multiplying the second term of regularisation by $\frac{1}{|D_i|}$.

**Server update.** First, the KL term in (6) can be easily derived as:

$$\text{KL}(q(\phi)||p(\phi)) = \frac{1}{2} \sum_{j=1}^{K} ||r_j||^2 + \text{const.} \tag{71}$$

and the second term of (6) approximated as follows:

$$\mathbb{E}_{q(\phi)q_i(\theta_i)}[\log p(\theta_i|\phi)] \approx \mathbb{E}_{q(\phi)}[\log p(m_i|\phi)] \approx \log \sum_{j=1}^{K} \frac{1}{K} \mathcal{N}(m_i; r_j, \sigma^2 I) \tag{72}$$

$$= \log \sum_{j=1}^{K} \exp\left( -\frac{||m_i - r_j||^2}{2\sigma^2} \right) + \text{const.} \tag{73}$$

where the approximations in (72) become accurate due to spiky $q_i(\theta_i)$ and $q(\phi)$, respectively. Combining the two terms leads to the optimisation problem for the server:

$$\min_{\{r_j\}_{j=1}^K} \frac{1}{2} \sum_{j=1}^K ||r_j||^2 - \sum_{i=1}^N \log \sum_{j=1}^K \exp\left( -\frac{||m_i - r_j||^2}{2\sigma^2} \right). \tag{74}$$

Interestingly, (74) generalises the well-known aggregation step of averaging local models in Fed-Avg (McMahan et al., 2017) and related methods: Especially when $K = 1$, (74) reduces to quadratic optimisation, admitting the optimal solution $r_1^* = \frac{1}{N+\sigma^2} \sum_{i=1}^N m_i$. The extra term $\sigma^2$ in the denominator can be explained by incorporating an extra *zero* local model originating from the prior (interpreted as a *neutral* model) with the discounted weight $\sigma^2$ rather than 1.

Although (74) for $K > 1$ can be solved by standard gradient descent, the objective function resembles the (regularised) Gaussian mixture log-likelihood, and we can apply the Expectation-Maximisation (EM) algorithm (Dempster et al., 1977) instead. Using Jensen's bound with convexity of the negative `log` function, we have the following alternating steps[8]:

- **E-step:** With the current $\{r_j\}_{j=1}^K$ fixed, compute the prototype assignment probabilities for each local model $m_i$:

$$c(j|i) = \frac{k_{ij}}{\sum_{j=1}^K k_{ij}}, \quad \text{where } k_{ij} = \exp\left( -\frac{||m_i - r_j||^2}{2\sigma^2} \right). \tag{75}$$

- **M-step:** With the current assignments $c(j|i)$ fixed, we solve:

$$\min_{\{r_j\}} \frac{1}{2} \sum_j ||r_j||^2 + \frac{1}{2\sigma^2} \sum_{i,j} c(j|i) \cdot ||m_i - r_j||^2, \tag{76}$$

which admits the closed form solution:

$$r_j^* = \frac{\frac{1}{N} \sum_{i=1}^N c(j|i) \cdot m_i}{\frac{\sigma^2}{N} + \frac{1}{N} \sum_{i=1}^N c(j|i)}, \quad j = 1, \dots, K. \tag{77}$$

The server update equation (77) has intuitive meaning that the new prototype $r_j$ becomes the *weighted* average of the local models $m_i$'s where the weights $c(j|i)$ are determined by the proximity to $r_j$ (i.e., those $m_i$'s that are closer to $r_j$ have more contribution, and vice versa). This can be seen as an extension of the aggregation step in Fed-Avg to the multiple prototype case.

**Global prediction.** By plugging the mixture prior $p(\theta|\phi)$ of (64) and the factorised spiky Gaussian $q(\phi)$ of (66) into the inner integral of (8), we have predictive distribution averaged equally over $\{r_j\}_{j=1}^K$ approximately, that is, $\int p(\theta|\phi) q(\phi) d\phi \approx \frac{1}{K} \sum_{j=1}^K p(y^*|x^*, r_j)$. Unfortunately this is not ideal for our original intention where only one specific model $r_j$ out of $K$ candidates is dominantly responsible for the local data. To meet this intention, we extend our model so that the input point $x^*$ can affect $\theta$ together with $\phi$, and with this modification our predictive probability can be derived as:

$$p(y^*|x^*, D_{1:N}) = \iint p(y^*|x^*, \theta) \, p(\theta|x^*, \phi) \, p(\phi|D_{1:N}) \, d\theta d\phi \tag{78}$$

$$\approx \iint p(y^*|x^*, \theta) \, p(\theta|x^*, \phi) \, q(\phi) \, d\theta d\phi \tag{79}$$

$$\approx \int p(y^*|x^*, \theta) \, p(\theta|x^*, \{r_j\}_{j=1}^K) \, d\theta. \tag{80}$$

To deal with the tricky part of inferring $p(\theta|x^*, \{r_j\}_{j=1}^K)$, we introduce a fairly practical strategy of fitting a *gating function*. The idea is to regard $p(\theta|x^*, \{r_j\}_{j=1}^K)$ as a mixture of experts (Jacobs et al., 1991; Jordan & Jacobs, 1994) where the prototypes $r_j$'s serving as experts,

$$p(\theta|x^*, \{r_j\}_{j=1}^K) := \sum_{j=1}^K g_j(x^*) \cdot \delta(\theta - r_j), \tag{81}$$

where $\delta(\cdot)$ is the Dirac's delta function, and $g(x)$ is a gating function that outputs a $K$-dimensional softmax vector. Intuitively, the gating function determines which of the $K$ prototypes $\{r_j\}_{j=1}^K$ the

---

[8]Although one can perform several EM steps until convergence, in practice, we find that only one EM step per round is sufficient.

model $\theta$ for the test point $x^*$ belongs to. With (81), the predictive probability in (80) is written as:

$$p(y^*|x^*, D_{1:N}) \approx \sum_{j=1}^{K} g_j(x^*) \cdot p(y^*|x^*, r_j). \tag{82}$$

However, since we do not have this oracle $g(x)$, we introduce and fit a neural network to the local training data during the training stage. Let $g(x; \beta)$ be the gating network with the parameters $\beta$. To train it, we follow the Fed-Avg[9] strategy. In the client update stage at each round, while we update the local model $m_i$ with a minibatch $B \sim D_i$, we also find the prototype closest to $m_i$, namely $j^* := \arg\min_j ||m_i - r_j||$. Then we form another minibatch of samples $\{(x, j^*)\}_{x \sim B}$ (input $x$ and class label $j^*$), and update $g(x; \beta)$ by SGD. The updated (local) $\beta$'s from the clients are then aggregated (by simple averaging) by the server, and distributed back to the clients as an initial iterate for the next round.

**Personalisation.** For $p(\theta|D^p, \phi^*)$ in the general framework (10), we define the variational distribution $v(\theta) \approx p(\theta|D^p, \phi^*)$ as:

$$v(\theta) = \mathcal{N}(\theta; m, \epsilon^2 I), \tag{83}$$

where $\epsilon$ is small positive constant, and $m$ is the only parameters that we learn. Our personalisation training amounts to ELBO optimisation for $v(\theta)$ as in (11), which reduces to:

$$\min_m \mathbb{E}_{v(\theta)}[-\log p(D^p|\theta)] - \log \sum_{j=1}^{K} \exp\left(-\frac{||m - r_j||^2}{2\sigma^2}\right). \tag{84}$$

Once we have optimal $m$ (i.e., $v(\theta)$), our predictive model becomes:

$$p(y^p|x^p, D^p, D_{1:N}) \approx p(y^p|x^p, m), \tag{85}$$

which is done by feed-forwarding test input $x^p$ through the network deployed with the parameters $m$.

## D    CONVERGENCE ANALYSIS

Our (general) FL algorithm is a special block-coordinate SGD optimisation of the ELBO function (3) with respect to the $(N + 1)$ parameter groups: $L_0$ (of $q(\phi; L_0)$), $L_1$ (of $q_1(\theta_1; L_1)$), ..., and $L_N$ (of $q_N(\theta_N; L_N)$). In this section we will provide a theorem that guarantees convergence of the algorithm to a local minimum of the ELBO objective function under some mild assumptions. We will also analyse the convergence rate. Note that although our FL algorithm is a special case of the general block-coordinate SGD optimisation, we may not directly apply the existing convergence results for the regular block-coordinate SGD methods since they mostly rely on non-overlapping blocks with cyclic or uniform random block selection strategies (Beck & Tetruashvili, 2013; Wang & Banerjee, 2014). As the block selection strategy in our FL algorithm is unique with overlapping blocks and non-uniform random block selection, we provide our own analysis here. Promisingly, we show that in accordance with general regular block-coordinate SGD (cyclic/uniform non-overlapping block selection), our FL algorithm has $O(1/\sqrt{t})$ convergence rate, which is also asymptotically the same as that of the conventional (holistic, non-block-coordinate) SGD optimisation. Note that this section is about the convergence of our algorithm to an (local) optimum of the training objective (ELBO). The question of how well this optimal model trained on empirical data performs on the unseen data points will be discussed in Sec. E.

First we formally describe our FL algorithm as a block-coordinate SGD optimisation. For ease of exhibition, we will simplify the notation: The objective function in (3) is denoted as $f(x)$ where $x = [x_0, x_1, \ldots, x_N]$ is the optimisation variables corresponding to $x_0 := L_0$, $x_1 := L_1$, ..., $x_N := L_N$. That is, $x_0$ is server's parameters while $x_i$ ($i = 1, \ldots, N$) is worker $i$'s parameters. We let $x_u$ be the partial vector of $x$ selected by the index set $u \subseteq \{0, 1, \ldots, N\}$, and $x_{-u}$ be the vector of the rest elements. Similarly $\nabla_u f(x)$ indicates the gradient vector with only elements at the indices in $u$. Let $x^t$ be the iterate at iteration $t$. Our FL algorithm is formally defined in Alg. 1.

Now we state our convergence theorem. We first need the following mild assumptions:

**Assumption 1.** Our objective function $f(x)$ is *locally* convex, and $f_t(x)$ is also *locally* convex for all iterations $t$, where $f_t$ is the minibatch version of $f$ defined on the minibatch data $batch_t$ (so that $\mathbb{E}_{batch_t}[f_t(x)] = f(x)$). Actually, the latter implies the former. Although the negative ELBO is in

---

[9]We also follow the Fed-BABU (Oh et al., 2022) strategy by updating only the body of $\beta$ and fixing/sharing the random classification head across the server and clients.

---

**Algorithm 1** Bayesian FL Algorithm as Block-Coordinate Descent.

---

We define the following hyperparameters:
- $N_f$ $(\leq N)$ = the number of participating clients at each round.
- $M$ = the number of SGD iterations per round for updating the (participating) clients.
- Let $M_S$ = the number of SGD iterations per round for updating the server.
- Let $\eta_t$ = the reciprocal of the SGD learning rate at iteration $t$.

Initialise the global iteration counter $t = 0$.

**for** each round **do**
- Select $N_f$ clients uniformly at random from $\{1, \ldots, N\}$ without replacement. Let $u_t \subseteq \{1, \ldots, N\}$ be the set of the participants ($|u_t| = N_f$).
- (Client update) For each of $M$ iterations,
    1. Perform an SGD update for the block $u_t$. That is,
    $$x^{t+1} := [x_{u_t}^{t+1}; x_{-u_t}^t], \quad \text{where} \quad x_{u_t}^{t+1} = x_{u_t}^t - \frac{1}{\eta_t} \nabla_{u_t} f_t(x^t), \qquad (86)$$
    where $f_t$ is the minibatch version of $f$ defined on the minibatch data $batch_t$ (so that $\mathbb{E}_{batch_t}[f_t(x)] = f(x)$). Note that this update is actually done independently over the participating clients $i \in u_t$ due to the separable objective, i.e., from (4) to (5).
    2. $t \leftarrow t + 1$.
- (Server update) For each of $M_S$ iterations,
    1. Perform SGD update for the index (singleton block) 0. That is,
    $$x^{t+1} := [x_0^{t+1}; x_{-0}^t], \quad \text{where} \quad x_0^{t+1} = x_0^t - \frac{1}{\eta_t} \nabla_0 f_t(x^t). \qquad (87)$$
    2. $t \leftarrow t + 1$.

**end for**

---

general non-convex globally, we can regard it as a convex function within a local neighborhood, as is usually assumed in non-convex analysis (Bertsekas, 2016) and other FL analysis (Li et al., 2020).

**Assumption 2.** For all $t$, $f_t(x)$ has Lipschitz continuous gradient with constant $\overline{L}$.

**Assumption 3.** For all $t$, $||\nabla f_t(x)|| \leq R_f$ and $||x - x'|| \leq D$ for any $x$, $x'$, where $R_f$ and $D$ are some constants.

**Theorem D.1** (Convergence analysis). *Let $\eta_t = \overline{L} + \sqrt{t}$, $M_S = \frac{M \cdot N_f}{N}$, and $\overline{x}^T = \frac{1}{T} \sum_{t=1}^{T} x^t$ by following our FL algorithm. With Assumptions 1–3, the following holds for any $T$:*

$$\mathbb{E}[f(\overline{x}^T)] - f(x^*) \leq \frac{N + N_f}{N_f} \cdot \frac{\frac{\sqrt{T}+\overline{L}}{2} D^2 + R_f^2 \sqrt{T}}{T} = O\left(\frac{1}{\sqrt{T}}\right), \qquad (88)$$

*where $x^*$ is the (local) optimum, and the expectation is taken over randomness in minibatches and block selections $\{u_t\}_{t=0}^{T-1}$.*

*Remark.* Theorem D.1 states that $\overline{x}^t$ converges to the optimal point $x^*$ in expectation at the rate of $O(1/\sqrt{t})$. This convergence rate asymptotically equals that of the conventional (holistic, non-block-coordinate) SGD algorithm.

To prove the theorem, we note that the algorithm Alg. 1 overall repeats the following three steps per round: i) sample the subset of clients $u_t$ from $\{1, \ldots, N\}$ with $|u_t| = N_f$, ii) update $x_{u_t}$ for $M$ iterations, and iii) update $x_0$ for $M_S$ iterations. Thus in the long-term view, we can see that the algorithm proceeds as follows: At each iteration $t$, we select $u_t$ as

$$u_t = \begin{cases} \{0\} & \text{with prob. } \frac{M_S}{M+M_S} \\ \text{Size-}N_f \text{ subset uniformly from } \{1, \ldots, N\} & \text{with prob. } \frac{M}{M+M_S} \end{cases}, \qquad (89)$$

and update the iterate as

$$x^{t+1} := [x_{u_t}^{t+1}; x_{-u_t}^t], \quad \text{where} \quad x_{u_t}^{t+1} = x_{u_t}^t - \frac{1}{\eta_t} \nabla_{u_t} f_t(x^t). \qquad (90)$$

We will use this long-term view strategy in our proof. Next we state the following lemma that is motivated from (Wang & Banerjee, 2014), which is useful in our proof.

**Lemma D.2.** *Assume $\eta_t > \overline{L}$, and $f_t$ is (locally) convex with Lipschitz continuous gradient with constant $\overline{L}$. For any subset $u \subseteq \{0, 1, \ldots, N\}$, we define $x_{t+1}$ as:*

$$x_u^{t+1} = x_u^t - \frac{1}{\eta_t} \nabla_u f_t(x^t) \quad and \quad x_{-u}^{t+1} = x_{-u}^t. \qquad (91)$$

*Then the following holds for any x:*

$$\langle \nabla_u f_t(x^t), x_u^t - x_u \rangle \leq \frac{\eta_t}{2} \left( ||x - x^t||^2 - ||x - x^{t+1}||^2 \right) + \frac{R_f^2}{2(\eta_t - \overline{L})}. \tag{92}$$

*Proof of Lemma D.2.* By definition, $\nabla_u f_t(x^t) + \eta_t(x_u^{t+1} - x_u^t) = 0$. Then for any $x$, we have

$$\langle \nabla_u f_t(x^t), x_u^{t+1} - x_u \rangle = -\eta_t \langle x_u^{t+1} - x_u^t, x_u^{t+1} - x_u \rangle \tag{93}$$

$$= \frac{\eta_t}{2} \left( ||x_u - x_u^t||^2 - ||x_u - x_u^{t+1}||^2 - ||x_u^{t+1} - x_u^t||^2 \right) \tag{94}$$

$$= \frac{\eta_t}{2} \left( ||x - x^t||^2 - ||x - x^{t+1}||^2 - ||x_u^{t+1} - x_u^t||^2 \right). \tag{95}$$

Note that (95) follows from (94) since $x^t$ and $x^{t+1}$ only differ at indices $u$. Since $f_t$ has Lipschitz continuous gradient,

$$f_t(x^{t+1}) - f_t(x^t) \leq \langle \nabla_u f_t(x^t), x_u^{t+1} - x_u^t \rangle + \frac{\overline{L}}{2} ||x_u^{t+1} - x_u^t||^2 \tag{96}$$

$$= \langle \nabla_u f_t(x^t), x_u^{t+1} - x_u \rangle + \frac{\overline{L}}{2} ||x_u^{t+1} - x_u^t||^2 - \langle \nabla_u f_t(x^t), x_u^t - x_u \rangle \tag{97}$$

$$= \frac{\eta_t}{2} \left( ||x - x^t||^2 - ||x - x^{t+1}||^2 - ||x_u^{t+1} - x_u^t||^2 \right) + \frac{\overline{L}}{2} ||x_u^{t+1} - x_u^t||^2$$
$$- \langle \nabla_u f_t(x^t), x_u^t - x_u \rangle, \tag{98}$$

where we plugged (95) into (97). We rearrange the terms in (98) as follows:

$$\langle \nabla_u f_t(x^t), x_u^t - x_u \rangle \leq \frac{\eta_t}{2} \left( ||x - x^t||^2 - ||x - x^{t+1}||^2 \right) + \frac{\overline{L} - \eta_t}{2} ||x_u^{t+1} - x_u^t||^2$$
$$+ (f_t(x^t) - f_t(x^{t+1})). \tag{99}$$

Due to the convexity of $f_t$, we can bound the last term in (99) as

$$f_t(x^t) - f_t(x^{t+1}) \leq \langle \nabla f_t(x^t), x^t - x^{t+1} \rangle \tag{100}$$

$$= \langle \nabla_u f_t(x^t), x_u^t - x_u^{t+1} \rangle \tag{101}$$

$$\leq \frac{1}{2\alpha} ||\nabla_u f_t(x^t)||^2 + \frac{\alpha}{2} ||x_u^t - x_u^{t+1}||^2 \quad \text{(for any } \alpha > 0\text{)}. \tag{102}$$

Plugging (102) into (99) and choosing $\alpha = \eta_t - \overline{L}(> 0)$ yields:

$$\langle \nabla_u f_t(x^t), x_u^t - x_u \rangle \leq \frac{\eta_t}{2} \left( ||x - x^t||^2 - ||x - x^{t+1}||^2 \right) + \frac{||\nabla_u f_t(x^t)||^2}{2(\eta_t - \overline{L})}. \tag{103}$$

Applying Assumption 3 of the bounded gradient norm completes the proof. □

Now we are ready to prove our convergence theorem (Theorem D.1).

*Proof of Theorem D.1.* We first aim to bound $\langle \nabla f(x^t), x^t - x \rangle$, as it upper-bounds of $f(x^t) - f(x)$ for convex $f$. Note that by conditioning on $x^t$, we can only deal with randomness in minibatch at $t$ ($batch_t$), that is,

$$\langle \nabla f(x^t), x^t - x \rangle = \mathbb{E}_{batch_t} [\langle \nabla f_t(x^t), x^t - x \rangle]. \tag{104}$$

Further conditioning on $batch_t$ leads to:

$$\langle \nabla f_t(x^t), x^t - x \rangle = \langle \nabla_0 f_t(x^t), x_0^t - x_0 \rangle + \sum_{i=1}^{N} \langle \nabla_i f_t(x^t), x_i^t - x_i \rangle. \tag{105}$$

Here we aim to rewrite the summation term in (105) in terms of size $N_f$ blocks. To this end, let us consider:

$$\sum_{u \in 2^{N,N_f}} \langle \nabla_u f_t(x^t), x_u^t - x_u \rangle, \tag{106}$$

where $2^{N,N_f}$ is defined as *the set of all subsets of $\{1, \ldots, N\}$ with size $N_f$ and no repeating elements.* For instance, $2^{5,3}$ contains $\{1, 3, 4\}$ and $\{2, 4, 5\}$, among others. Obviously $|2^{N,N_f}| = \binom{N}{N_f}$, and each particular index $i \in \{1, \ldots, N\}$ appears exactly $\binom{N-1}{N_f-1}$ times in the sum (106). Thus we can

establish the following identity:

$$\sum_{u \in 2^{N,N_f}} \langle \nabla_u f_t(x^t), x_u^t - x_u \rangle = \binom{N-1}{N_f-1} \sum_{i=1}^{N} \langle \nabla_i f_t(x^t), x_i^t - x_i \rangle. \tag{107}$$

We plug (107) into (105) and apply Lemma D.2:

$$\langle \nabla f_t(x^t), x^t - x \rangle = \langle \nabla_0 f_t(x^t), x_0^t - x_0 \rangle + \frac{1}{\binom{N-1}{N_f-1}} \sum_{u \in 2^{N,N_f}} \langle \nabla_u f_t(x^t), x_u^t - x_u \rangle \tag{108}$$

$$\leq \frac{\eta_t}{2} \left( ||x - x^t||^2 - ||x - x^{t+1}(0, batch_t)||^2 \right) + \frac{R_f^2}{2(\eta_t - \overline{L})}$$

$$+ \frac{1}{\binom{N-1}{N_f-1}} \sum_{u \in 2^{N,N_f}} \left( \frac{\eta_t}{2} \left( ||x - x^t||^2 - ||x - x^{t+1}(u, batch_t)||^2 \right) + \frac{R_f^2}{2(\eta_t - \overline{L})} \right), \tag{109}$$

where we define $x^{t+1}(u, batch_t)$ for any subset $u \subseteq \{0, 1, \dots, N\}$ as: $x_u^{t+1} := x_u^t - (1/\eta_t) \nabla_u f_t(x^t)$ and $x_{-u}^{t+1} := x_{-u}^t$. Although we can simply use $x^{t+1}$, here we use this explicit notation to emphasise dependency of $x^{t+1}$ on $i$ and $batch_t$. By letting $g(x, x^t, x^{t+1}) := \frac{\eta_t}{2} \left( ||x - x^t||^2 - ||x - x^{t+1}||^2 \right) + \frac{R_f^2}{2(\eta_t - \overline{L})}$, we can express (109) succinctly to yield:

$$\langle \nabla f_t(x^t), x^t - x \rangle \leq g(x, x^t, x^{t+1}(0, batch_t)) + \frac{1}{\binom{N-1}{N_f-1}} \sum_{u \in 2^{N,N_f}} g(x, x^t, x^{t+1}(u, batch_t)). \tag{110}$$

For the second term, we use the uniform expectation to replace the sum (i.e., $\sum_{u \in 2^{N,N_f}} \psi(u) = \binom{N}{N_f} \mathbb{E}_{u \sim 2^{N,N_f}} [\psi(u)]$ for any function $\psi$). Using $\binom{N}{N_f} / \binom{N-1}{N_f-1} = N/N_f$,

$$\langle \nabla f_t(x^t), x^t - x \rangle \leq g(x, x^t, x^{t+1}(0, batch_t)) + \frac{N}{N_f} \mathbb{E}_{u \sim 2^{N,N_f}} [g(x, x^t, x^{t+1}(u, batch_t))], \tag{111}$$

and the right hand side of (111) can be written as:

$$\frac{M + M_S}{M_S} \left( \frac{M_S}{M + M_S} g(x, x^t, x^{t+1}(0, batch_t)) + \frac{M}{M + M_S} \mathbb{E}_{u \sim 2^{N,N_f}} [g(x, x^t, x^{t+1}(u, batch_t))] \right), \tag{112}$$

where we use our specification of $M_S = \frac{M \cdot N_f}{N}$. Note that the expression inside the parentheses is exactly the expectation of $g(x, x^t, x^{t+1}(u_t, batch_t))$ over the random index set $u_t$ following our client-server selection strategy in the long term view, that is, (89). Then we have the following result:

$$\langle \nabla f_t(x^t), x^t - x \rangle \leq \frac{M + M_S}{M_S} \mathbb{E}_{u_t} [g(x, x^t, x^{t+1}(u_t, batch_t))], \tag{113}$$

where $u_t$ follows (89).

As we have conditioned all quantities on $batch_t$, we now take the expectation over $batch_t$.

$$\langle \nabla f(x^t), x^t - x \rangle = \mathbb{E}_{batch_t} [\langle \nabla f_t(x^t), x^t - x \rangle] \tag{114}$$

$$\leq \frac{M + M_S}{M_S} \mathbb{E}_{batch_t, u_t} \left[ \frac{\eta_t}{2} \left( ||x - x^t||^2 - ||x - x^{t+1}||^2 \right) + \frac{R_f^2}{2\sqrt{t}} \right], \tag{115}$$

where we drop the dependency in $x^{t+1}(u_t, batch_t)$ in notation, and use $\eta_t = \overline{L} + \sqrt{t}$. Since $f$ is convex,

$$f(x^t) - f(x) \leq \langle \nabla f(x^t), x^t - x \rangle, \tag{116}$$

and taking the expectation over $x^t$ leads to:

$$\mathbb{E}[f(x^t)] - f(x) \leq \frac{M + M_S}{M_S} \left( \frac{\eta_t}{2} \left( \mathbb{E}||x - x^t||^2 - \mathbb{E}||x - x^{t+1}||^2 \right) + \frac{R_f^2}{2\sqrt{t}} \right). \tag{117}$$

By telescoping ($\frac{1}{T} \sum_{t=1}^{T}$) and using $\frac{M + M_S}{M_S} = \frac{N + N_f}{N_f}$, we have:

$$\mathbb{E} \left[ \frac{1}{T} \sum_{t=1}^{T} f(x^t) \right] - f(x) \leq \frac{N + N_f}{N_f} \frac{1}{T} \left( \frac{1}{2} \sum_{t=1}^{T} \eta_t \left( \mathbb{E}||x - x^t||^2 - \mathbb{E}||x - x^{t+1}||^2 \right) + R_f^2 \sum_{t=1}^{T} \frac{1}{2\sqrt{t}} \right). \tag{118}$$

There are two sums in the right hand side of (118), and they can be bounded succinctly as follows. First, we use the simple calculus to bound the second sum: $\sum_{t=1}^{T} \frac{1}{2\sqrt{t}} \leq \int_{1}^{T} \frac{1}{2\sqrt{z}} dz + \frac{1}{2} \leq \sqrt{T}$. Next, we let $a_t := \mathbb{E}||x - x^t||^2$, and the first sum is written as: $\eta_1(a_1 - a_2) + \cdots + \eta_T(a_T - a_{T+1}) = \sum_{t=1}^{T}(\eta_t - \eta_{t-1})a_t - \eta_T a_{T+1}$ by letting $\eta_0 = 0$. Using $a_t \leq D^2$ from Assumption 3, this sum is bounded above by $D^2 \sum_{t=1}^{T}(\eta_t - \eta_{t-1}) = D^2(\eta_T - \eta_0) = (\sqrt{T} + \overline{L})D^2$. Plugging these bounds into (118) and applying Jensen's inequality to the left hand side (i.e., $\mathbb{E}[(1/T)\sum_{t=1}^{T} f(x^t)] \geq \mathbb{E}[f((1/T)\sum_{t=1}^{T} x^t)] = \mathbb{E}[f(\overline{x}^T)]$) yields:

$$\mathbb{E}[f(\overline{x}^T)] - f(x) \leq \frac{N + N_f}{N_f} \cdot \frac{\frac{\sqrt{T}+\overline{L}}{2}D^2 + R_f^2\sqrt{T}}{T}, \tag{119}$$

for any $x$, which completes the proof. $\qquad\square$

## E  GENERALISATION ERROR BOUND

In this section we will discuss generalisation performance of our proposed algorithm, answering the question of how well the Bayesian FL model trained on empirical data performs on the unseen data points. We aim to provide the upper bound of the generalisation error averaged over the posterior distribution of the model parameters $(\phi, \{\theta_i\}_{i=1}^{N})$, by linking it to the expected empirical error with some additional complexity terms.

To this end, we first consider the PAC-Bayes bounds (McAllester, 1999; Langford & Caruana, 2001; Seeger, 2002; Maurer, 2004), naturally because they have similar forms relating the two error terms (generalisation and empirical) expected over the posterior distribution via the KL divergence term between the posterior and the prior distributions. However, the original PAC-Bayes bounds have the square root of the KL in the bound, which deviates from the ELBO objective function that has the sum of the expected data loss and the KL term as it is (instead of the square root). However, there are some recent variants of PAC-Bayes bounds, specifically the *PAC-Bayes-$\lambda$ bound*, which removes the square root of the KL and suits better with the ELBO objective function (See (Thiemann et al., 2017) or Eq. (5) of (Rivasplata et al., 2019)).

To discuss it further, the objective function of our FL algorithm (3) can be viewed as a conventional variational inference ELBO objective with the prior $p(\beta)$ and the posterior $q(\beta)$, where $\beta = \{\phi, \theta_1, \ldots, \theta_N\}$ indicates the set of all latent variables in our model. More specifically, the negative ELBO (function of the variational posterior distribution $q$) can be written as:

$$\text{-ELBO}(q) = \mathbb{E}_{q(\beta)}[\hat{l}_n(\beta)] + \frac{1}{n}\text{KL}(q(\beta)||p(\beta)), \tag{120}$$

where $\hat{l}_n(\beta)$ is the empirical error/loss of the model $\beta$ on the training data of size $n$. We then apply the PAC-Bayes-$\lambda$ bound (Thiemann et al., 2017; Rivasplata et al., 2019); for any $\lambda \in (0, 2)$, the following holds with probability at least $1 - \delta$:

$$\mathbb{E}_{q(\beta)}[l(\beta)] \leq \frac{1}{1 - \lambda/2}\mathbb{E}_{q(\beta)}[\hat{l}_n(\beta)] + \frac{1}{\lambda(1 - \lambda/2)}\frac{\text{KL}(q(\beta)||p(\beta)) + \log(2\sqrt{n}/\delta)}{n}, \tag{121}$$

where $l(\beta)$ is the generalisation error/loss of the model $\beta$. Thus, when $\lambda = 1$, the right hand side of (121) reduces to $-2 \cdot \text{ELBO}(q)$ plus some complexity term, justifying why maximizing ELBO with respect to $q$ can be helpful for reducing the generalisation error. Although this argument may look partially sufficient, but strictly saying, the extra factor 2 in the ELBO (for the choice $\lambda = 1$) may be problematic, potentially making the bound trivial and less useful. Other choice of $\lambda$ fails to recover the original ELBO with slightly deviated coefficients for the expected loss and the KL.

In what follows, we state our new generalisation error bound for our FL algorithm, which does not rely on the PAC-Bayes but the recent regression analysis technique for variational Bayes (Pati et al., 2018; Bai et al., 2020). It was also adopted in the analysis of some personalised FL algorithm (Zhang et al., 2022) recently.

### E.1  GENERALISATION ERROR BOUND VIA REGRESSION ANALYSIS TECHNIQUE

We begin with the regression-based data modeling perspective and related assumptions/notations. We denote by $P^i(x, y)$ the *true* data distribution for client $i$ ($i = 1, \ldots, N$). We assume that the target $y$ is real vector-valued ($y \in \mathbb{R}^{S_y}$), and there exists a *true regression function* $f^i : \mathbb{R}^{S_x} \to \mathbb{R}^{S_y}$ for each

$i$. That is,

$$P^i(y|x) = \mathcal{N}(y; f^i(x), \sigma_\epsilon^2 I), \tag{122}$$

where $\sigma_\epsilon^2$ is constant Gaussian output noise variance. Let $D_i = (X^i, Y^i) \sim P^i(x, y)$ be the i.i.d. training data of size $n$ for each client $i$.

In our FL model, we assume that our backbone network is an MLP with $L$ hidden layers of width $M$, and all activation functions $\sigma(\cdot)$ are Lipschitz continuous with constant 1. The parameters $\theta$ of the MLP are also assumed to be bounded, more formally, the parameter space $\Theta$ is defined as:

$$\theta \in \Theta = \{\theta \in \mathbb{R}^G : ||\theta||_\infty \le B, \text{MLP with } L \text{ layers of width } M\}. \tag{123}$$

Note that $G = \dim(\theta)$ and $B$ is the maximal norm bound. The MLP defines a regression function $f_\theta : \mathbb{R}^{S_x} \to \mathbb{R}^{S_y}$, and the $\theta$-induced predictive distribution is denoted as:

$$P_\theta(y|x) = \mathcal{N}(y; f_\theta(x), \sigma_\epsilon^2 I), \tag{124}$$

where we assume that the true noise variance is known.

For notational convenience, we denote by $f(X^i)$ the concatenated vector of $f(x)$ for all $x \in X^i$, i.e., $f(X^i) = [f(x)]_{x \in X^i}$, where $f(\cdot)$ is either the true $f^i(\cdot)$ or the model $f_\theta(\cdot)$. Extending this notation, simply writing $f^i$ or $f_\theta$ means infinite-dimensional (population) responses, that is, $f^i = [f^i(x)]_{x \in \mathbb{R}^{S_x}}$ and $f_\theta$ similarly. For instance, $||f_\theta - f^i||_\infty$ stands for the worst-case difference, namely $\max_{x \in \mathbb{R}^{S_x}} ||f_\theta(x) - f^i(x)||$. As a generalisation error measure, we consider the *expected squared Hellinger distance* between the true $P^i$ and the model $P_\theta$. Formally,

$$d^2(P_\theta, P^i) = \mathbb{E}_{x \sim P^i(x)}\big[H^2(P_\theta(y|x), P^i(y|x))\big] = \mathbb{E}_{x \sim P^i(x)}\left[1 - \exp\left(-\frac{||f_\theta(x) - f^i(x)||_2^2}{8\sigma_\epsilon^2}\right)\right]. \tag{125}$$

More specifically, we will bound the posterior-averaged distance $\frac{1}{N}\sum_{i=1}^N \mathbb{E}_{q_i^*(\theta_i)}[d^2(P_{\theta_i}, P^i)]$, where $\{q_i^*(\theta_i)\}_{i=1}^N$ is an optimal solution of our FL-ELBO optimisation problem[10] (We showed in Sec. D that our block-coordinate FL algorithm converges to this optimal solution in $O(1/\sqrt{t})$ rate).

**Theorem E.1** (Generalisation error analysis). *Assume that the variational density family for $q_i(\theta_i)$ is rich enough to subsume Gaussian. The optimal solution $(\{q_i^*(\theta_i)\}_{i=1}^N, q^*(\phi))$ of our FL-ELBO optimisation problem (3) satisfies (with high probability):*

$$\frac{1}{N}\sum_{i=1}^N \mathbb{E}_{q_i^*(\theta_i)}[d^2(P_{\theta_i}, P^i)] \le O\left(\frac{1}{n}\right) + C \cdot \epsilon_n^2 + C'\left(r_n + \frac{1}{N}\sum_{i=1}^N \lambda_i^*\right), \tag{126}$$

*where $C, C' > 0$ are constant, $\lambda_i^* = \min_{\theta \in \Theta} ||f_\theta - f^i||_\infty^2$ is the best error within our backbone $\Theta$,*

$$r_n = \frac{G(L+1)}{n}\log M + \frac{G}{n}\log\left(S_x\sqrt{\frac{n}{G}}\right), \tag{127}$$

*and $\epsilon_n = \sqrt{r_n}\log^\delta(n)$ for $\delta > 1$ constant.*

*Remark.* Theorem E.1 implies that the optimal solution for our FL-ELBO optimisation problem (attainable by our block-coordinate FL algorithm) is *asymptotically optimal*, since the right hand side of (126) converges to 0 as the training data size $n \to \infty$. This is easy to verify: as $n \to \infty$, $r_n \to 0$ obviously, accordingly $\epsilon_n \to 0$, and the last term $\frac{1}{N}\sum_i \lambda_i^*$ can be made arbitrarily close to 0 by increasing the backbone capacity (MLPs as universal function approximators). But practically for fixed $n$, as enlarging the backbone capacity (i.e., large $G$, $L$, and $M$) also increases $\epsilon_n$ and $r_n$, it is important to choose the backbone network architecture properly. Note also that our assumption on the variational density family for $q_i(\theta_i)$ is easily met; for instance, the families of the mixtures of Gaussians adopted in NIW (Sec. 3.1) and mixture models (Sec. 3.2) obviously subsume a single Gaussian family.

*Proof of Theorem E.1.* We first aim to link the variational ELBO objective function to the Hellinger distance via Donsker-Varadhan's (DV) theorem (Boucheron et al., 2013), motivated from (Bai et al., 2020; Zhang et al., 2022). The DV theorem allows us to express the expectation of any exponential function variationally using the KL divergence. More specifically, the following holds for any distributions $p, q$ and any (bounded) function $h(z)$:

$$\log \mathbb{E}_{p(z)}[e^{h(z)}] = \max_q \big(\mathbb{E}_{q(z)}[h(z)] - \text{KL}(q||p)\big). \tag{128}$$

---

[10]Note that the optimal posterior on $\phi$ (i.e., $q^*(\phi)$) does not appear here since it only affects $d^2(P_\theta, P^i)$ implicitly. See our detailed analysis/proof provided below.

Here we set $p(z) := p(\theta_i|\phi)$, $q(z) := q_i(\theta_i)$, $h(z) := \log \eta_i(\theta_i)$, where

$$\eta_i(\theta_i) := \exp\left(l_n(P_{\theta_i}(D_i), P^i(D_i)) + nd^2(P_{\theta_i}, P^i)\right) \text{ and} \tag{129}$$

$$l_n(P_{\theta_i}(D_i), P^i(D_i)) := \log \frac{P_{\theta_i}(D_i)}{P^i(D_i)}, \tag{130}$$

and we have the following inequality that holds for any $\phi$:

$$\log \mathbb{E}_{p(\theta_i|\phi)}[\eta_i(\theta_i)] \geq \mathbb{E}_{q_i(\theta_i)}[\log \eta_i(\theta_i)] - \mathrm{KL}(q_i(\theta_i)||p(\theta_i|\phi)). \tag{131}$$

By taking the expectation with respect to $q(\phi)$ and rearranging terms, we have

$$n \cdot \mathbb{E}_{q_i(\theta_i)}[d^2(P_{\theta_i}, P^i)] \leq \mathbb{E}_{q_i(\theta_i)}[-l_n(P_{\theta_i}(D_i), P^i(D_i))] + \mathbb{E}_{q(\phi)}[\mathrm{KL}(q_i(\theta_i)||p(\theta_i|\phi))] +$$
$$\mathbb{E}_{q(\phi)}\left[\log \mathbb{E}_{p(\theta_i|\phi)}[\eta_i(\theta_i)]\right]. \tag{132}$$

For the last term of the right hand side, we use the bound $\mathbb{E}_{s(\theta)}[\eta(\theta)] \leq e^{Cn\epsilon_n^2}$ from the regression theorem (Pati et al., 2018), which holds for any distribution $s(\theta)$ with high probability. The details and proof of this bound can be found in the proof of Theorem 3.1 in (Pati et al., 2018). Applying this bound and telescoping over $i = 1, \ldots, N$ yields:

$$n \cdot \sum_{i=1}^{N} \mathbb{E}_{q_i(\theta_i)}[d^2(P_{\theta_i}, P^i)] \leq \sum_{i=1}^{N} \mathbb{E}_{q_i(\theta_i)}[-l_n(P_{\theta_i}(D_i), P^i(D_i))] +$$
$$\sum_{i=1}^{N} \mathbb{E}_{q(\phi)}[\mathrm{KL}(q_i(\theta_i)||p(\theta_i|\phi))] + NCn\epsilon_n^2. \tag{133}$$

We add $\mathrm{KL}(q(\phi)||p(\phi))$ to the right hand side, which retains the inequality since KL divergence is nonnegative. Then we have the following result that holds for *any q* with high probability,

$$\frac{n}{N} \cdot \sum_{i=1}^{N} \mathbb{E}_{q_i(\theta_i)}[d^2(P_{\theta_i}, P^i)] \leq L_n(q(\phi), \{q_i(\theta_i)\}_{i=1}^{N}) + Cn\epsilon_n^2, \tag{134}$$

where $L_n(q(\phi), \{q_i(\theta_i)\}_{i=1}^{N})$ equals:

$$\frac{1}{N}\left\{\sum_{i=1}^{N}\left(\mathbb{E}_{q_i(\theta_i)}[-l_n(P_{\theta_i}(D_i), P^i(D_i))] + \mathbb{E}_{q(\phi)}[\mathrm{KL}(q_i(\theta_i)||p(\theta_i|\phi))]\right) + \mathrm{KL}(q(\phi)||p(\phi))\right\}, \tag{135}$$

which exactly coincides with our FL-ELBO objective (3) up to constant, by the factor of $1/N$.

Next, we define $\tilde{q}_i(\theta_i)$ and $\tilde{q}(\phi)$ as follows:

$$\tilde{q}_i(\theta_i) = \mathcal{N}(\theta_i; \theta_i^*, \sigma_n^2 I) \text{ with } \theta_i^* = \arg\min_{\theta \in \Theta} ||f_\theta - f^i||_\infty^2, \ \sigma_n^2 = \frac{G}{8n}A, \text{ where} \tag{136}$$

$$A^{-1} = \log(3S_x M) \cdot (2BM)^{2(L+1)} \cdot \left(\left(S_x + 1 + \frac{1}{BM-1}\right)^2 + \frac{1}{(2BM)^2-1} + \frac{2}{(2BM-1)^2}\right),$$

$$\tilde{q}(\phi) = \arg\min_{q(\phi)} \sum_{i=1}^{N} \mathbb{E}_{q(\phi)}[\mathrm{KL}(\tilde{q}_i(\theta_i)||p(\theta_i|\phi))] + \mathrm{KL}(q(\phi)||p(\phi)). \tag{137}$$

Since $(\{q_i^*(\theta_i)\}_{i=1}^{N}, q^*(\phi))$ is the optimal solution of the FL-ELBO optimisation problem, it is obvious that $L_n(q^*(\phi), \{q_i^*(\theta_i)\}_{i=1}^{N}) \leq L_n(\tilde{q}(\phi), \{\tilde{q}_i(\theta_i)\}_{i=1}^{N})$ if the variational density family for $q_i(\theta_i)$ is rich enough to subsume Gaussian. Now we look closely at $L_n(\tilde{q}(\phi), \{\tilde{q}_i(\theta_i)\}_{i=1}^{N})$, and we note that the last two terms as per (135) are constant (i.e., not a function of data size $n$). That is,

$$\frac{1}{N}\left\{\sum_{i=1}^{N} \mathbb{E}_{\tilde{q}(\phi)}[\mathrm{KL}(\tilde{q}_i(\theta_i)||p(\theta_i|\phi))] + \mathrm{KL}(\tilde{q}(\phi)||p(\phi))\right\} = \tilde{C}, \tag{138}$$

for some constant $\tilde{C}$. Then we can write

$$L_n(\tilde{q}(\phi), \{\tilde{q}_i(\theta_i)\}_{i=1}^{N}) = \frac{1}{N}\sum_{i=1}^{N} \mathbb{E}_{\tilde{q}_i(\theta_i)}[-l_n(P_{\theta_i}(D_i), P^i(D_i))] + \tilde{C}. \tag{139}$$

We bound the expected $-l_n$ term in (139) using use Lemma E.2 below[11], which states that with high probability,

$$\mathbb{E}_{\tilde{q}_i(\theta_i)}[-l_n(P_{\theta_i}(D_i), P^i(D_i))] \leq C'n(r_n + \lambda_i^*), \tag{140}$$

---

[11]Although Lemma E.2 can be found in the proof of Lemma 4.1 of (Bai et al., 2020), we state this lemma more clearly with separate proof for self-containment.

for some constant $C' > 0$. Plugging this bound into (139), we have the following derivation where we start from (134) with $q(\phi) = q^*(\phi)$ and $q_i(\theta_i) = q_i^*(\theta_i)$:

$$\frac{n}{N} \cdot \sum_{i=1}^{N} \mathbb{E}_{q_i^*(\theta_i)}[d^2(P_{\theta_i}, P^i)] \leq L_n(q^*(\phi), \{q_i^*(\theta_i)\}_{i=1}^N) + Cn\epsilon_n^2 \tag{141}$$

$$\leq L_n(\tilde{q}(\phi), \{\tilde{q}_i(\theta_i)\}_{i=1}^N) + Cn\epsilon_n^2 \tag{142}$$

$$\leq \tilde{C} + C'n\left(r_n + \frac{1}{N}\sum_{i=1}^{N}\lambda_i^*\right) + Cn\epsilon_n^2. \tag{143}$$

By dividing both sides by $n$, we complete the proof. $\qquad\square$

**Lemma E.2** (From the proof of Lemma 4.1 in (Bai et al., 2020)). *For $\tilde{q}_i(\theta_i)$ defined[12] as in (136) and $r_n$, $\lambda_i^*$ defined as in Theorem E.1, the inequality (140) holds with high probability.*

*Proof of Lemma E.2.* From our regression model assumption (122) and (124),

$$\mathbb{E}_{\tilde{q}_i(\theta_i)}\big[-l_n(P_{\theta_i}(D_i), P^i(D_i))\big] = \mathbb{E}_{\tilde{q}_i(\theta_i)}\big[\log P^i(D_i) - \log P_{\theta_i}(D_i)\big] \tag{144}$$

$$= \frac{1}{2\sigma_\epsilon^2}\Big(\mathbb{E}_{\tilde{q}_i(\theta_i)}||Y^i - f_{\theta_i}(X^i)||_2^2 - \mathbb{E}_{\tilde{q}_i(\theta_i)}||Y^i - f^i(X^i)||_2^2\Big) \tag{145}$$

$$= \frac{1}{2\sigma_\epsilon^2}\Big(\underbrace{\mathbb{E}_{\tilde{q}_i(\theta_i)}||f_{\theta_i}(X^i) - f^i(X^i)||_2^2}_{\triangleq R_1} + 2 \cdot \underbrace{\mathbb{E}_{\tilde{q}_i(\theta_i)}\langle Y^i - f^i(X^i), f^i(X^i) - f_{\theta_i}(X^i)\rangle}_{\triangleq R_2}\Big). \tag{146}$$

We first work on $R_1$. Since

$$||f_{\theta_i}(X^i) - f^i(X^i)||_2^2 \leq n \cdot ||f_{\theta_i} - f^i||_\infty^2 \leq 2n\Big(||f_{\theta_i} - f_{\theta_i^*}||_\infty^2 + ||f_{\theta_i^*} - f^i||_\infty^2\Big), \tag{147}$$

we have

$$R_1 = 2n \cdot \mathbb{E}_{\tilde{q}_i(\theta_i)}||f_{\theta_i} - f_{\theta_i^*}||_\infty^2 + 2n \cdot ||f_{\theta_i^*} - f^i||_\infty^2 \tag{148}$$

$$\leq 2n(r_n + \lambda_i^*). \tag{149}$$

where in (149), we use the definition of $\lambda_i^*$ and the fact $\mathbb{E}_{\tilde{q}_i(\theta_i)}||f_{\theta_i} - f_{\theta_i^*}||_\infty^2 \leq r_n$ from Appendix G in (Chérief-Abdellatif, 2020).

Next, we bound $R_2$. Since $(Y^i - f^i(X^i)) \sim \mathcal{N}(0, \sigma_\epsilon^2 I)$ and independent of $\theta_i$, we can let $\epsilon := Y^i - f^i(X^i)$ for $\epsilon \sim \mathcal{N}(0, \sigma_\epsilon^2 I)$. Then

$$R_2 = \epsilon^\top \cdot \mathbb{E}_{\tilde{q}_i(\theta_i)}\big[f^i(X^i) - f_{\theta_i}(X^i)\big] \sim \mathcal{N}(0, c_f\sigma_\epsilon^2), \tag{150}$$

where $c_f = \big\|\mathbb{E}_{\tilde{q}_i(\theta_i)}\big[f^i(X^i) - f_{\theta_i}(X^i)\big]\big\|_2^2$. Applying Jensen's inequality on the convexity of $||\cdot||_2^2$, $c_f \leq \mathbb{E}_{\tilde{q}_i(\theta_i)}||f^i(X^i) - f_{\theta_i}(X^i)||_2^2 = R_1$. Due to the property of Gaussian, there exists some constant $C_0'$ such that $R_2 \leq C_0' \cdot c_f \leq C_0' \cdot R_1$ with high probability. Plugging these bounds on $R_1$ and $R_2$ back to (146) leads to:

$$\mathbb{E}_{\tilde{q}_i(\theta_i)}\big[-l_n(P_{\theta_i}(D_i), P^i(D_i))\big] = \frac{1}{2\sigma_\epsilon^2}(R_1 + 2R_2) \leq \frac{1 + 2C_0'}{2\sigma_\epsilon^2}R_1 \tag{151}$$

$$\leq \frac{1 + 2C_0'}{\sigma_\epsilon^2}n(r_n + \lambda_i^*). \tag{152}$$

Letting $C' := \frac{1+2C_0'}{\sigma_\epsilon^2}$ (constant) completes the proof. $\qquad\square$

## F  ADDITIONAL EXPERIMENTAL RESULTS

### F.1  MORE RESULTS ON CIFAR-100 & CIFAR-C-100

We test our mixture model with different mixture orders $K = 2, 5, 10$ on CIFAR-100 (Table 3) and CIFAR-C-100 (Table 4). In the last columns of the tables, we also report the performance of the centralised (non-FL) training, in which the batch sampling follows the corresponding FL settings.

---

[12]In (Bai et al., 2020), they defined $\tilde{q}_i(\theta_i)$ as a spike-and-slab model to deal with sparsity. Essentially it is a mixture of two components, selecting $\mathcal{N}(\theta_i; \theta_i^*, \sigma_n^2 I)$ when $\theta_i^*$ entries are non-zero and selecting the delta function at 0 when $\theta_i^*$ entries equal zero. Without loss of generality (or under mild numerical approximation), we can assume all entries of $\theta_i^*$ are non-zero, which makes $\tilde{q}_i(\theta_i)$ Gaussian equal to $\mathcal{N}(\theta_i; \theta_i^*, \sigma_n^2 I)$ as in (136).

Table 3: (CIFAR-100) Global prediction and personalisation accuracy. Mixture order $K$ varied. Comparison with centralised (non-FL) training.

(a) Global prediction performance (initial accuracy)

| FL settings | | | NIW (Ours) | Mixture (Ours) | | | Fed-BABU | Centralised |
|---|---|---|---|---|---|---|---|---|
| $s$ | $f$ | $\tau$ | | $K=2$ | $K=5$ | $K=10$ | | |
| 100 | 0.1 | 1 | $49.76^{\pm0.12}$ | $49.37^{\pm0.30}$ | $46.97^{\pm0.13}$ | $48.35^{\pm0.07}$ | $42.35^{\pm0.42}$ | $52.21^{\pm0.19}$ |
| | | 10 | $29.02^{\pm0.33}$ | $29.02^{\pm0.29}$ | $29.08^{\pm0.63}$ | $29.90^{\pm0.25}$ | $27.93^{\pm0.28}$ | $36.87^{\pm1.72}$ |
| | 1.0 | 1 | $57.80^{\pm0.10}$ | $52.94^{\pm0.36}$ | $52.24^{\pm0.20}$ | $51.92^{\pm0.04}$ | $48.17^{\pm0.56}$ | $58.50^{\pm0.52}$ |
| | | 10 | $29.53^{\pm0.42}$ | $30.55^{\pm0.15}$ | $30.85^{\pm0.13}$ | $30.24^{\pm0.49}$ | $28.67^{\pm0.51}$ | $48.16^{\pm0.58}$ |
| 10 | 0.1 | 1 | $37.54^{\pm0.25}$ | $38.07^{\pm0.40}$ | $39.96^{\pm0.63}$ | $39.97^{\pm0.52}$ | $35.04^{\pm0.56}$ | $28.55^{\pm1.05}$ |
| | | 10 | $18.99^{\pm0.03}$ | $18.95^{\pm0.13}$ | $18.93^{\pm0.37}$ | $18.95^{\pm0.03}$ | $18.54^{\pm0.37}$ | $9.92^{\pm0.33}$ |
| | 1.0 | 1 | $50.40^{\pm0.11}$ | $49.52^{\pm0.88}$ | $49.82^{\pm0.43}$ | $49.51^{\pm0.36}$ | $45.41^{\pm0.11}$ | $78.37^{\pm0.88}$ |
| | | 10 | $22.87^{\pm0.41}$ | $23.59^{\pm0.47}$ | $23.69^{\pm0.76}$ | $24.28^{\pm1.04}$ | $21.92^{\pm0.66}$ | $4.40^{\pm0.21}$ |

(b) Personalisation performance

| FL settings | | | NIW (Ours) | Mixture (Ours) | | | Fed-BABU | Centralised |
|---|---|---|---|---|---|---|---|---|
| $s$ | $f$ | $\tau$ | | $K=2$ | $K=5$ | $K=10$ | | |
| 100 | 0.1 | 1 | $54.16^{\pm0.50}$ | $56.17^{\pm0.16}$ | $54.93^{\pm0.25}$ | $55.83^{\pm0.47}$ | $50.43^{\pm0.93}$ | $53.18^{\pm0.10}$ |
| | | 10 | $36.68^{\pm0.37}$ | $36.32^{\pm0.27}$ | $37.30^{\pm0.64}$ | $37.34^{\pm0.38}$ | $35.45^{\pm0.34}$ | $38.20^{\pm1.58}$ |
| | 1.0 | 1 | $60.36^{\pm0.89}$ | $58.82^{\pm0.37}$ | $58.16^{\pm0.26}$ | $58.32^{\pm0.34}$ | $55.87^{\pm0.91}$ | $58.49^{\pm0.50}$ |
| | | 10 | $35.92^{\pm0.17}$ | $36.22^{\pm0.17}$ | $36.44^{\pm0.15}$ | $35.91^{\pm0.15}$ | $35.58^{\pm0.24}$ | $48.17^{\pm0.59}$ |
| 10 | 0.1 | 1 | $79.41^{\pm0.24}$ | $79.70^{\pm0.19}$ | $79.29^{\pm0.19}$ | $77.44^{\pm0.54}$ | $75.44^{\pm0.36}$ | $42.04^{\pm1.38}$ |
| | | 10 | $67.35^{\pm1.02}$ | $67.57^{\pm0.62}$ | $67.84^{\pm0.40}$ | $67.33^{\pm0.26}$ | $66.24^{\pm0.53}$ | $17.40^{\pm1.05}$ |
| | 1.0 | 1 | $82.71^{\pm0.37}$ | $81.03^{\pm0.35}$ | $79.91^{\pm0.25}$ | $80.92^{\pm0.19}$ | $78.92^{\pm0.23}$ | $78.43^{\pm0.90}$ |
| | | 10 | $67.78^{\pm1.02}$ | $66.74^{\pm0.27}$ | $66.50^{\pm0.24}$ | $67.30^{\pm0.29}$ | $66.25^{\pm0.46}$ | $5.13^{\pm0.19}$ |

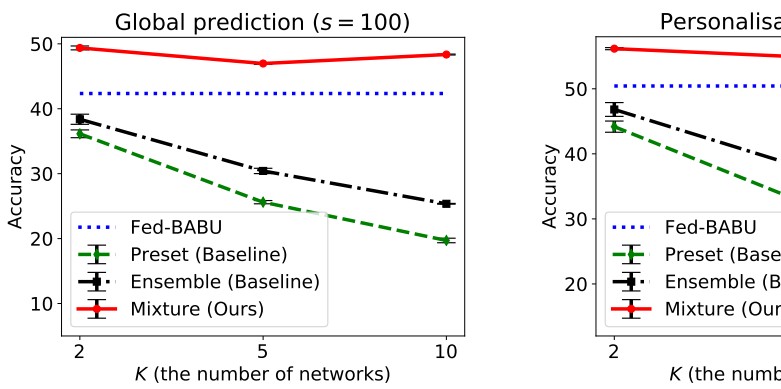
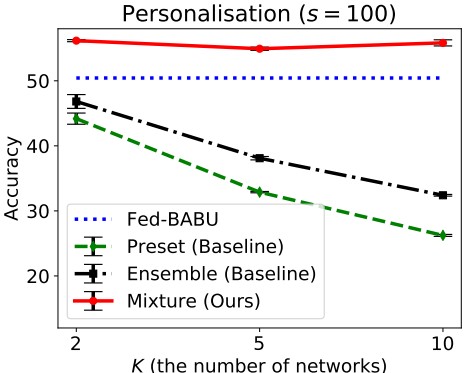

Figure 3: Comparison between our mixture model and ensemble baselines ($K$ varied) on CIFAR-100.

That is, at each round, the minibatches for SGD (for conventional cross-entropy loss minimisation) are sampled from the data of the participating clients. The centralised training sometimes outperforms the best FL algorithms (our models), but can fail completely especially when data heterogeneity is high (small $s$) and $\tau$ is large. This may be due to overtraining on biased client data for relatively few rounds. Our FL models perform well consistently and stably being comparable to centralised training on its ideal settings (small $\tau$ and/or large $s$).

### F.2 COMPARISON WITH SIMPLE ENSEMBLE BASELINES.

Our mixture model maintains $K$ backbone networks (specifically $\{r_j\}_{j=1}^{K}$) where the mixture order $K$ is usually small but greater than 1 (e.g., $K=2$). Thus it requires extra computational resources than other methods (including our NIW model) that only deal with a single backbone. As a baseline comparison, we aim to come up with some simple extension of Fed-Avg (McMahan et al., 2017) that incorporates multiple (the same $K$) networks. Here are the detailed descriptions of the baseline ensemble extensions:

Table 4: (CIFAR-C-100) Global prediction and personalisation accuracy. Mixture order $K$ varied. Comparison with centralised (non-FL) training.

(a) Global prediction (initial accuracy) on test splits for the 10 training corruption types

| FL settings | | | NIW (Ours) | Mixture (Ours) | | | Fed-BABU | Centralised |
|---|---|---|---|---|---|---|---|---|
| $s$ | $f$ | $\tau$ | | $K=2$ | $K=5$ | $K=10$ | | |
| 100 | 0.1 | 1 | $81.22^{\pm0.14}$ | $80.34^{\pm1.44}$ | $81.30^{\pm0.21}$ | $81.23^{\pm0.79}$ | $79.45^{\pm0.71}$ | $87.68^{\pm0.48}$ |
| | | 4 | $67.69^{\pm0.74}$ | $65.81^{\pm0.84}$ | $64.37^{\pm0.99}$ | $66.72^{\pm1.46}$ | $63.58^{\pm1.28}$ | $76.82^{\pm1.17}$ |
| | 1.0 | 1 | $91.26^{\pm0.83}$ | $86.84^{\pm0.22}$ | $86.41^{\pm0.58}$ | $87.04^{\pm0.33}$ | $86.84^{\pm0.83}$ | $90.22^{\pm0.57}$ |
| | | 4 | $73.58^{\pm1.02}$ | $74.55^{\pm0.22}$ | $74.53^{\pm0.81}$ | $75.03^{\pm0.45}$ | $71.03^{\pm0.75}$ | $87.71^{\pm0.91}$ |
| 50 | 0.1 | 1 | $78.63^{\pm0.39}$ | $79.36^{\pm0.24}$ | $78.23^{\pm0.33}$ | $78.45^{\pm0.40}$ | $77.44^{\pm1.17}$ | $84.97^{\pm0.22}$ |
| | | 4 | $65.08^{\pm1.75}$ | $63.52^{\pm0.48}$ | $63.08^{\pm0.55}$ | $63.13^{\pm0.21}$ | $62.65^{\pm0.12}$ | $70.76^{\pm2.93}$ |
| | 1.0 | 1 | $89.31^{\pm0.17}$ | $88.24^{\pm0.71}$ | $87.81^{\pm0.66}$ | $87.02^{\pm0.19}$ | $86.44^{\pm0.97}$ | $89.92^{\pm0.47}$ |
| | | 4 | $70.33^{\pm0.18}$ | $70.19^{\pm1.41}$ | $71.87^{\pm2.31}$ | $74.18^{\pm0.46}$ | $67.66^{\pm0.78}$ | $86.40^{\pm1.90}$ |

(b) Global prediction (initial accuracy) on the original (uncorrupted) CIFAR-100 training sets

| FL settings | | | NIW (Ours) | Mixture (Ours) | | | Fed-BABU | Centralised |
|---|---|---|---|---|---|---|---|---|
| $s$ | $f$ | $\tau$ | | $K=2$ | $K=5$ | $K=10$ | | |
| 100 | 0.1 | 1 | $41.55^{\pm0.11}$ | $36.99^{\pm0.09}$ | $37.75^{\pm0.50}$ | $37.78^{\pm0.44}$ | $34.76^{\pm0.50}$ | $35.10^{\pm0.65}$ |
| | | 4 | $30.84^{\pm0.07}$ | $30.60^{\pm0.27}$ | $30.51^{\pm0.20}$ | $29.13^{\pm0.15}$ | $28.31^{\pm0.28}$ | $33.64^{\pm1.09}$ |
| | 1.0 | 1 | $41.32^{\pm0.32}$ | $38.35^{\pm0.73}$ | $38.85^{\pm0.96}$ | $38.97^{\pm0.26}$ | $35.58^{\pm0.41}$ | $32.78^{\pm0.22}$ |
| | | 4 | $30.67^{\pm0.12}$ | $30.40^{\pm0.44}$ | $30.57^{\pm0.28}$ | $30.95^{\pm0.28}$ | $28.60^{\pm0.28}$ | $28.95^{\pm2.09}$ |
| 50 | 0.1 | 1 | $41.04^{\pm0.14}$ | $36.41^{\pm0.47}$ | $36.03^{\pm0.24}$ | $38.32^{\pm0.07}$ | $35.44^{\pm0.58}$ | $35.71^{\pm0.13}$ |
| | | 4 | $32.29^{\pm0.36}$ | $31.50^{\pm0.34}$ | $31.19^{\pm0.26}$ | $31.06^{\pm0.58}$ | $29.68^{\pm0.08}$ | $35.28^{\pm1.47}$ |
| | 1.0 | 1 | $41.64^{\pm0.21}$ | $38.54^{\pm0.42}$ | $39.19^{\pm0.37}$ | $38.93^{\pm0.15}$ | $36.09^{\pm0.28}$ | $33.49^{\pm0.17}$ |
| | | 4 | $32.17^{\pm0.48}$ | $30.68^{\pm0.46}$ | $31.29^{\pm0.65}$ | $32.46^{\pm0.20}$ | $29.28^{\pm0.17}$ | $29.50^{\pm1.24}$ |

(c) Personalisation performance on the 9 held-out corruption types

| FL settings | | | NIW (Ours) | Mixture (Ours) | | | Fed-BABU | Centralised |
|---|---|---|---|---|---|---|---|---|
| $s$ | $f$ | $\tau$ | | $K=2$ | $K=5$ | $K=10$ | | |
| 100 | 0.1 | 1 | $72.63^{\pm2.13}$ | $74.16^{\pm3.04}$ | $74.43^{\pm4.94}$ | $75.65^{\pm4.68}$ | $69.93^{\pm1.24}$ | $88.76^{\pm0.31}$ |
| | | 4 | $62.74^{\pm0.94}$ | $61.56^{\pm1.69}$ | $60.46^{\pm0.13}$ | $64.41^{\pm2.66}$ | $60.33^{\pm2.12}$ | $80.47^{\pm0.99}$ |
| | 1.0 | 1 | $83.62^{\pm1.84}$ | $84.88^{\pm0.85}$ | $83.01^{\pm1.79}$ | $82.58^{\pm0.91}$ | $77.55^{\pm2.05}$ | $87.55^{\pm2.93}$ |
| | | 4 | $64.84^{\pm1.05}$ | $67.35^{\pm1.46}$ | $64.34^{\pm0.75}$ | $65.12^{\pm0.77}$ | $53.25^{\pm2.19}$ | $76.58^{\pm12.63}$ |
| 50 | 0.1 | 1 | $75.50^{\pm0.74}$ | $67.33^{\pm2.83}$ | $67.32^{\pm1.35}$ | $65.34^{\pm3.19}$ | $59.47^{\pm2.48}$ | $84.08^{\pm1.03}$ |
| | | 4 | $44.90^{\pm1.23}$ | $46.39^{\pm0.83}$ | $45.95^{\pm0.37}$ | $45.43^{\pm0.34}$ | $44.74^{\pm1.30}$ | $74.14^{\pm2.51}$ |
| | 1.0 | 1 | $81.46^{\pm0.67}$ | $81.77^{\pm3.11}$ | $78.03^{\pm1.05}$ | $74.62^{\pm1.95}$ | $67.43^{\pm2.58}$ | $87.57^{\pm1.62}$ |
| | | 4 | $50.84^{\pm0.74}$ | $48.90^{\pm0.51}$ | $45.07^{\pm1.15}$ | $48.78^{\pm0.49}$ | $40.87^{\pm3.01}$ | $81.46^{\pm1.11}$ |

1. The server maintains $K$ networks (denoted by $\theta^1, \ldots, \theta^K$).

2. We partition the clients into $K$ groups with equal proportions. We will assign each $\theta^j$ to each group $j$ ($j = 1, \ldots, K$).

3. At each round, each participating client $i$ receives the current model $\theta^{j(i)}$ from the server, where $j(i)$ means the group index to which client $i$ belongs.

4. The clients perform local updates as usual by warm-start with the received models, and send the updated models back to the server.

5. The server collects updated local models from the clients, and takes the average within each group $j$ to update $\theta^j$.

6. After training, we have trained $K$ networks. At test time, we can use these $K$ networks in two different ways/options: (*Preset* option) Each client $i$ uses the network assigned to its group, i.e., $\theta^{j(i)}$, for both prediction and finetuning/personalisation; (*Ensemble* option) We use all $K$ networks (as an ensemble) for prediction and finetuning.

Table 5: Running times (in seconds) on CIFAR-100 with ($s = 100, f = 1.0, \tau = 1$) setting.

|  |  | Fed-BABU | NIW | Mix. ($K = 2$) | Mix. ($K = 5$) |
|---|---|---|---|---|---|
| Train | Client | 0.283 | 0.362 | 0.481 | 0.595 |
|  | Server | 0.027 | 0.267 | 1.283 | 1.434 |
| Global prediction |  | 0.021 | 0.025 | 0.040 | 0.060 |
| Personalisation |  | 1.581 | 2.158 | 2.421 | 2.766 |

Note that $K = 1$ exactly reduces to Fed-Avg (or Fed-BABU). In Fig. 3 we visualise the performance of these ensemble baselines, compared with our mixture model for different $K = 2, 5, 10$ on CIFAR-100 with ($f = 0.1, \tau = 1$) setting. It clearly shows that these simple ensemble strategies are prone to overfit. The result signifies the importance of the sophisticated negative `log-sum-exp` regularisation in the client/server updates as in (22) and (23) in our mixture model.

### F.3 RUNNING TIMES

Although our models achieve significant improvement in prediction accuracy, we have extra computational overhead compared to simpler FL methods like Fed-BABU. To see if this extra cost is allowable, we measure/compare wall clock times in Table 5, where all methods are tested on the same machine, Xeon 2.20GHz CPU with a single RTX 2080 Ti GPU. For NIW, the extra cost in the local client update and personalisation (training) originates from the penalty term in (17), while model weight squaring to compute $V_0$ in (18) incurs additional cost in server update. For Mixture, the increased time in training is mainly due to the overhead of computing distances from the $K$ server models in (22) and (23). However, overall the extra costs are not prohibitively large, rendering our methods sufficiently practical.

## G (REVISION) ALGORITHMS/PSEUDOCODES AND COMPUTATIONAL COMPLEXITY ANALYSIS

### G.1 ALGORITHMS/PSEUDOCODES

We provide pseudocodes for our algorithms:

1. **Training algorithms**
   - General framework in Alg. 2.
   - Normal-Inverse-Wishart case in Alg. 3.
   - Mixture case in Alg. 4.

2. **Global prediction algorithms**
   - General framework in Alg. 5.
   - Normal-Inverse-Wishart case in Alg. 6.
   - Mixture case in Alg. 7.

3. **Personalisation algorithms**
   - General framework in Alg. 8.
   - Normal-Inverse-Wishart case in Alg. 9.
   - Mixture case in Alg. 10.

### G.2 COMPUTATIONAL COMPLEXITY ANALYSIS

Based on the detailed algorithms in Sec. G.1, we can easily analyse the computational complexity of the proposed algorithms. They are summarised in Table 6 (training complexity with communication costs), Table 7 (global prediction complexity), and Table 8 (personalisation complexity).

---

**Algorithm 2** Training algorithm: **General framework**.

---

**Input:** Initial parameters $L_0$ in the variational posterior $q(\phi; L_0)$.
**Output:** Trained parameters $L_0$.
**For** each round $r = 1, 2, \ldots, R$ **do**:
   1. Sample a subset $\mathcal{I}$ of participating clients ($|\mathcal{I}| = N_f \leq N$).
   2. Server sends $L_0$ to all clients $i \in \mathcal{I}$.
   3. **For each client $i \in \mathcal{I}$ in parallel do**:
      Solve (by SGD) with $L_0$ fixed:
$$\min_{L_i} \ \mathbb{E}_{q_i(\theta_i; L_i)}[-\log p(D_i|\theta_i)] + \mathbb{E}_{q(\phi; L_0)}\big[\mathrm{KL}(q_i(\theta_i; L_i)||p(\theta_i|\phi))\big],$$

      Initial $L_i$ can be either copied from $L_0$ or the last iterate if the client is able to save $L_i$ locally.
   4. Each client $i \in \mathcal{I}$ sends the updated $L_i$ back to the server.
   5. Upon receiving $\{L_i\}_{\in \mathcal{I}}$, the server updates $L_0$ by solving (with $\{L_i\}_{\in \mathcal{I}}$ fixed):
$$\min_{L_0} \ \mathrm{KL}(q(\phi; L_0)||p(\phi)) - \frac{N}{N_f}\sum_{i \in \mathcal{I}} \mathbb{E}_{q(\phi; L_0)q_i(\theta_i; L_i)}[\log p(\theta_i|\phi)].$$

---

**Algorithm 3** Training algorithm: **Normal-Inverse-Wishart case**.

---

**Input:** Initial $L_0 = (m_0, V_0)$ in $q(\phi; L_0) = \mathcal{NIW}(\phi; \{m_0, V_0, l_0 = |D| + 1, n_0 = |D| + d + 2\})$ where
   $|D| = \sum_{i=1}^{N} |D_i|$ and $d = $ the number of parameters in the backbone network $p(y|x, \theta)$.
**Output:** Trained parameters $L_0 = (m_0, V_0)$.
**For** each round $r = 1, 2, \ldots, R$ **do**:
   1. Sample a subset $\mathcal{I}$ of participating clients ($|\mathcal{I}| = N_f \leq N$).
   2. Server sends $L_0 = (m_0, V_0)$ to all clients $i \in \mathcal{I}$.
   3. **For each client $i \in \mathcal{I}$ in parallel do**:
      Solve (by SGD) with $L_0 = (m_0, V_0)$ fixed:
$$\min_{m_i} \ -\log p(D_i|\tilde{m}_i) + \frac{p}{2}(n_0 + d + 1)(m_i - m_0)^{\top}V_0^{-1}(m_i - m_0),$$

      where $\tilde{m}_i$ is the dropout version (with probability $1 - p$) of $m_i$.
      Initial $m_i$ can be either copied from $m_0$ or the last iterate if the client is able to save $m_i$ locally.
   4. Each client $i \in \mathcal{I}$ sends the updated $L_i = m_i$ back to the server.
   5. Upon receiving $\{m_i\}_{\in \mathcal{I}}$, the server updates $L_0 = (m_0, V_0)$ by:
$$m_0^* = \frac{p}{N+1}\frac{N}{N_f}\sum_{i \in \mathcal{I}} m_i, \ \ V_0^* = \frac{n_0}{N+d+2}\left((1 + N\epsilon^2)I + m_0^*(m_0^*)^{\top} + \frac{N}{N_f}\sum_{i \in \mathcal{I}}\rho(m_0^*, m_i, p)\right),$$
   where $\rho(m_0, m_i, p) = pm_i m_i^{\top} - pm_0 m_i^{\top} - pm_i m_0^{\top} + m_0 m_0^{\top}$.

---

**Algorithm 4** Training algorithm: **Mixture case**.

---

**Input:** Initial $L_0 = \{r_j\}_{j=1}^{K}$ in $q(\phi; L_0) = \prod_j \mathcal{N}(\mu_j; r_j, \epsilon^2 I)$ and $\beta$ in the gating network $g(x; \beta)$.
**Output:** Trained parameters $L_0 = \{r_j\}_{j=1}^{K}$ and $\beta$.
**For** each round $r = 1, 2, \ldots, R$ **do**:
   1. Sample a subset $\mathcal{I}$ of participating clients ($|\mathcal{I}| = N_f \leq N$).
   2. Server sends $L_0 = \{r_j\}_{j=1}^{K}$ and $\beta$ to all clients $i \in \mathcal{I}$.
   3. **For each client $i \in \mathcal{I}$ in parallel do**:
      Solve (by SGD) with $L_0 = \{r_j\}_{j=1}^{K}$ fixed:
$$\min_{m_i} \ \mathbb{E}_{q_i(\theta_i; m_i)}[-\log p(D_i|\theta_i)] - \log \sum_{j=1}^{K}\exp\left(-\frac{||m_i - r_j||^2}{2\sigma^2}\right), \ \text{where } q_i(\theta_i; m_i) = \mathcal{N}(\theta_i; m_i, \epsilon^2 I).$$

      Initial $m_i$ can be either the center of $\{r_j\}_{j=1}^{K}$ or the last iterate if the client is able to save $m_i$ locally.

      $\beta_i = $ SGD update of $\beta$ in $g(x; \beta)$ with data $\{(x, j^*)\}_{x \sim D_i}$ where $j^* = \arg\min_j ||m_i - r_j||$.
   4. Each client $i \in \mathcal{I}$ sends the updated $L_i = m_i$ and $\beta_i$ back to the server.
   5. Upon receiving $\{m_i\}_{\in \mathcal{I}}$ and $\{\beta_i\}_{\in \mathcal{I}}$, the server updates $L_0 = \{r_j\}_{j=1}^{K}$ by the one-step EM:
$$\text{(E-step)} \ c(j|i) = \frac{e^{-||m_i - r_j||^2/(2\sigma^2)}}{\sum_{j=1}^{K} e^{-||m_i - r_j||^2/(2\sigma^2)}}, \ \ \text{(M-step)} \ r_j^* = \frac{\frac{1}{N_f}\sum_{i \in \mathcal{I}} c(j|i) \cdot m_i}{\frac{\sigma^2}{N} + \frac{1}{N_f}\sum_{i \in \mathcal{I}} c(j|i)},$$
   and updates $\beta$ by aggregation: $\beta^* = \frac{1}{N_f}\sum_{i \in \mathcal{I}} \beta_i$.

---

---

**Algorithm 5** Global prediction: **General framework**.

---

**Input:** Test input $x^*$. Learned model $L_0$ in the variational posterior $q(\phi; L_0)$.
**Output:** Predictive distribution $p(y^*|x^*, D_{1:N})$.
1. Sample $\theta^{(s)} \sim \int p(\theta|\phi)\, q(\phi; L_0)\, d\phi$ for $s = 1, \ldots, S$.
2. Return $p(y^*|x^*, D_{1:N}) \approx \frac{1}{S} \sum_{s=1}^{S} p(y^*|x^*, \theta^{(s)})$.

---

**Algorithm 6** Global prediction: **Normal-Inverse-Wishart case**.

---

**Input:** Test input $x^*$. Learned model $L_0 = (m_0, V_0)$ in $q(\phi; L_0) = \mathcal{NIW}(\phi; \{m_0, V_0, l_0, n_0\})$.
**Output:** Predictive distribution $p(y^*|x^*, D_{1:N})$.
1. Sample $\theta^{(s)} \sim t_{n_0 - d + 1}\left(\theta; m_0, \frac{(l_0+1)V_0}{l_0(n_0-d+1)}\right)$ for $s = 1, \ldots, S$.
2. Return $p(y^*|x^*, D_{1:N}) \approx \frac{1}{S} \sum_{s=1}^{S} p(y^*|x^*, \theta^{(s)})$.

---

**Algorithm 7** Global prediction: **Mixture case**.

---

**Input:** Test input $x^*$. Learned model $L_0 = \{r_j\}_{j=1}^{K}$ in $q(\phi; L_0) = \prod_j \mathcal{N}(\mu_j; r_j, \epsilon^2 I)$ and $\beta$ in $g(x; \beta)$.
**Output:** Predictive distribution $p(y^*|x^*, D_{1:N})$.
1. Return $p(y^*|x^*, D_{1:N}) \approx \sum_{j=1}^{K} g_j(x^*) \cdot p(y^*|x^*, r_j)$.

---

**Algorithm 8** Personalisation: **General framework**.

---

**Input:** Personal training data $D^p$. Test input $x^p$. Learned model $L_0$ in the variational posterior $q(\phi; L_0)$.
**Output:** Predictive distribution $p(y^p|x^p, D^p, D_{1:N})$.
1. Estimate the variational density $v(\theta) \approx p(\theta|D^p, \phi^*)$ by solving (via SGD):
$$\min_v \ \mathbb{E}_{v(\theta)}[-\log p(D^p|\theta)] + \text{KL}(v(\theta)||p(\theta|\phi^*)), \ \text{where } \phi^* = \arg\max_\phi q(\phi; L_0).$$
2. Sample $\theta^{(s)} \sim v(\theta)$ for $s = 1, \ldots, S$.
3. Return $p(y^p|x^p, D^p, D_{1:N}) \approx \frac{1}{S} \sum_{s=1}^{S} p(y^p|x^p, \theta^{(s)})$.

---

**Algorithm 9** Personalisation: **Normal-Inverse-Wishart case**.

---

**Input:** Personal training data $D^p$. Test input $x^p$.
       Learned model $L_0 = (m_0, V_0)$ in $q(\phi; L_0) = \mathcal{NIW}(\phi; \{m_0, V_0, l_0, n_0\})$.
**Output:** Predictive distribution $p(y^p|x^p, D^p, D_{1:N})$.
1. Estimate $m$ in $v(\theta; m) = \prod_l \left( p \cdot \mathcal{N}(\theta[l]; m[l], \epsilon^2 I) + (1-p) \cdot \mathcal{N}(\theta[l]; 0, \epsilon^2 I) \right)$ by solving (via SGD):
$$\min_m \ -\log p(D^p|\tilde{m}) + \frac{p}{2}(n_0 + d + 1)(m - m_0)^\top V_0^{-1}(m - m_0),$$
where $\tilde{m}$ is the dropout version (with probability $1-p$) of $m$.
2. Return $p(y^p|x^p, D^p, D_{1:N}) \approx p(y^p|x^p, m)$.

---

**Algorithm 10** Personalisation: **Mixture case**.

---

**Input:** Personal training data $D^p$. Test input $x^p$.
       Learned model $L_0 = \{r_j\}_{j=1}^{K}$ in $q(\phi; L_0) = \prod_j \mathcal{N}(\mu_j; r_j, \epsilon^2 I)$.
**Output:** Predictive distribution $p(y^p|x^p, D^p, D_{1:N})$.
1. Estimate $m$ in $v(\theta; m) = \mathcal{N}(\theta; m, \epsilon^2 I)$ by solving (via SGD):
$$\min_m \ \mathbb{E}_{v(\theta;m)}[-\log p(D^p|\theta)] - \log \sum_{j=1}^{K} \exp\left(-\frac{||m - r_j||^2}{2\sigma^2}\right).$$
2. Return $p(y^p|x^p, D^p, D_{1:N}) \approx p(y^p|x^p, m)$.

---

# H (REVISION): ADDITIONAL EXPERIMENTS

## H.1 CIFAR-100 WITH THE EXTRA FOUR FL METHODS

The results are summarized in Table 9 where we use the following hyperparameters for the competing methods. FedPA (Al-Shedivat et al., 2021): shrinkage parameter $\rho = 0.01$; FedBE (Chen & Chao,

Table 6: **Training complexity** of the proposed algorithms (NIW and Mixture) and Fed-Avg. All quantities are per-round, per-batch, and per-client costs. In the entries, $d$ = the number of parameters in the backbone network, $F$ = time for feed-forward pass, $B$ = time for backprop, and $N_f$ = the number of participating clients per round.

| | Communication cost | | Client Local update | Server update |
|---|---|---|---|---|
| | Server $\rightarrow$ Client | Client $\rightarrow$ Server | | |
| NIW | $2d$ (sent: $m_0, V_0$) | $d$ (sent: $m_i$) | $F + B + O(d)$ ($O(d)$ from quadratic penalty) | $O(N_f \cdot d)$ |
| Mixture (Order $K$) | $(K+1)d$ (sent: $\{r_j\}_{j=1}^K, \beta$) | $2d$ (sent: $m_i, \beta_i$) | $2(F + B) + O(K \cdot d)$ ($O(K \cdot d)$ from log-sum-exp) | $O(K \cdot N_f \cdot d)$ |
| Fed-Avg | $d$ (sent: $\theta$) | $d$ (sent: $\theta_i$) | $F + B$ | $O(N_f \cdot d)$ (aggregation) |

Table 7: **Global prediction complexity** of the proposed algorithms (NIW and Mixture) and Fed-Avg. All quantities are per-test-batch costs. In the entries, $d$ = the number of parameters in the backbone network, $F$ = time for feed-forward pass, and $S$ = the number of samples $\theta^{(s)}$ from the Student-$t$ distribution in the NIW case (we use $S = 1$).

| | Per-test-batch complexity |
|---|---|
| NIW | $S \cdot F + O(S \cdot d)$ ($O(S \cdot d)$ from the cost of $t$-sampling) |
| Mixture (Order $K$) | $(K+1)F$ (a forward pass for the gating network) |
| Fed-Avg | $F$ |

Table 8: **Personalisation complexity** of the proposed algorithms (NIW and Mixture) and Fed-Avg. All quantities are per-train/test-batch costs. In the entries, $d$ = the number of parameters in the backbone network, $F$ = time for feed-forward pass, and $B$ = time for backprop.

| | Training (personalisation) complexity | Test complexity |
|---|---|---|
| NIW | $F + B + O(d)$ ($O(d)$ from quadratic penalty) | $F$ |
| Mixture (Order $K$) | $F + B + O(K \cdot d)$ ($O(K \cdot d)$ from log-sum-exp) | $F$ |
| Fed-Avg | $F + B$ | $F$ |

2021): the number of ensemble components 3; FedEM (Marfoq et al., 2021): the number of base models 3.

## H.2 MNIST AND FASHION-MNIST

The FL setting is as follows: the number of clients $N = 100$, the number of shards per client $s = 5$, the fraction of participating clients per round $f = 0.1$, and the number of local training epochs per round $\tau = 1$ (total number of rounds 100) or $5$ (total number of rounds 20). FedBE and FedEM use three component models. The backbone is an MLP with a single hidden layer with 256 units. The results are summarized in Table 10 (MNIST) and Table 11 (Fashion-MNIST).

## H.3 EMNIST

The FL setting is as follows: the number of clients $N = 200$, the fraction of participating clients per round $f = 0.2$, and the number of local training epochs per round $\tau = 1$ (total number of rounds 300). We follow the standard Dirichlet-based client data splitting. FedBE and FedEM use three

component models. The backbone is a standard ConvNet with two hidden layers. The results are summarized in Table 12.

## I  (REVISION): EXTENDED RELATED WORK

**General FL approaches.** Perhaps the seminal pioneering work on FL is attributed to FedAvg (McMahan et al., 2017), which proposed fairly intuitive local training and global aggregation strategies with minimal training and communication complexity. A potential issue of divergence of global and local models due to the separated steps of local training and aggregation was addressed by model regularisation in the follow-up works (Li et al., 2018; Acar et al., 2021), which is shown to help the global model converge more reliably. Recent approaches aimed to incorporate benefits from existing machine learning approaches including domain adaptation/generalisation, clustering, multi-task learning, transfer learning, and meta-learning. To deal with heterogeneous client data distributions, those works in (Peterson et al., 2019; Zhang et al., 2021; Sun et al., 2021) attempted to tackle the FL problem in the perspective of (multi-)Domain Adaptation/Generalisation. Another interesting line of works aims to cluster clients with similar data distributions together (Briggs et al., 2020; Mansour et al., 2020). Along the line, the shared representations among the related or similar clients can be modeled motivated from general multi-task learning (Smith et al., 2017; Dinh et al., 2021). Motivated from transfer learning, reasonable attempts are made to exploit the idea of learning/transferring knowledge from related clients (Chen et al., 2020; Yang et al., 2020; Dinh et al., 2020; Li et al., 2021), Last but not least, there have been attempts to the personalised FL methods based on meta learning since the fientuning from the global trained model can be seen as adaptation to new data (Chen et al., 2018; Fallah et al., 2020).

**Comparison to existing Bayesian FL approaches.** Some recent studies tried to address the FL problem using Bayesian methods. As we mentioned earlier, the key difference is that these methods do not introduce Bayesian *hierarchy*, and ultimately treat network weights $\theta$ as a random variable *shared* across all clients, while our approach assigns individual $\theta_i$ to each client $i$ governed by a common prior $p(\theta_i|\phi)$. The non-hierarchical approaches must all resort to ad hoc heuristics or strong assumptions in parts of their algorithm. More specifically, **FedPA** (Posterior Averaging) (Al-Shedivat et al., 2021) aims to establish the decomposition, $p(\theta|D_{1:N}) \propto \prod_{i=1}^{N} p(\theta|D_i)$ also known as *product of experts*, to allow client-wise inference/optimisation of $p(\theta|D_i)$. Unfortunately this decomposition does not hold in general unless we make a strong assumption of uninformative prior $p(\theta) \propto 1$ as they did. **FedBE** (Bayesian Ensemble) (Chen & Chao, 2021) aims to build the global posterior distribution $p(\theta|D_{1:N})$ from the individual posteriors $p(\theta|D_i)$ in either of two ad-hoc ways: SWAG (Maddox et al., 2019)-like model averaging over clients, or a convex combination of the modes of the local posteriors. **pFedBayes** (Zhang et al., 2022) can be seen as an implicit regularisation-based method to approximate $p(\theta|D_{1:N})$ from individual posteriors $p(\theta|D_i)$. To combine the individual posteriors, they introduce the so-called global distribution $w(\theta)$, which essentially serves as a *regulariser* that aims to enforce local posteriors $p(\theta|D_i)$ not to deviate from it, i.e., $p(\theta|D_i) \approx w(\theta)$ for all $i$. The introduction of $w(\theta)$ and its update strategy appears to be a hybrid treatment rather than solely Bayesian perspective. Finally, **FedEM** (Marfoq et al., 2021) forms a seemingly reasonable hypothesis that local client data distributions can be identified as mixtures of a fixed number of base distributions (with different mixing proportions). However, although they have probabilistic modeling, mixture estimation, and base distribution learning under this hypothesis, this method is not a Bayesian approach.

**FedGP** (Achituve et al., 2021b) aims to extend the GP-Tree algorithm (Achituve et al., 2021a) to the FL setting via the shared deep kernel learning. To this end, the clients perform GP-Tree kernel learning locally on its own data while the server aggregation simply follows the FedAvg algorithm to learn a global kernel. In this sense, the overall approach is quite different from our hierarchical Bayesian treatment. **FedPop** (Kotelevskii et al., 2022): It has a similar hierarchical Bayesian model structure as ours. But they split the backbone network parameters into those of the feature extractor (denoted by $\phi$ in the paper) and the linear classification head ($\beta$). In their model, the feature extractor weights $\phi$ are shared across the clients (called *fixed effects*), and the client-wise classification head parameters $z^i$ are sampled from $\beta$, i.e., $z^i \sim p(z|\beta)$. Thus the client data $D_i$ is generated by $\phi$ and $z^i$. The main differences from our approach are in four folds: 1) The higher-level variables $\beta$ and local variables $z^i$ sampled from $\beta$ are both restricted to the linear classification head part of the network, which makes imposing uncertainty in model parameters quite limited; 2) Moreover, they do not

Table 9: (CIFAR-100) Global prediction and personalisation accuracy.

(a) Global prediction performance (initial accuracy)

| FL settings | | | Our Methods | | Fed-BABU | Fed-Avg | Fed-Prox | pFedBayes | FedPA | FedBE | FedEM | FedPop |
|---|---|---|---|---|---|---|---|---|---|---|---|---|
| $s$ | $f$ | $\tau$ | NIW | Mix. ($K=2$) | | | | | | | | |
| 100 | 0.1 | 1 | $\mathbf{49.76}^{\pm\mathbf{0.12}}$ | $49.37^{\pm0.30}$ | $42.35^{\pm0.42}$ | $40.87^{\pm0.62}$ | $41.49^{\pm0.75}$ | $37.23^{\pm0.88}$ | $42.15^{\pm0.78}$ | $42.49^{\pm0.89}$ | $43.83^{\pm1.07}$ | $43.09^{\pm0.30}$ |
| | | 10 | $\mathbf{29.02}^{\pm\mathbf{0.33}}$ | $\mathbf{29.02}^{\pm\mathbf{0.29}}$ | $27.93^{\pm0.28}$ | $28.26^{\pm0.19}$ | $27.11^{\pm0.11}$ | $28.21^{\pm1.42}$ | $28.05^{\pm0.28}$ | $28.39^{\pm0.45}$ | $28.62^{\pm0.26}$ | $28.31^{\pm0.42}$ |
| | 1.0 | 1 | $\mathbf{57.80}^{\pm\mathbf{0.10}}$ | $52.94^{\pm0.36}$ | $48.17^{\pm0.56}$ | $47.44^{\pm0.20}$ | $47.66^{\pm1.49}$ | $44.89^{\pm0.32}$ | $47.96^{\pm0.17}$ | $48.69^{\pm0.70}$ | $50.28^{\pm0.72}$ | $48.36^{\pm0.44}$ |
| | | 10 | $29.53^{\pm0.42}$ | $\mathbf{30.55}^{\pm\mathbf{0.15}}$ | $28.67^{\pm0.51}$ | $28.79^{\pm0.68}$ | $27.43^{\pm0.38}$ | $28.25^{\pm0.81}$ | $28.89^{\pm0.38}$ | $28.60^{\pm1.18}$ | $29.51^{\pm0.12}$ | $28.99^{\pm0.47}$ |
| 10 | 0.1 | 1 | $37.54^{\pm0.25}$ | $\mathbf{38.07}^{\pm\mathbf{0.40}}$ | $35.04^{\pm0.56}$ | $27.48^{\pm0.86}$ | $34.73^{\pm0.21}$ | $31.49^{\pm0.18}$ | $35.51^{\pm0.55}$ | $35.17^{\pm0.40}$ | $37.28^{\pm0.26}$ | $35.01^{\pm0.58}$ |
| | | 10 | $\mathbf{18.99}^{\pm\mathbf{0.03}}$ | $18.95^{\pm0.13}$ | $18.54^{\pm0.37}$ | $14.69^{\pm0.40}$ | $16.84^{\pm0.48}$ | $17.93^{\pm0.68}$ | $18.59^{\pm0.19}$ | $18.67^{\pm0.09}$ | $18.45^{\pm0.10}$ | $18.68^{\pm0.12}$ |
| | 1.0 | 1 | $\mathbf{50.40}^{\pm\mathbf{0.11}}$ | $49.52^{\pm0.88}$ | $45.41^{\pm0.11}$ | $37.10^{\pm0.44}$ | $44.33^{\pm0.31}$ | $39.95^{\pm0.89}$ | $45.08^{\pm0.72}$ | $45.56^{\pm0.52}$ | $47.52^{\pm0.59}$ | $44.98^{\pm0.27}$ |
| | | 10 | $22.87^{\pm0.41}$ | $\mathbf{23.59}^{\pm\mathbf{0.47}}$ | $21.92^{\pm0.66}$ | $17.38^{\pm0.32}$ | $19.54^{\pm0.38}$ | $21.85^{\pm0.50}$ | $22.60^{\pm0.29}$ | $21.73^{\pm0.64}$ | $22.51^{\pm0.06}$ | $22.06^{\pm0.40}$ |

(b) Personalisation performance

| FL settings | | | Our Methods | | Fed-BABU | Fed-Avg | Fed-Prox | pFedBayes | FedPA | FedBE | FedEM | FedPop |
|---|---|---|---|---|---|---|---|---|---|---|---|---|
| $s$ | $f$ | $\tau$ | NIW | Mix. ($K=2$) | | | | | | | | |
| 100 | 0.1 | 1 | $54.16^{\pm0.50}$ | $\mathbf{56.17}^{\pm\mathbf{0.16}}$ | $50.43^{\pm0.93}$ | $46.43^{\pm0.82}$ | $49.91^{\pm0.78}$ | $45.83^{\pm1.12}$ | $49.88^{\pm0.49}$ | $50.57^{\pm1.03}$ | $47.28^{\pm0.88}$ | $51.22^{\pm0.37}$ |
| | | 10 | $\mathbf{36.68}^{\pm\mathbf{0.37}}$ | $36.32^{\pm0.27}$ | $35.45^{\pm0.34}$ | $33.57^{\pm0.06}$ | $33.92^{\pm0.22}$ | $35.74^{\pm1.36}$ | $35.06^{\pm0.30}$ | $35.51^{\pm0.62}$ | $34.41^{\pm1.13}$ | $35.69^{\pm0.47}$ |
| | 1.0 | 1 | $\mathbf{60.36}^{\pm\mathbf{0.89}}$ | $58.82^{\pm0.37}$ | $55.87^{\pm0.91}$ | $53.15^{\pm0.25}$ | $55.50^{\pm0.90}$ | $53.00^{\pm0.48}$ | $55.43^{\pm0.31}$ | $56.25^{\pm0.42}$ | $54.65^{\pm0.49}$ | $55.85^{\pm0.51}$ |
| | | 10 | $35.92^{\pm0.17}$ | $\mathbf{36.22}^{\pm\mathbf{0.17}}$ | $35.58^{\pm0.24}$ | $33.82^{\pm1.04}$ | $33.70^{\pm0.42}$ | $35.57^{\pm1.02}$ | $35.21^{\pm0.50}$ | $34.92^{\pm0.94}$ | $35.21^{\pm0.36}$ | $35.36^{\pm0.36}$ |
| 10 | 0.1 | 1 | $79.41^{\pm0.24}$ | $\mathbf{79.70}^{\pm\mathbf{0.19}}$ | $75.44^{\pm0.36}$ | $70.36^{\pm1.02}$ | $75.06^{\pm0.67}$ | $73.93^{\pm0.14}$ | $75.76^{\pm0.36}$ | $76.19^{\pm0.42}$ | $75.52^{\pm0.50}$ | $74.97^{\pm0.72}$ |
| | | 10 | $67.35^{\pm1.02}$ | $\mathbf{67.57}^{\pm\mathbf{0.62}}$ | $66.24^{\pm0.53}$ | $61.39^{\pm0.27}$ | $64.86^{\pm0.73}$ | $65.82^{\pm0.33}$ | $65.87^{\pm0.21}$ | $66.64^{\pm0.25}$ | $67.11^{\pm0.11}$ | $66.70^{\pm0.55}$ |
| | 1.0 | 1 | $\mathbf{82.71}^{\pm\mathbf{0.37}}$ | $81.03^{\pm0.35}$ | $78.92^{\pm0.23}$ | $76.98^{\pm0.66}$ | $78.56^{\pm0.55}$ | $78.08^{\pm0.28}$ | $78.84^{\pm0.16}$ | $79.82^{\pm0.52}$ | $80.65^{\pm0.32}$ | $78.96^{\pm0.14}$ |
| | | 10 | $\mathbf{67.78}^{\pm\mathbf{1.02}}$ | $66.74^{\pm0.27}$ | $66.25^{\pm0.46}$ | $63.81^{\pm0.40}$ | $63.81^{\pm0.51}$ | $66.15^{\pm1.29}$ | $66.23^{\pm0.29}$ | $66.06^{\pm0.54}$ | $66.34^{\pm0.12}$ | $66.57^{\pm0.44}$ |

Table 10: (MNIST) Global prediction and personalisation accuracy.

(a) Global prediction performance (initial accuracy)

| FL settings | | | Our Methods | | Fed-BABU | Fed-Avg | Fed-Prox | pFedBayes | FedPA | FedBE | FedEM | FedPop |
|---|---|---|---|---|---|---|---|---|---|---|---|---|
| $s$ | $f$ | $\tau$ | NIW | Mix. ($K=2$) | | | | | | | | |
| 5 | 0.1 | 1 | 97.81 | **97.94** | 97.16 | 97.15 | 97.38 | 97.32 | 93.38 | 97.16 | 97.38 | 97.42 |
| | | 5 | 95.51 | **95.68** | 94.59 | 94.86 | 95.28 | 94.28 | 94.87 | 95.11 | 95.33 | 94.98 |

(b) Personalisation performance

| FL settings | | | Our Methods | | Fed-BABU | Fed-Avg | Fed-Prox | pFedBayes | FedPA | FedBE | FedEM | FedPop |
|---|---|---|---|---|---|---|---|---|---|---|---|---|
| $s$ | $f$ | $\tau$ | NIW | Mix. ($K=2$) | | | | | | | | |
| 5 | 0.1 | 1 | 98.78 | **98.85** | 97.73 | 97.83 | 97.96 | 97.88 | 96.66 | 97.62 | 97.89 | 97.91 |
| | | 5 | 96.53 | **96.67** | 95.89 | 96.10 | 96.45 | 95.49 | 95.93 | 96.37 | 96.46 | 95.80 |

Table 11: (Fashion-MNIST) Global prediction and personalisation accuracy.

(a) Global prediction performance (initial accuracy)

| FL settings | | | Our Methods | | Fed-BABU | Fed-Avg | Fed-Prox | pFedBayes | FedPA | FedBE | FedEM | FedPop |
|---|---|---|---|---|---|---|---|---|---|---|---|---|
| $s$ | $f$ | $\tau$ | NIW | Mix. ($K=2$) | | | | | | | | |
| 5 | 0.1 | 1 | 84.18 | **84.28** | 83.86 | 81.98 | 83.25 | 81.20 | 81.10 | 80.35 | 83.51 | 82.40 |
| | | 5 | 77.48 | **77.60** | 76.10 | 73.70 | 73.67 | 72.35 | 73.47 | 73.28 | 76.69 | 72.93 |

(b) Personalisation performance

| FL settings | | | Our Methods | | Fed-BABU | Fed-Avg | Fed-Prox | pFedBayes | FedPA | FedBE | FedEM | FedPop |
|---|---|---|---|---|---|---|---|---|---|---|---|---|
| $s$ | $f$ | $\tau$ | NIW | Mix. ($K=2$) | | | | | | | | |
| 5 | 0.1 | 1 | 92.48 | **92.54** | 91.03 | 90.59 | 90.72 | 90.31 | 91.60 | 89.96 | 92.10 | 91.14 |
| | | 5 | **89.91** | 89.53 | 89.02 | 86.85 | 87.35 | 86.38 | 88.76 | 85.00 | 89.65 | 86.40 |

Table 12: (EMNIST) Global prediction and personalisation accuracy.

(a) Global prediction performance (initial accuracy)

| FL setting | | Our Methods | | Fed-BABU | Fed-Avg | Fed-Prox | pFedBayes | FedPA | FedBE | FedEM | FedPop |
|---|---|---|---|---|---|---|---|---|---|---|---|
| $f$ | $\tau$ | NIW | Mix. ($K=2$) | | | | | | | | |
| 0.2 | 1 | 85.40 | **85.58** | 84.33 | 85.27 | 85.27 | 84.65 | 83.20 | 85.24 | 85.21 | 85.27 |

(b) Personalisation performance

| FL setting | | Our Methods | | Fed-BABU | Fed-Avg | Fed-Prox | pFedBayes | FedPA | FedBE | FedEM | FedPop |
|---|---|---|---|---|---|---|---|---|---|---|---|
| $f$ | $\tau$ | NIW | Mix. ($K=2$) | | | | | | | | |
| 0.2 | 1 | 88.84 | **88.97** | 83.09 | 87.92 | 88.39 | 88.12 | 85.10 | 88.37 | 88.32 | 88.40 |

actually treat $\beta$ (and $\phi$ of feature extractor) as random variables, but deterministic variables which are optimized in empirical Bayes learning. This hinders the model from benefiting from hierarchical Bayesian modeling (e.g., they do not have prior distribution $p(\beta)$ at all); 3) Their optimization is alternating between the feature extractor $\phi$ and the head prior parameters $\beta$, utterly different from our block coordinate optimization alternating between higher level random variables and individual local variables; 4) They did not use variational inference for inference $p(z^i | D_i, \phi, \beta)$, but MCMC sampling (Lagevin dynamics), which is the very reason why they had to reduce the size of the latents $z^i$ only limited to classification heads, instead of full network parameters as we did.

