# OpenReview forum: "A Hierarchical Bayesian Approach to Federated Learning"
_ICLR.cc/2023/Conference — Submitted to ICLR 2023_

### Official Review · Reviewer_SHVY · 2022-10-23

**Confidence:** 3
**Clarity, Quality, Novelty And Reproducibility:** The code to reproduce the numerical r…
**Correctness:** 4
**Technical Novelty And Significance:** 3
**Empirical Novelty And Significance:** 2
**Recommendation:** 6

**Strength And Weaknesses:**

Strength:
1.	The idea of employing Bayesian approach and variational inference techniques for federated learning is of novelty.
2.	Theoretical analysis shows the convergence of the proposed algorithm is guaranteed.
3.	The develop framework can tackle the problem of distribution disparity among the local learner.
Weaknesses:
There are some studies that deal with the heterogeneous distribution in domain generalization and adaption, which is not covered by the analysis of related works; the empirical studies also only compare the proposed algorithm with standard FL methods. The conclusions will be more convincing if more comparison is made with some approaches that deal with the heterogeneous distribution.


**Summary Of The Paper:**

This paper proposes a novel hierarchical Bayesian approach to federated learning, and derive ELBO objective function using variational inference techniques.  Then the block-coordinate descent algorithm is devised, which fit well in the federated learning regime. Theoretical result on the convergence of the algorithm is obtained. Empirical studies demonstrated the superior performance of the proposed method over existing ones.

**Summary Of The Review:**

This paper develops a novel federated learning method with hierarchical Bayesian approach, and it can cope with the heterogeneous local distributions. Sound theoretical results on the convergence and the generalization performance are attained. Empirical studies are relatively weak, because only simple dataset, cifar100, is used and the compared algorithms are some standard FL approaches.

---

> ### Author Response · Authors · 2022-11-18
> **Responses to Reviewer SHVY**
>
> **1. Missing related works on heterogeneous distribution in domain generalization/adaption.**
>
> We have included some related works in this regard in the revision. Please see extended related work section in the Appendix I.
>
>
> **2. More empirical studies on different FL approaches and datasets other than CIFAR.**
>
> As the reviewer suggested, we have conducted extra experiments. In addition to CIFAR-100 and larger-scale CIFAR-C (corrupted-CIFAR) in our original submission, we have tested our method on the popular FL Benchmarks: MNIST, Fashion-MNIST and EMNIST.
>
> Moreover, we have compared our approach with the four latest FL methods that aimed to deal with heterogeneous data distributions in probabilistic or Bayesian manners:
>
> 1) *FedPA* (Al-Shedivat et al., 2021), the maximum-a-posterior method based on the product-of-experts model,
>
> 2) *FedBE* (Chen & Chao, 2021), the Bayesian ensemble method where the ensemble components are sampled from a Gaussian global model that is estimated from local client models,
>
> 3) *FedEM* (Marfoq et al., 2021) which aims to identify local client data distributions as mixtures of a fixed number of base distributions with different mixing proportions, and
>
> 4) *FedPop* (Kotelevskii et al., 2022) is a hierarchical Bayesian model similar to ours, but with deterministic higher-level variates with stochastic modeling restricted only to the  classification head.
>
> Please see Table 9,10,11,12 in Appendix H of the revised paper. As shown our approach outperforms all these latest methods in both global prediction and personalisation, although the differences are rather minor for smaller datasets like MNIST, Fashion-MNIST, and EMNIST.

---

### Official Review · Reviewer_UL1w · 2022-10-23

**Confidence:** 3
**Correctness:** 4
**Technical Novelty And Significance:** 3
**Empirical Novelty And Significance:** 3
**Recommendation:** 6

**Clarity, Quality, Novelty And Reproducibility:**

The paper is well written. Although Bayesian hierarchical modeling is a standard probabilistic modeling for group/individual data, the application to Federated Learning is quite novel.

**Strength And Weaknesses:**

# Strength
1. The paper is well structured. The paper first introduced a general Bayesian FL framework and derived the ELBO objective optimization function with block-coordinate optimization solution. Then, the paper presents two concreate models (NIW and Gaussian Mixture), which illustrate the hierarchical Bayesian framework very well.
2. The paper presents principled hierarchical Bayesian approach, in which client’s individual parameters are governed by a common prior. Compare with other Bayesian FL approaches, the proposed method is fully Bayesian and do not need ad-hoc heuristics nor strong assumptions.
3. The effectiveness of the proposed Bayesian hierarchical FL is empirically tested on benchmark datasets (outperforms the best by large margins most of the time) in both global and personalization prediction.
4. The paper provides strong theoretical analysis of the convergence and the generalization error of the proposed block-coordinate FL algorithm.

# Weakness
1. The discussion with related work needs to include other Bayesian treatment for personalized FL, e.g. [1]
2. The empirical evaluation is only on two benchmark datasets, considering including more relevant datasets in [2]

References:
[1] Idan Achituve, Aviv Shamsian, Aviv Navon, Gal Chechik, and Ethan Fetaya. Personalized
federated learning with gaussian processes. In Advances in Neural Information Processing
Systems 34: Annual Conference on Neural Information Processing Systems, pages 8392–8406,
2021.
[2] Chen, D., Gao, D., Kuang, W., Li, Y., & Ding, B. (2022). pFL-Bench: A Comprehensive Benchmark for Personalized Federated Learning. arXiv preprint arXiv:2206.03655.


**Summary Of The Paper:**

This paper proposes a hierarchical Bayesian approach to Federated Learning. The proposed Bayesian model makes the block-coordinate descent solution becomes a distributed algorithm. In addition, the paper also derives convergence analysis that shows the block-coordinate FL algorithm converges to a local optimum of the objective at the rate of $O(1/\sqrt{t})$, which is the same as the regular SGD; and a generalization error analysis that shows the test error is asymptotically optimal. The effectiveness of this approach is demonstrated empirically on two benchmark datasets (CIFAR-100 and CIFAR-C-100). The proposed approach outperforms the best work in terms of both global prediction and personalization.

**Summary Of The Review:**

I would like to recommend the paper for acceptance based on the following reasons:
1) The paper is clearly written and well organized.
2) The proposed method by the paper is well motivated and technically correct.
3) The effectiveness of the paper is demonstrated both empirically and theoretically.

---

> ### Author Response · Authors · 2022-11-18
> **Responses to Reviewer UL1w**
>
> **1. Need discussions on other Bayesian personalized FL methods in related work, e.g. FedGP (Achituve et al., 2021b).**
>
> In addition to those Bayesian FL methods (FedPA, pFedBayes, FedBE, and FedEM) in our original submission, we take the reviewer's suggestion by  including two recent Bayesian FL works in the revision: FedGP (Achituve et al., 2021b) and FedPop (Kotelevskii et al., 2022). In particular, FedGP aims to extend the GP-Tree algorithm (Achituve et al., 2021a) to the FL setting via the shared deep kernel learning. To this end, the clients perform GP-Tree kernel learning locally on its own data while the server aggregation simply follows the FedAvg algorithm to learn a global kernel. In this sense, the overall approach is quite different from our hierarchical Bayesian treatment.
> Please see extended related work section in Appendix I.
>
>
> **2. Empirical evaluation on other benchmark datasets.**
>
> We have included extra experimental results on some  popular FL benchmark datasets including MNIST, Fashion-MNIST, and EMNIST. Please see Table 10,11,12 in Appendix H of the revised paper.

---

### Official Review · Reviewer_FDT8 · 2022-10-24

**Confidence:** 3
**Correctness:** 3
**Technical Novelty And Significance:** 2
**Empirical Novelty And Significance:** 2
**Recommendation:** 5

**Clarity, Quality, Novelty And Reproducibility:**

- The paper is in some parts hard to read. For example, in the section on global prediction and personalization,  the authors suggest to use a lot of approximations that they should be more discussed. The theoretical results are barely commented in the current version. Finally,  while I appreciated the illustration of the method on practical examples in Section 3, I think that the authors could only focus on a particular example and give all the details they postpone to the supplement. Indeed, without these complements, it was very difficult to me to catch the main ideas.

- Related work and the literature should be more discussed. From what I saw, the discussion is for the moment limited to Bayesian approaches.

- The use of hierarchical Bayesian model for FL has already been proposed in https://arxiv.org/abs/2206.03611
I wonder how the method of the paper under review compares with the one developed in this paper.

**Strength And Weaknesses:**

Strength:
- The paper tackles the important problem of personalization in federated learning using a hierarchical Bayesian approach.
- The approach using variational inference to infer local and global parameters is interesting.
- The paper provides theoretical guarantees on the convergence of the method.

Weakness:
- Practical implementation of the method could be better presented as well as its computational complexity.
Moreover, the paper would benefit from a better description of how the proposed solution addresses the specific problems of FL, in particular communication constrains and privacy. In the current version, it seems to me that it only focuses on personalization.
- The method does not address the problem of partial communication which is common in FL.

**Summary Of The Paper:**

This paper uses Bayesian paradigm for personalized Federated Learning.
Specifically, this paper suggests a two-level hierarchical model for data $D_{1:N}$ distributed on $N$ clients of the form:
$$
p(D_{1:N} | \theta_{1:N}) = \prod_{i=1}^N p(D_i | \theta_i) \quad , \quad p(\theta_{1:N} | \phi ) = \prod_{i=1}^N p(\theta_i[\phi) .
$$
Here, $\theta_{1:N}$ represent individual client parameters which are linked trough the globally shared variable  $\phi$.
Taking some prior for $\phi$, the posterior distribution is
$$
p(\phi, \theta_{1:N}| D_{1:N} ) \propto p(\phi) \prod_{i=1}^N p (\theta_{i} | \phi) p(D_i | \theta_i).
$$
This distribution is intractable and the authors use variational inference to approximate it by
$$
q(\phi,\theta_{1:N} | L_{0:N}) = q(\phi | L_0) \prod_{i=1}^N q(\theta_i|L_i),
$$
where $L_{0:N}$ is the parameters which are learnt maximizing the ELBO
$$
\mathcal{L} = \sum_{i=1}^N \mathcal{L}_i  + \mathcal{L}_0 ,
$$
where $\mathcal{L}_i$ depends on $L_i$ and $D_i$.

Therefore, $\mathcal{L}$ can be optimized using a block-coordinate optimization scheme. At each step, and given $L_0$, each client optimizes $\mathcal{L}_i$ and sends its update to a central server which in turn optimizes $\mathcal{L}_0$ to update the global parameter $\phi$ for fixed $L\_{1:N}$.

Based on their inference, the authors also suggest how to make personalization for an unseen new client.
In addition, they derive convergence bounds for their FL optimization algorithm and obtain generalization bounds for their variational approximations as the number of observations goes to $\infty$. Finally, numerical experiments compare the method proposed in the paper with some FL baselines.



**Summary Of The Review:**

The paper is interesting but should be more specific on how it addresses FL problems. In addition, comparisons with other personalization methods should be added.

---

> ### Author Response · Authors · 2022-11-18
> **Responses to Reviewer FDT8 (Part 2)**
>
> **8. The use of hierarchical Bayesian model for FL has already been proposed in https://arxiv.org/abs/2206.03611 I wonder how the method of the paper under review compares with the one developed in this paper.**
>
> The paper the reviewer refers to is FedPop (Kotelevskii et al., 2022). It has a similar hierarchical Bayesian model structure as ours. But they split the backbone network parameters into those of the feature extractor (denoted by $\phi$ in the paper) and the linear classification head ($\beta$). In their model, the feature extractor weights $\phi$ are shared across the clients (called *fixed effects*), and the  client-wise classification head parameters $z^i$ are sampled from $\beta$, i.e., $z^i\sim p(z|\beta)$.  Thus the client data $D_i$ is generated by $\phi$ and $z^i$. The main differences from our approach are four folds:
>
> 1) The higher-level variables $\beta$ and local variables $z^i$ sampled from $\beta$ are both restricted to the linear classification head part of the network, which makes imposing uncertainty in model parameters quite limited. Their hierarchical Bayesian treatment only covers the shallow classifier, while ours covers the full deep network.
>
> 2) Moreover, they do not actually treat $\beta$ (and $\phi$ of feature extractor) as random variables, but deterministic variables which are optimized in empirical Bayes learning. This hinders the model from benefiting from hierarchical Bayesian modeling (e.g., they do not have prior distribution $p(\beta)$ at all)
>
> 3) Their optimization alternates between the feature extractor $\phi$ and the head prior parameters $\beta$, utterly different from our block coordinate optimization alternating between higher level random variables and individual local variables
>
> 4) They did not use variational inference for inference $p(z^i|D_i,\phi,\beta)$, but MCMC sampling (Lagevin dynamics), which is the very reason why they had to reduce the size of the latents $z^i$ only limited to classification heads, instead of full network parameters as we did.

---

> ### Author Response · Authors · 2022-11-18
> **Responses to Reviewer FDT8 (Part 1)**
>
> **1. Practical implementation of the method could be better presented as well as its computational complexity. Moreover, the paper would benefit from a better description of how the proposed solution addresses the specific problems of FL, in particular communication constraints and privacy.**
>
> Considering the reviewer's concerns about the presentation of the proposed methods and removing any potential confusion, we have included detailed  algorithms/pseudocodes of our methods in Appendix G.1 in the revision. Therein, we provide separate detailed algorithms on training stage, global prediction, and personalisation, in each of three cases: general framework, NIW model, and mixture model cases. Please refer to Algorithm 2-10. Based on these detailed algorithms, we can easily analyse the computational complexity and communication costs of the proposed algorithms. They are summarised in Table 6,7,8 in Appendix G.2 (Table 6: training complexity with communication costs, Table 7: global prediction complexity, and Table 8: personalisation complexity.).
>
> In summary, we tackle communication constraints and privacy by developing a  hierarchical Bayesian model for which variational inference with block-coordinate descent naturally decomposes over the client and server variational posteriors (Sec 2.1). This means that data need not be exchanged (privacy is maintained) and communication is only needed at convergence of each block (communication cost is low).
>
>
> **2. In the current version, it seems to me that it only focuses on personalization.**
>
> We actually cover both global prediction and personalisation. In both the abstract case (Sec 2.2) and each of the two concrete model cases (Sec 3.1 and 3.2), we provided procedures for global prediction and personalisation. Please see those paragraphs with the titles/headings in boldface in Sec. 3.1/Sec. B.1 (NIW) and Sec. 3.2/Sec. C.1 (Mixture). We now further provide pseudocode in Algorithm 2-10 of Appendix G.1 in the revision that gives detailed algorithms for both tasks.
>
>
> **3. The method does not address the problem of partial communication which is common in FL.**
>
> We do address the partial communication/participation as we stated in the formal description of our block coordinate descent algorithm (Alg 1 in the Appendix), where $N_f$ ($\leq N$) is the number of participating clients for each round. To remove any confusion, we have also included more detailed training algorithms/pseudocodes in the revised version (Alg 2,3,4 in Appendix G), in which we sample a subset $\mathcal{I}$ of participating clients ($|\mathcal{I}|=N_f\leq N$) at each round.
>
>
> **4. In the section on global prediction and personalization, the authors suggest to use a lot of approximations that they should be more discussed.**
>
> We did discuss the meaning and quality of approximations every time they appear. For example, we stated that the approximation in Eq.(8) is variational inference approximation for the true posterior, and about the first approximation in Eq.(10) we said we disregard the impact of $D^p$ on the higher-level $\phi$ given the joint evidence, $p(\phi|D^p,D_{1:N})\approx p(\phi|D_{1:N})$ due to the dominance of $D_{1:N}$ compared to smaller $D^p$. We're happy to provide further clarification on any queries the reviewer has about specific approximations of interest.
>
>
> **5. The theoretical results are barely commented in the current version. The authors could only focus on a particular example and give all the details they postpone to the supplement.**
>
> It would be ideal to put more details in the main text, but due to the lack of space, we had to put only the main theoretical results in the main text. We expect that  interested readers can easily refer to the detailed materials in Appendix D and E since we clearly mentioned those pointers in the main text.
>
>
> **6. Related work and the literature should be more discussed beyond Bayesian approaches.**
>
> Thanks. We have included more prior works on FL (not limited to Bayesian approaches) in the revision. Please see extended related work section in Appendix I.
>
>
> **7. The paper is interesting but should be more specific on how it addresses FL problems. In addition, comparisons with other personalization methods should be added.**
>
> To clarify how our approach addresses FL problems specifically, we have added the detailed pseudocodes for training, global prediction, and personalisation in Alg 2-10 in Appendix G.1 of the revised paper. Comparison with some latest probabilistic/Bayesian personalised FL approaches (FedPA, FedBE, FedEM, and FedPop) have been added in the revised paper. Please see Table 9,10,11,12 in Appendix H of the revised paper.
>
>
> ---
> *(Continued in Part 2 below.)*

---

### Official Review · Reviewer_7vmX · 2022-10-24

**Confidence:** 3
**Correctness:** 3
**Technical Novelty And Significance:** 3
**Empirical Novelty And Significance:** 3
**Recommendation:** 5

**Clarity, Quality, Novelty And Reproducibility:**

At the current stage the paper is not clearly written and the quality cannot be easily judged. I believe there is some novelty in this work. About the reproducibility, the authors attached the code but it cannot be run. There is no package imported and the data collection process is not clear. Therefore, the reproducibility of this paper cannot be checked.

**Strength And Weaknesses:**

Strengths:
1) The Bayesian approach to FL is interesting and seems suitable for personalization. Using the hierachical approach and variational inference, the authors propose two different choices of hierachical models, that seem to be work well in practice.
2) The proposed Bayesian FL algorithms are supported by convergence analysis and generalization error bound.
3) The method is tested on CIFAR-100 and CIFAR-C-100 and compared with baseline algorithms, with noticeable improvement.

Weaknesses:
Although this hierachical Bayesian approach seems interesting, and it is supported by some theoretical analysis as well as experimental comparison, I feel this paper is currently not ready for publication due to several reasons:

1. The motivation is not very clear and some claims are not well-supported. For example:
 1) Sec 1, paragraph 1: "FedAvg and FedProx are well-known to suffer convergence issues". Any reference? What are the assumptions? Why would personalization resolve the convergence issues?
2) Sec 1, paragraph 2: why would Bayesian perspective be good for personalization? Is the current method complete or principled? Despite this being a bit subjective, I think at least the choices of NIW and Mix. are not principled and ad hoc.
3) What is the meaning of the higher level random variable $\phi$? Is it like some prior or shared knowledge? More intuitive explanation is needed.
4) Figure 1: what are $y^p$ and $x^p$?
5) Sec 2.1: reference of ELBO and the original ref of block coordinate optimization are needed.
6) Sec 2.2: the notation $p^p$ seems a bit weird as the two $p$'s have different meanings.
7) eq.10: what does the approximation sign mean? In which cases does the equality hold?
8) Sec 3: how the two models are proposed does not seem straightforward. Why are NIW and Mix. good choices and how are they proposed?
2. Convergence analysis. In Appendix D the authors added 3 assumptions. In Assumption 1, the definitions of "locally convex" are not quite clear. Assumptions 2 and 3 have some overlap regarding the Lipschitzness and the radius of the domain needs to be justified.
1) The convergence of the proposed methods has $O(1/\sqrt{T})$ convergence rate. This is at the same level as FedAvg (see e.g. [1]). Therefore the proposed method has no advantage over FedAvg which the authors claim to have some convergence issues.
3. The generalization error bound is not well explained. For example, what is $\epsilon_n$? What does $\lambda_i^*$ mean? Why do you use Hellinger distance? How is the error bound compared to (Mohri et al 2019)?
4. The empirical evalution can be improved. The dataset is mainly CIFAR-100 and its variant, and only 4 baselines are compared. FedAvg and FedProx are pretty similar and more Bayesian approaches should be compared, such as FedPA, FedBE, FedEM, and PredictiveBayes [2].

[1] Adaptive federated optimization. Reddi et al, ICLR 2021.
[2] Robust One Round Federated Learning with Predictive Space Bayesian Inference, Hasan et al, https://arxiv.org/pdf/2206.09526v1.pdf.

**Summary Of The Paper:**

This paper proposes a novel hierachical Bayesian approach to FL, with variational inference. The idea is to use two levels of random variables, the higher level $\phi$ which is shared among clients, and lower level $\theta_i$'s for each client $i$. This allows flexibility of conditional independence and personalization. There are two main methods: Normal-inverse-Wishart (NIW) and Mixture Model (Mix.). The methods are supported by convergence and generalization proofs, as well as experiments on two datasets.

--post rebuttal--
Thanks to the authors for the careful response to my questions. I have updated my score accordingly. After reading the response, I think the paper indeed improves, but there are still some weaknesses to address. It seems that on the theoretical side, the only main novelty is the generalization bound, as the authors admit. However, after checking the proof, I found this generalization bound shares a lot of similarities with Bai et al. 2020. Moreover, in the main paper, Theorem 4.2 needs a lot of discussions, such as the meaning of each term (e.g. $r_n$, $\epsilon_n$), the implication of this theorem, and comparison with existing results (e.g. Bai et al). On the algorithmic side, since hierarchical Bayesian is proposed before, what is the main novelty of this paper? Is it simply an application for personalized FL? Finally, on the experimental side, after adding more baselines, the improvement over baselines is not significant. For instance, for MNIST the improvement is less than 1%. Even for CIFAR-100, for $\tau = 10$ the improvement is much less than $\tau = 1$. This casts doubt on the practicability of the proposed algorithms.

**Summary Of The Review:**

In summary, this paper proposes hierachical Bayesian FL using variational inference which looks interesting. However, the paper largely suffers from presentation, and it is not ready for acceptance till it's more clearly presented.

---

> ### Author Response · Authors · 2022-11-18
> **Responses to Reviewer 7vmX (Part 2)**
>
> **6. Sec 3: how the two models are proposed does not seem straightforward. Why are NIW and Mix. good choices and how are they proposed?**
>
> Regarding the NIW model, the motivation is that NIW is a natural conjugate prior choice for the Gaussian likelihood model, as we also mentioned in the second paragraph of Appendix B. The choice of this conjugate prior choice enables particularly efficient closed-form learning and inference algorithms.
>
> Regarding the mixture model, we described in the beginning of Sec. 3.2 the main motivation: the mixture provides more flexibility with multiple different prototypes, diverse enough to cover the heterogeneity in data distributions across clients. The mixture model is ideal for task heterogeneity that occurs in clusters, for example federated image recognition among users with different intersts such as taking photos in day vs night, or of sports vs animals. Each cluster of users (day/night photographers, sport/animal enthusiasts) should be assigned to one mixture component.
>
>
> **7. Convergence analysis. In Appendix D the authors added three assumptions. In Assumption 1, the definitions of ``locally convex'' are not quite clear. Assumptions 2 and 3 have some overlap regarding the Lipschitzness and the radius of the domain needs to be justified.**
>
> Locally convex assumption means that the function has upward curvature in the vicinity of each point, which is common for convergence analysis of optimization algorithms. Lipschitzness is also a mild assumption imposing smoothness of the objective function; without this assumption theoretical property of an algorithm can be hardly analyzed. Lastly, the bounded input space and bounded gradients are also very reasonable assumptions, also commonly made in analysis of other related FL algorithms.
>
>
> **8. The generalization error bound is not well explained. For example, what is** $\epsilon_n$? **What does** $\lambda_i^*$ **mean? Why do you use Hellinger distance? How is the error bound compared to (Mohri et al 2019)?**
>
> In the main context/theorem, we stated that $\lambda_i^* = \min_{\theta\in\Theta} ||f_\theta-f^i||_\infty^2$ is the best error within our backbone network family $\Theta$. The sequence $\epsilon_n$ is also defined in the more detailed version Theorem E.1 in the Appendix.
>
> Comparing our bound to that in (Mohri et al. 2019) shows that our bound is much tighter. The Mohri bound is a Rademacher bound, which means that the bound is exponential in the depth of the network. This means that it is completely loose for any deep network. Our bound contains no terms that are exponential in network complexity, and thus is significantly tighter.
>
>
> **9. The empirical evaluation can be improved. The dataset is mainly CIFAR-100 and its variant, and only 4 baselines are compared. More Bayesian approaches should be compared, such as FedPA, FedBE, FedEM, and PredictiveBayes.**
>
> Comparison with the four latest probabilistic/Bayesian FL approaches have been added in the revised paper. They are: *FedPA*, the maximum-a-posterior method based on the product-of-experts model, 2) *FedBE* (Chen & Chao, 2021), the Bayesian ensemble method where the ensemble components are sampled from a Gaussian global model that is estimated from local client models, 3) *FedEM* (Marfoq et al., 2021) which aims to identify local client data distributions as mixtures of a fixed number of base distributions with different mixing proportions, and 4) *FedPop* (Kotelevskii et al., 2022).
>
> Moreover, we have conducted extra experiments on other FL benchmarks: MNIST, Fahsion-MNIST, and EMNIST. Please see Table 9,10,11,12 in Appendix H of the revised paper.
>
>
> **10.  About the reproducibility, the authors attached the code but it cannot be run. There is no package imported and the data collection process is not clear. Therefore, the reproducibility of this paper cannot be checked.**
>
> The code we attached as supplementary material was mainly for the purposed of conveying implementation details of the proposed ideas. We plan to publicize our full code to reproduce the results in the paper should the paper be accepted.

---

> ### Author Response · Authors · 2022-11-18
> **Responses to Reviewer 7vmX (Part 1)**
>
>
> **1. Some claims are not well-supported. For example: Sec 1, paragraph 1: ``FedAvg and FedProx are well-known to suffer convergence issues''. Any reference? What are the assumptions? The convergence of the proposed methods has $O(1/\sqrt{T})$ convergence rate. This is at the same level as FedAvg (see e.g. [R1]). Therefore the proposed method has no advantage over FedAvg which the authors claim to have some convergence issues.**
>
> First, our statement about the convergence issues of FedAvg/Prox mainly originates from some existing works [R2,R3,R4], where FedAvg/Prox algorithms take special account of data heterogeneity, local iterations per round, and the participation rate. However, as the reviewer pointed with [R1], asymptotically the convergence rate of our method is the same as that of FedAvg, so we agree to remove this statement. Instead, we prefer emphasize the benefit of the proposed Bayesian approach in terms of generalisation error bound -- Whereas non-Bayesian methods including FedAvg usually have weaker bounds relying on the  Rademacher complexity or VC dimension (the generalisation error bound is exponential in model complexity, and thus are very loose for deep networks), our (Bayesian) approach has tighter bounds (the model complexity term appears as a linear form, e.g., $r_n$ in Eq.(126-127)), thanks to the regression analysis technique.
>
> [R1] Adaptive federated optimization. Reddi et al., ICLR 2021.
>
> [R2] On the Convergence of FedAvg on Non-IID Data, X. Li et al., 2019
>
> [R3] SCAFFOLD: Stochastic Controlled Averaging for Federated Learning, S. P. Karimireddy et al., 2019
>
> [R4] Tackling the Objective Inconsistency Problem in Heterogeneous Federated Optimization, J. Wang et al., 2020
>
>
> **2. Sec 1, paragraph 2: why would Bayesian perspective be good for personalization? Is the current method complete or principled? Despite this being a bit subjective, I think at least the choices of NIW and Mix. are not principled and ad hoc.**
>
> The main benefit of our Bayesian approach in personalisation against non-Bayesian ones was discussed in the paragraph titled Personalisation in page 3 -- Many existing (non-Bayesian) approaches simply resort to finetuning, training on the personal data $D^p$ warm-starting with the FL-trained model. However, a potential issue is the lack of a solid principle on how to balance the initial FL-trained model and personal data fitting to avoid falling into either underfitting or overfitting regimes. In our Bayesian framework, personalisation can be seen as another posterior inference problem with additional evidence of the personal training data $D^p$.
>
> Meanwhile hierarchical Bayesian models are well established as being ideal for modeling data heterogeneity [R5,R6] including personalized inference [R7] in centralised learning. And a major contribution of ours is going beyond these prior works to bring this contribution to bear in deep federated learning.
>
> [R5] Latent Dirichlet Allocation, Blei et al., JMLR 2003
>
> [R6] C. Bishop, Pattern Recognition and Machine Learning, 2007.
>
> [R7] Efficient Bayesian hierarchical user modeling for recommendation system, Zhang and Koren, ACM SIGIR, 2007
>
>
> **3. What is the meaning of the higher level random variable? More intuitive explanation is needed.**
>
> The higher-level random variable $\phi$ determines the prior distribution of the individual random variables (client model parameters) $\theta_i$. Since each client's model $p(\theta_i|\phi)$ shares the same prior $\phi$, it indeed plays the role of the shared knowledge as the reviewer identified. It is exactly this principled decomposition of a global shared variable, and local client-specific random variables that makes the hierarchical Bayesian framework a principled and empirically effective solution to both global model learning and personalised model learning.
>
>
> **4. (Minor questions) Figure 1: what are $y^p$ and $x^p$? Sec 2.1: reference of ELBO and the original ref of block coordinate optimization are needed. Sec 2.2: the notation $p^p$ seems a bit weird as the two $p$'s have different meanings.**
>
> As described in the paragraph titled personalisation in page 3, the superscript $p$ stands for personalised data/distribution. But we have included this clarification in the caption of Fig. 1. About $p^p$: According to our definition, this means personalized (superscript) data distribution/probability (main text). We don't see this notation confusing since the notation is defined clearly. We have also included the references for ELBO in the revised manuscript.
>
>
> **5. Eq.10: what does the approximation sign mean? In which cases does the equality hold?**
>
> We described the meaning and quality of these approximations in the sentences that immediately follow Eq.(10), except for the second one, which is the obvious variational inference approximation, replacing the true posterior $p(\phi|D_{1:N})$ by the variational one $q(\phi)$.
>
>
> ---
> *(Continued in Part 2 below.)*

---

### Author Response · Authors · 2022-11-18
**Public thanks and summary to all reviewers and AC**

We thank all reviewers for their detailed and constructive comments/questions. We believe our feedback clearly addresses all reviewer concerns. We summarise the most salient points below for convenience:

**1. Empirical Comparisons (All Reviewers)**: We now provide more FL benchmarks (MNIST, FMNIST, EMNIST) as well as the existing CIFAR-100 and larger CIFAR-100-C. We also now also compare more FL algorithms especially Bayesian algorithms (FedPA, FedBE, FedEM, FedPop) and show improved performance over these. Please see Appendix H.

**2. Clarification of Details and how does Bayesian model help with FL challenges? (FDT8)**:  In summary, our FedHB tackles communication constraints and privacy by developing a hierarchical Bayesian model for which variational inference with block-coordinate descent naturally decomposes over the client and server variational posteriors (Sec 2.1). This means that data need not be exchanged
(privacy is maintained) and communication is only needed at convergence of each block (communication cost is low). Algorithmic details are now clarified in detail by Alg 2-10 in Appendix G.1. Table 6-8 of Appendix G.2 detail the modest computational/communication costs of FedHB.

**3. Clarification of claims (7vmX)**: We clarify that we are not claiming to be unique in having an $O(1/\sqrt{T})$ convergence rate, sorry for any confusion in this regard. However, most Bayesian FL models do not have clear convergence guarantees. We emphasise instead that we more uniquely have a strong generalisation bound (Thm 4.2), which is tighter than standard Rademacher or VC dimension bounds.

**4. Why is a Bayesian model particularly good for personalisation? (7vmX)**: Hierarchical Bayesian models are a canonical formalisation for modeling heterogenous data, including personalisation, as widely agreed upon in many seminal works (e.g., Latent Dirichlet Allocation, Blei et al. JMLR 2003). This is because they offer a principled way to decompose shared (global) and local  (personalised) knowledge and to learn both jointly. Furthermore, by making specific choices about the distributions involved hierarchical Bayesian models can be explicitly configured to model different types of data heterogeneity. E.g., when users group into clusters our mixture prior provides an ideal solution. This kind of transparent mapping between the algorithm and the nature of the data heterogeneity is not provided by other tools. The key contribution of our paper is to bring this seminal formalism together with state of the art  federated deep learning.

---

### Decision · Program_Chairs · 2023-01-20

**Decision:**

Reject

**Justification For Why Not Higher Score:**

However, practical implementation of the method could be better presented as well as its computational complexity. The paper would benefit from a better description of how the proposed solution addresses the specific problems of FL, in particular communication constrains and privacy. In the current version, it only focuses on personalization. The method does not address the problem of partial communication which is common in FL. Moreover, there are some studies that deal with the heterogeneous distribution in domain generalization and adaption, which is not covered by the analysis of related works. The empirical studies only compare the proposed algorithm with standard FL methods. The conclusions will be more convincing if more comparison is made with some approaches that deal with the heterogeneous distribution.

**Justification For Why Not Lower Score:**

N/A

**Metareview: Summary, Strengths And Weaknesses:**

This paper proposes a hierarchical Bayesian approach to Federated Learning. The proposed Bayesian model makes the block-coordinate descent solution becomes a distributed algorithm. In addition, the paper also derives convergence analysis that shows the block-coordinate FL algorithm converges to a local optimum of the objective at the rate of O(1/sqrt{t}), which is the same as the regular SGD; and a generalization error analysis that shows the test error is asymptotically optimal. The effectiveness of this approach is demonstrated empirically on two benchmark datasets (CIFAR-100 and CIFAR-C-100). The proposed approach outperforms the best work in terms of both global prediction and personalization.

The paper tackles the important problem of personalization in federated learning using a hierarchical Bayesian approach. The Bayesian approach to FL is interesting. Using the hierachical approach and variational inference, the authors propose two different choices of hierachical models and they work well in practice. The proposed Bayesian FL algorithms are supported by convergence analysis and generalization error bound.

However, practical implementation of the method could be better presented as well as its computational complexity. The paper would benefit from a better description of how the proposed solution addresses the specific problems of FL, in particular communication constrains and privacy. In the current version, it only focuses on personalization. The method does not address the problem of partial communication which is common in FL. Moreover, there are some studies that deal with the heterogeneous distribution in domain generalization and adaption, which is not covered by the analysis of related works. The empirical studies only compare the proposed algorithm with standard FL methods. The conclusions will be more convincing if more comparison is made with some approaches that deal with the heterogeneous distribution.